# Unveiling the Cognitive Compass: Theory-of-Mind-Guided Multimodal Emotion Reasoning

**Meng Luo**[1]*, **Bobo Li**[1], **Shanqing Xu**[2], **Shize Zhang**[1], **Qiuchan Chen**[2], **Menglu Han**[2],
**Wenhao Chen**[1], **Yanxiang Huang**[3], **Hao Fei**[1]†, **Mong-Li Lee**[1], **Wynne Hsu**[1]

[1]National University of Singapore
[2]Huazhong University of Science and Technology
[3]The Hong Kong Polytechnic University

## ABSTRACT

Despite rapid progress in multimodal large language models (MLLMs), their capability for deep emotional understanding remains limited. We argue that genuine affective intelligence requires explicit modeling of Theory of Mind (ToM), the cognitive substrate from which emotions arise. To this end, we introduce HitEmotion, a ToM-grounded hierarchical benchmark that diagnoses capability breakpoints across increasing levels of cognitive depth. Second, we propose a ToM-guided reasoning chain that tracks mental states and calibrates cross-modal evidence to achieve faithful emotional reasoning. We further introduce TMPO, a reinforcement learning method that uses intermediate mental states as process-level supervision to guide and strengthen model reasoning. Extensive experiments show that HitEmotion exposes deep emotional reasoning deficits in state-of-the-art models, especially on cognitively demanding tasks. In evaluation, the ToM-guided reasoning chain and TMPO improve end-task accuracy and yield more faithful, more coherent rationales. In conclusion, our work provides the research community with a practical toolkit for evaluating and enhancing the cognition-based emotional understanding capabilities of MLLMs. Our dataset and code are available at: https://HitEmotion.github.io/.

## 1 INTRODUCTION

Emotional intelligence (Picard, 2000) lies at the heart of machine intelligence and plays a pivotal role in the development of human-centric AI systems. Despite the remarkable progress of Multimodal Large Language Models (MLLMs, (OpenAI, 2025; Gemini et al., 2023; Li et al., 2024c)) across various tasks, their capability in deep emotional understanding remain suboptimal (Huang et al., 2023; Sabour et al., 2024). Existing studies primarily focus on surface-level emotion recognition, often neglecting the dynamic, context-dependent nature of emotions and their intricate relationships with other mental states such as beliefs and intentions (Khare et al., 2024). Such oversimplification overlooks the complexity of human affect, limiting the interpretability and performance of MLLMs in emotional understanding (Zhang et al., 2025c;d; Luo et al., 2024b; Kang et al., 2025).

Recent benchmarks such as EmoBench (Sabour et al., 2024) and EmotionHallucer (Xing et al., 2025) have empirically validated this bottleneck: even state-of-the-art (SOTA) MLLMs struggle with emotionally complex tasks that require nuanced perspective-taking or reasoning over conflicting multimodal cues (Yang et al., 2024). These failures often manifest as emotional hallucinations and other distortions. However, while these evaluations (Hu et al., 2025; Huang et al., 2023) successfully expose the symptoms, their own fragmented task design limits deeper diagnosis of the underlying causes. We argue that the core limitation of current evaluation paradigms lies in the absence of a unified cognitive framework, a veritable Cognitive Compass, to guide the evaluation of mental state reasoning (Chen et al., 2025). Specifically, they fail to organize emotional reasoning tasks according to developmental levels of Theory of Mind (ToM, (Lake et al., 2017))—e.g.,

---

*Email: mluo@u.nus.edu
†Corresponding author: Hao Fei (haofei37@nus.edu.sg)

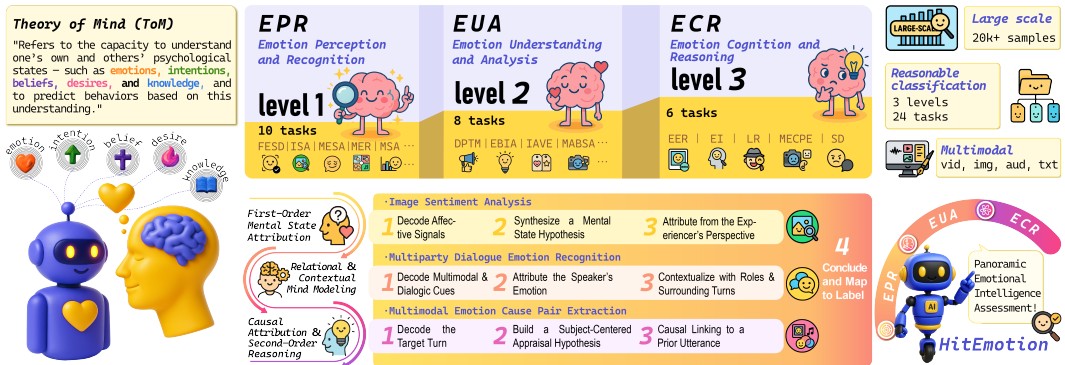

Figure 1: **Overview of our HitEmotion benchmark.**

first-order belief inference (Wimmer & Perner, 1983) vs. second-order recursive reasoning (Perner & Wimmer, 1985). Without such a compass, benchmarks provide only a coarse overall score and cannot pinpoint the exact ceiling or breaking point of a model's reasoning capacity.

This lack of precision in evaluation, in turn, hides fundamental flaws in the models' reasoning process. Even with a generic Chain-of-Thought (CoT, (Wei et al., 2022)) approach, the reasoning abilities of MLLMs (Fei et al., 2025; Luo et al., 2025) tend to emerge from general properties rather than from cognition-specific, supervised training. This leads to reasoning chains that often look coherent but are ultimately unfaithful. Specific problems include substituting causal attribution with simple template matching, being highly sensitive to small changes in wording and prompts, lacking robust ways to update in response to counterfactuals, and failing to explicitly track or maintain consistency among intermediate mental states like beliefs and intentions. Ultimately, these systems are measuring a shallow emotional fact retriever, skilled only at mapping superficial cues, rather than a deep mental state simulator capable of inferring the complex interplay between mind and emotion.

To tackle these dual shortcomings in evaluation and reasoning, and to help shift the paradigm in emotion understanding from fact retrieval to mental simulation, this study makes two core contributions. ❶ A **hi**erarchical, **T**heory-of-Mind–based benchmark for multimodal **emotion** understanding (**HitEmotion**). As presented in Figure 1, HitEmotion systematically arranges evaluation tasks into three cognitive levels of increasing depth: Emotion Perception and Recognition, Emotion Understanding and Analysis, and Emotion Cognition and Reasoning. This hierarchical structure is designed to precisely pinpoint and measure a model's capability breakpoints at different cognitive depths. ❷ A novel framework for **T**heory-of-**M**ind reasoning chain **p**reference **o**ptimization (**TMPO**). This approach begins by designing structured reasoning templates for specific tasks based on ToM principles. It then pioneers the use of intermediate mental states from these reasoning chains as both supervisory signals and reward sources for reinforcement learning. The method aims to shift a model's reasoning from a "general emergent" ability to a "domain-acquired" skill, significantly boosting its performance, robustness, and auditability in complex situations.

To validate the proposed framework, the first step was to construct a new evaluation resource. We curated 24 diverse datasets spanning sentiment, humor, sarcasm, and causal reasoning, and systematically cleaned and re-purposed them. Following the cognitive hierarchy of the proposed ToM framework, these datasets were rigorously restructured and aligned to create a benchmark capable of fine-grained assessment of model capabilities. Extensive experiments on our benchmark yielded three key findings. ❶ The performance of baseline models decisively substantiated our critique. Even SOTA MLLMs performed inconsistently across tasks and exhibited profound deficiencies at the highest tier of our framework. ❷ ToM reasoning chain prompting by itself shows considerable potential. When used simply as a prompting strategy, it significantly improves the performance of powerful closed-source models, offering initial proof of ToM's effectiveness as a reasoning "scaffold." ❸ The TMPO optimization delivers significant and consistent improvements across all evaluation tasks. It not only scores higher than most baseline models but also generates reasoning chains with demonstrably greater faithfulness and logical consistency, highlighting the advantages of the "domain acquisition" approach. In conclusion, the HitEmotion benchmark and TMPO method offer the research community a powerful toolkit for evaluating and advancing the deep emotional intelligence of MLLMs, facilitating the development of genuinely empathetic AI systems.

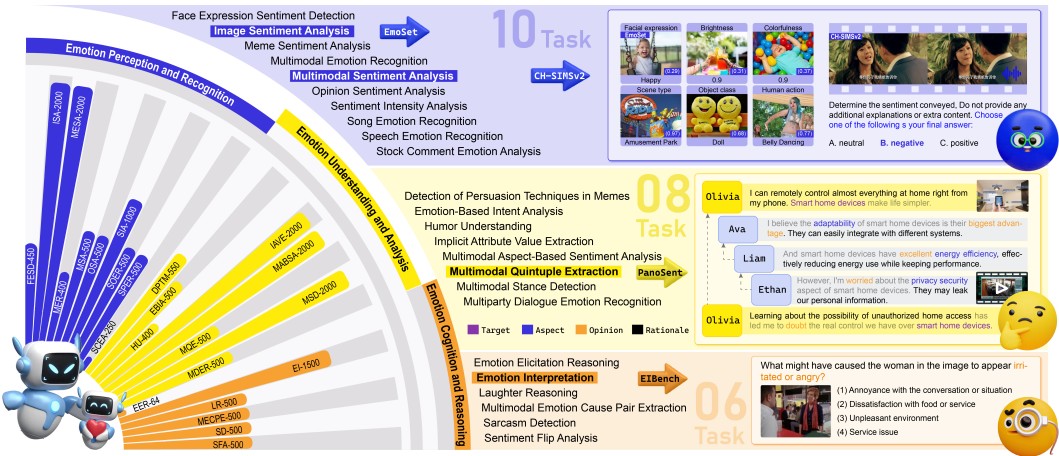

Figure 2: **Task taxonomy and examples in our HitEmotion benchmark.**

Table 1: **Comparison with other benchmarks related to emotional intelligence**. Psych-based indicates grounding in psychological theory; Rea-chain indicates whether reasoning traces are provided; Rationale indicates whether model rationales are included.

| Benchmark | # Task | Modality | # Instances | Type | Psy-based | Rea-chain | Rationale |
|---|---|---|---|---|---|---|---|
| EQ-bench (Paech, 2023b) | 1 | Text | 60 | Open-ended | ✓ | ✗ | ✗ |
| EmotionBench (Huang et al., 2023) | 1 | Text | 428 | Open-ended | ✓ | ✗ | ✗ |
| EmoBench (Sabour et al., 2024) | 4 | Text | 400 | MCQ | ✓ | ✗ | ✗ |
| MOSABench (Song et al., 2024b) | 3 | Image | 1,047 | MCQ | ✗ | ✗ | ✗ |
| MM-InstructEval (Yang et al., 2025) | 6 | Image | 34,602 | MCQ | ✗ | ✗ | ✗ |
| EmoBench-M (Hu et al., 2025) | 13 | Video | 5,646 | MCQ, Open-ended | ✓ | ✗ | ✗ |
| MER-UniBench (Lian et al., 2025) | 3 | Video | 12,799 | Open-ended | ✗ | ✗ | ✗ |
| EmotionHallucer (Xing et al., 2025) | 7 | Video, Image | 2,742 | Binary QA | ✓ | ✗ | ✗ |
| MME-Emotion (Zhang et al., 2025b) | 8 | Video | 6,500 | Open-ended | ✗ | ✓ | ✓ |
| HitEmotion (Ours) | 24 | Video, Image | 20,114 | MCQ, Open-ended | ✓ | ✓ | ✓ |

## 2 RELATED WORK

**Multimodal Affective Computing.** Multimodal Affective Computing aims to understand human emotions by learning cross-modal representations from heterogeneous signals like language, vision, and acoustics (Ramirez et al., 2011; Jiang et al., 2021; Zhu et al., 2024b;a). Evolving from unimodal studies (Ji et al., 2020; Donnelly & Prestwich, 2022), the field now emphasizes unified alignment-and-fusion frameworks, a shift accelerated by large-scale pretraining (Gemini et al., 2023) and enabled by parameter-efficient adaptation (Houlsby et al., 2019). Fusion techniques have similarly advanced from early/late strategies (Tsai et al., 2019; Chen et al., 2024a) to more sophisticated intermediate schemes that deepen cross-modal interaction and yield more discriminative features (Luo et al., 2021; Zou et al., 2023). Recent work further refines how affect is modeled, for instance by treating emotion as inherently ordinal to better infer intensity (Mai et al., 2025) or by leveraging human actions as a sparse but highly credible signal for emotional understanding (Yu et al., 2025).

**Evaluation of Emotional Intelligence.** The evaluation of emotional intelligence has progressed from text-based queries to comprehensive multimodal benchmarks. Initial text-only assessments established reproducible formats for testing emotional inference and reasoning (EQ-Bench, (Paech, 2023a); EmotionBench, (Huang et al., 2023); EmoBench, (Sabour et al., 2024)). The focus has since expanded to multimodal settings that probe for more contextual understanding. Current benchmarks assess a wide range of capabilities, including multi-object sentiment analysis (MOSABench; (Song et al., 2024b)), instruction-following (MM-InstructEval; (Yang et al., 2025)), hierarchical skills from recognition to social awareness (EmoBench-M; (Hu et al., 2025)), and unified evaluation of classic and free-form responses (MER-UniBench; (Lian et al., 2025)). Complementary work explicitly audits emotion-related hallucinations (EmotionHallucer; (Xing et al., 2025)) or provides holistic, multi-agent scoring across diverse scenarios (MME-Emotion; (Zhang et al., 2025b)). As detailed in Table 1, while extensive, prior benchmarks offer a fragmented evaluation. We are the first to connect psychological theory with the model's reasoning process and its ability to generate rationales, thereby providing a unified evaluation framework.

Table 2: **Evaluation tasks of our HitEmotion benchmark.** "N-CLS" denotes an n-class classification task, and "GEN" represents a generation task. "ACC", "MF", "WAF" and "EMF" denote accuracy, micro F1 score, weighted average F1-score and exact match F1, respectively.

| Task Name | Data Source | Type | #Instances | Metric |
|---|---|---|---|---|
| *Level 1: Emotion Perception and Recognition* | | | | |
| Face Expression Sentiment Detection (FESD) | CH-SIMS | 3-CLS | 450 | ACC, WAF |
| Image Sentiment Analysis (ISA) | EmoSet | 8-CLS | 2,000 | ACC, WAF |
| Meme Sentiment Analysis (MESA) | Memotion | 5-CLS | 2,000 | ACC, WAF |
| Multimodal Emotion Recognition (MER) | MER2023 | 6-CLS | 400 | ACC, WAF |
| Multimodal Sentiment Analysis (MSA) | CH-SIMSv2 | 3-CLS | 500 | ACC, WAF |
| Opinion Sentiment Analysis (OSA) | CMU-MOSI | 3-CLS | 500 | ACC, WAF |
| Sentiment Intensity Analysis (SIA) | CMU-MOSEI | 7-CLS | 1,000 | ACC, WAF |
| Song Emotion Recognition (SOER) | RAVDESS | 6-CLS | 500 | ACC, WAF |
| Speech Emotion Recognition (SPER) | RAVDESS | 8-CLS | 500 | ACC, WAF |
| Stock Comment Emotion Analysis (SCEA) | FMSA-SC | 5-CLS | 250 | ACC, WAF |
| *Level 2: Emotion Understanding and Analysis* | | | | |
| Detection of Persuasion Techniques in Memes (DPTM) | SemEval-2021 Task 6 | Multi-label | 550 | MF |
| Emotion-Based Intent Analysis (EBIA) | MC-EIU | 7-&8-CLS | 500 | ACC |
| Humor Understanding (HU) | UR-FUNNY | 2-CLS | 400 | ACC, WAF |
| Implicit Attribute Value Extraction (IAVE) | ImplicitAVE | N-CLS | 2,000 | ACC, WAF |
| Multimodal Aspect-Based Sentiment Analysis (MABSA) | Twitter2015/2017 | 3-CLS | 2,000 | MF |
| Multimodal Quintuple Extraction (MQE) | PanoSent | GEN | 500 | MF |
| Multimodal Stance Detection (MSD) | MMWTWT | 4-CLS | 2,000 | ACC |
| Multiparty Dialogue Emotion Recognition (MDER) | MELD | 7-CLS | 500 | ACC, WAF |
| *Level 3: Emotion Cognition and Reasoning* | | | | |
| Emotion Elicitation Reasoning (EER) | FilmStim | 7-CLS | 64 | ACC, WAF |
| Emotion Interpretation (EI) | EIBench | GEN | 1,500 | LLM |
| Laughter Reasoning (LR) | SMILE | GEN | 500 | LLM |
| Multimodal Emotion Cause Pair Extraction (MECPE) | ECF | GEN | 500 | MF |
| Sarcasm Detection (SD) | MUStARD | 2-CLS | 500 | ACC, WAF |
| Sentiment Flip Analysis (SFA) | PanoSent | GEN | 500 | EMF |

**Theory-of-Mind Reasoning.** ToM is the capacity to represent and infer others' mental states such as beliefs, intentions, and emotions (Premack & Woodruff, 1978; Baron-Cohen et al., 1985; Decety & Jackson, 2004; Lake et al., 2017), providing a cognitive foundation for affective computing. Psychology formalizes emotion as a core ToM dimension (Beaudoin et al., 2020; Chen et al., 2024b) and shows that tracking mental states is crucial for its attribution (Lillard, 1993; Qu et al., 2015). These insights have inspired inference-time strategies for LLMs that decompose ToM queries into tractable subproblems, such as simulating perspectives and checking knowledge access, thereby improving reasoning without extra training (Wei et al., 2022; Sarangi et al., 2025; Rahwan et al., 2019). ToM reasoning also extends to temporal, counterfactual, and non-literal communication, which are vital for affect interpretation (Byrne, 2017). However, recent benchmarks reveal that even state-of-the-art MLLMs still lack robust ToM capabilities (ToMBench, (Chen et al., 2024b); MMToM-QA, (Jin et al., 2024)), highlighting a key challenge. Consequently, the absence of targeted optimization in current models hinders the acquisition of more robust and complex ToM reasoning.

## 3 HITEMOTION BENCHMARK

### 3.1 TASK TAXONOMY

As shown in Figure 2, we organize our benchmark into three hierarchical levels of emotional intelligence, each targeting progressively advanced capabilities (see Appendix C for more details). ❶ **Emotion Perception and Recognition (EPR)** establishes the foundation by evaluating models' ability to perceive and classify explicit emotional states across modalities, mapping multimodal inputs to predefined categories. ❷ **Emotion Understanding and Analysis (EUA)** requires contextual awareness and relational reasoning, emphasizing the interpretation of emotions' functions and intents in situational settings. ❸ **Emotion Cognition and Reasoning (ECR)** advances to causal and second-order reasoning, requiring models to explain emotion causes, track temporal dynamics, and interpret nuanced expressions, thereby engaging with the cognitive processes underlying emotions.

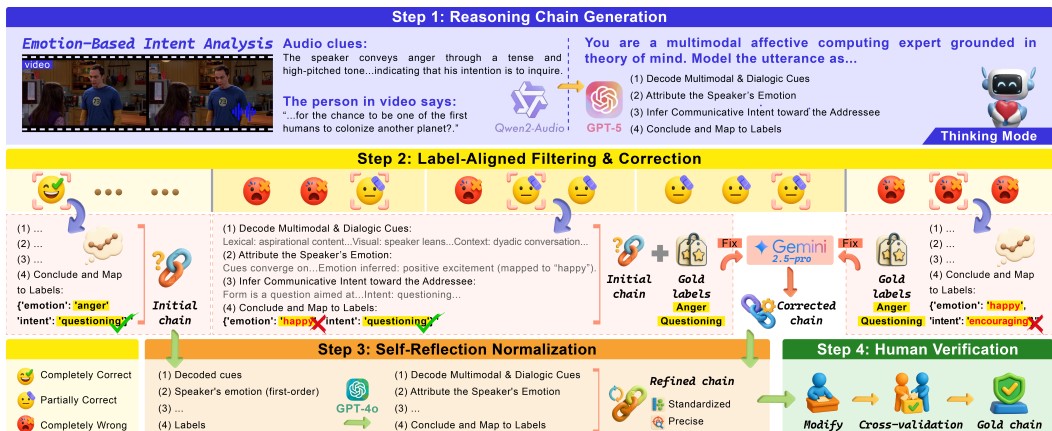

Figure 3: **Our reasoning chain curation pipeline.**

## 3.2 BENCHMARK CONSTRUCTION

To curate data and construct our benchmark while reducing annotation costs, we leverage publicly available datasets from the field of multimodal affective computing with task-specific annotations. Building on this foundation, we aggregate and curate 24 publicly available datasets spanning diverse affective domains, including emotion recognition, sentiment analysis, humor understanding, and causal reasoning. As shown in Table 2, these datasets are systematically organized into a three-tiered hierarchy reflecting increasing cognitive complexity: Emotion Perception and Recognition, Emotion Understanding and Analysis, and Emotion Cognition and Reasoning. While preserving the original task names, data structures, and evaluation metrics, we unify all datasets into a standardized closed-label QA format. To ensure the benchmark's integrity and reliability, we implement two critical enhancements without altering the source semantics. First, we institute a rigorous quality assurance protocol, wherein a stratified sample representing one-third of each dataset undergo a dual-annotator cross-review and arbitration process to validate the consistency of "prompt-answer-context" triplets. Second, to prevent data leakage and ensure a fair evaluation, we exclusively incorporate the official test splits from each source dataset. This meticulous curation process yields a benchmark with high internal consistency and a more uniform label distribution, providing a robust and systematic environment for assessing the affective capabilities of MLLMs.

## 4 METHODOLOGY

This section introduces TMPO, our framework for enhancing emotional understanding in MLLMs. We organize this section into four core components: task definition, ToM based prompting, a supervised fine-tuning stage, and ToM preference optimization.

### 4.1 TASK DEFINITION

Our objective is to leverage a Multimodal Large Language Model (MLLM) to infer an emotion-related output ($o$) and the underlying cognitive reasoning chain ($\tau$) from multimodal inputs (Text $T$, Audio $A$, and Video $V$). This task can be formally represented as a mapping: $(T, A, V) \rightarrow (\tau, o)$. Since ground-truth reasoning chains ($\tau$) are unavailable in existing datasets, we construct a gold-standard version to guide model generation. As illustrated in Figure 3, this is achieved through a strict four-step pipeline involving LLM-driven generation, filtering, enhancement, and correction (see Appendix D.4 for details).

### 4.2 TOM-STYLE PROMPTING MECHANISM

To elicit the desired reasoning chain $\tau$, we utilize a ToM-style prompting mechanism, denoted as a task-specific prompt $\mathcal{P}$, to structure the expected output format. Our prompt $\mathcal{P}$ is structured across three levels of cognitive complexity to elicit increasingly sophisticated reasoning chains. For concrete examples of these prompts, please refer to Figures 24 through 47 in Appendix F.

**Level 1: First-Order Mental State Attribution.** Prompts at this level guide the model to map multimodal cues to an immediate emotional state. This involves synthesizing observable signals into a first-order attribution of what the subject feels, while remaining flexible to task-specific modalities like text-image incongruities in memes.

**Level 2: Relational & Contextual Mind Modeling.** This level requires reasoning about the relationship between an emotional state and its context, such as a specific entity or communicative goal. It builds upon Level 1 attributions by contextualizing them, for example, by linking an emotion to a specific target in aspect-based sentiment analysis.

**Level 3: Causal Attribution & Second-Order Reasoning.** The highest level elicits reasoning about the causes of emotions and their social interpretation, involving causal inference and second-order ToM, which involves inferring what others believe about a subject's state. Prompts guide the model to explain why an emotion arises or detect incongruity between literal and intended meaning, as in sarcasm, moving beyond what is felt to why it is felt and how it is meant to be interpreted.

### 4.3 STAGE 1: TOM-ALIGNED SUPERVISED FINE-TUNING

To establish a foundational capability for structured reasoning, we first perform SFT on a multimodal backbone model. The core objective of this stage is to teach the model to generate responses that not only produce the correct task-specific output but also articulate the underlying cognitive process in a clear, step-by-step manner. We explicitly wrap the intermediate reasoning steps ($\tau$) with a `<think></think>` tag and encapsulate the final task-specific output ($o$) within an `<answer></answer>` tag. This structural disentanglement forces the model to learn the distinct functions of cognitive deliberation and final conclusion generation. The model is trained on the native multimodal inputs ($T, A, V$) along with a task-specific prompt $\mathcal{P}$. The target for the model is to generate the complete structured string $y = $ `<think>`$\tau$`</think><answer>`$o$`</answer>`. The fine-tuning objective is to minimize the standard negative log-likelihood loss over our dataset:

$$\mathcal{L}_{\text{SFT}}(\theta) = -\mathbb{E}_{((\mathcal{P},T,A,V),y)}[\log \pi_\theta(y|\mathcal{P},T,A,V)] \tag{1}$$

where $\pi_\theta$ is the policy of the MLLM with parameters $\theta$. After this stage, the model acquires a preliminary ability to mimic structured, multi-step reasoning patterns from our curated data.

### 4.4 STAGE 2: TOM-BASED PREFERENCE OPTIMIZATION WITH GRPO

While SFT imparts the basic structure of ToM-aligned reasoning, the generated chains may still lack factual grounding, exhibit logical inconsistencies, or fail to generalize robustly across diverse scenarios. To overcome these limitations, we further refine the model using Group-wise Reward Policy Optimization (GRPO) (DeepSeek, 2025), which enhances the model's ability to generate reasoning chains that are structurally correct as well as cognitively plausible and factually accurate.

The GRPO process begins by sampling $N$ candidate outputs $\{y_1, y_2, \cdots, y_N\}$ from our current policy for a given prompt, where each $y_i = $ `<think>`$\tau_i$`</think><answer>`$o_i$`</answer>`. Each candidate is then evaluated using a custom-designed, multi-dimensional reward function $R(y)$. The resulting scores guide the policy update via the GRPO objective:

$$\max_{\pi_\theta} \mathbb{E}_{y_i \sim \pi_{\text{old}}} \left[ \frac{\pi_\theta(y_i)}{\pi_{\text{old}}(y_i)} A_i \right] - \beta D_{KL}(\pi_\theta \| \pi_{\text{ref}}) \tag{2}$$

where $A_i$ are the computed normalized advantage scores and the KL-divergence term penalizes deviation from a reference policy $\pi_{\text{ref}}$ (typically the initial SFT model) to stabilize the optimization process. The advantage scores $A_i$ are derived from the relative ranking or value of the rewards $R(y_i)$ within the sampled group, guiding the model to prefer higher-scoring responses.

#### 4.4.1 REWARD ASSIGNMENT

The cornerstone of our GRPO strategy is a comprehensive reward function $R(y)$ that decomposes the quality of a response into four distinct, complementary components. This function is formulated as a weighted sum:

$$R(y) = \mu_1 R_{\text{structure}} + \mu_2 R_{\text{content}} + \mu_3 R_{\text{process}} + \mu_4 R_{\text{consistency}} \tag{3}$$

Table 3: Performance on **Emotion Perception and Recognition**, with ACC as the evaluation metric. **Bold** and underlined indicate the best and the worst results among all models, respectively.

| Category | Model | FESD | ISA | MESA | MER | MSA | OSA | SIA | SOER | SPER | SCEA |
|---|---|---|---|---|---|---|---|---|---|---|---|
| Open Source | VideoLLaMA3-7B | 61.78 | 46.85 | 21.60 | 52.18 | 64.62 | 67.89 | 35.20 | 45.80 | 41.80 | 42.00 |
| | LLaVA-One-Vision-7B | 63.44 | 49.19 | 17.05 | 39.50 | 65.40 | 63.00 | 27.00 | 53.40 | 44.60 | 34.80 |
| | LLaVA-NeXT-Video-7B | 54.44 | 41.20 | 11.85 | 41.31 | 56.11 | 65.80 | 25.03 | 48.60 | 43.40 | 31.20 |
| | Qwen2.5-VL-7B | 62.00 | 43.15 | 21.25 | 56.75 | 61.21 | 64.20 | 32.60 | 52.80 | 41.80 | 47.20 |
| | InternVL3-8B | 62.33 | 50.65 | 21.40 | 53.00 | 63.80 | 68.00 | 31.20 | 49.00 | 42.60 | 48.60 |
| | MiniCPM-V-2.6-8B | 57.53 | 49.39 | 25.15 | 50.13 | 62.65 | 52.45 | 37.42 | 44.90 | 37.59 | 45.21 |
| | Qwen2.5-VL-32B | 63.78 | 53.70 | 25.28 | 57.14 | 65.80 | 68.80 | 34.20 | 43.40 | 41.80 | 47.60 |
| | InternVL3-38B | 63.22 | 53.58 | 24.00 | 57.16 | 68.80 | 68.80 | 35.73 | 53.46 | 48.00 | 50.60 |
| | R1-Omni-0.5B | 42.28 | 51.55 | 23.72 | 50.88 | 41.74 | 32.20 | 19.50 | 30.12 | 24.38 | 43.60 |
| | HumanOmni-7B | 64.44 | 53.77 | 23.82 | 56.75 | 48.20 | 35.20 | 33.90 | 50.31 | 46.20 | 47.60 |
| | Qwen2.5-Omni-7B | 64.67 | 51.56 | 22.71 | 56.08 | 64.00 | 68.00 | 32.30 | 54.72 | 44.60 | 48.60 |
| | Emotion-LLaMA-7B | 33.11 | 53.63 | 24.00 | 43.75 | 44.40 | 56.60 | 37.00 | 47.00 | 47.27 | 48.44 |
| | AffectGPT-7B | 66.67 | 50.33 | 25.46 | 38.69 | 66.60 | 67.76 | 34.50 | 49.19 | 41.25 | 38.80 |
| Closed Source | GPT-4o | 70.22 | 54.48 | 30.12 | 57.64 | 69.20 | 69.53 | 40.00 | 54.00 | 49.60 | 49.96 |
| | + ToM prompt | 74.00 (+3.78) | 56.44 (+1.96) | 33.18 (+3.06) | 63.32 (+5.68) | 74.60 (+5.40) | 73.87 (+7.34) | 41.34 (+1.34) | 56.10 (+2.10) | 54.31 (+4.71) | 52.80 (+2.84) |
| | GPT-4.1 | 71.46 | 56.80 | 31.43 | 64.00 | 72.46 | 69.60 | 40.81 | 66.19 | 55.20 | 53.20 |
| | + ToM prompt | 74.74 (+3.28) | 58.06 (+1.26) | 34.55 (+3.12) | 66.00 (+2.00) | 76.06 (+3.60) | 74.80 (+8.20) | 43.17 (+2.36) | 69.19 (+3.00) | 57.20 (+2.00) | 56.01 (+2.81) |
| | Gemini-2.5-Flash | 67.11 | 55.41 | 27.12 | 58.73 | 68.40 | 70.91 | 38.44 | 57.47 | 56.51 | 50.83 |
| | + ToM prompt | 76.44 (+9.33) | 59.01 (+3.60) | 29.20 (+2.08) | 64.19 (+5.46) | 74.84 (+6.44) | 76.03 (+5.12) | 43.36 (+4.92) | 63.33 (+5.86) | 62.22 (+5.71) | 53.04 (+2.21) |
| | Gemini-2.5-Pro | 78.39 | 61.12 | 28.96 | 72.11 | 74.20 | 75.71 | 46.53 | 67.96 | 65.00 | 55.02 |
| | + ToM prompt | **79.11** (+0.72) | 63.13 (+2.01) | 31.02 (+2.06) | 72.92 (+0.81) | 77.97 (+3.77) | **79.19** (+3.48) | 51.74 (+5.21) | **69.00** (+1.04) | **69.31** (+4.31) | **62.25** (+7.23) |
| Ours | TMPO (SFT) | 69.39 | 60.85 | 31.34 | 66.23 | 72.49 | 71.33 | 45.58 | 58.71 | 55.43 | 50.20 |
| | + GRPO | 77.12 | **67.63** | **37.18** | **75.41** | **79.12** | 77.03 | **53.91** | 66.13 | 65.91 | 58.74 |

These components evaluate the reasoning process from different perspectives: the **Structure Reward** ($R_{structure}$) enforces the correct sequence of reasoning steps; the **Content Reward** ($R_{content}$) evaluates the final answer's correctness; the **Process Reward** ($R_{process}$) encourages domain-specific language; and the **Consistency Reward** ($R_{consistency}$) penalizes logical and factual inconsistencies. The weights $\mu_{(*)}$ are calibrated to prioritize correctness and logical grounding. A full description of each component and the rationale for weight assignments are provided in the Appendix D.1.

Table 4: Performance on **Emotion Understanding and Analysis**. By default, ACC is used as the evaluation metric, while DPTM, MABSA, and MQE use the MF metric.

| Category | Model | DPTM | EBIA | HU | IAVE | MABSA | MQE | MSD | MDER |
|---|---|---|---|---|---|---|---|---|---|
| Open Source | VideoLLaMA3-7B | 31.17 | 14.42 | 44.89 | 62.50 | 61.96 | 23.67 | 51.15 | 42.61 |
| | LLaVA-One-Vision-7B | 31.54 | 11.33 | 42.25 | 60.37 | 63.89 | 14.02 | 39.75 | 33.00 |
| | LLaVA-NeXT-Video-7B | 31.28 | 12.37 | 43.50 | 40.50 | 59.40 | 13.45 | 44.75 | 25.25 |
| | Qwen2.5-VL-7B | 31.41 | 11.02 | 54.25 | 64.49 | 64.65 | 32.59 | 52.55 | 45.60 |
| | InternVL3-8B | 36.77 | 14.79 | 53.50 | 60.13 | 63.11 | 33.13 | 50.90 | 39.48 |
| | MiniCPM-V-2.6-8B | 30.80 | 14.51 | 49.25 | 55.03 | 61.82 | 27.77 | 39.85 | 43.51 |
| | Qwen2.5-VL-32B | 40.22 | 14.62 | 57.50 | 62.67 | 63.29 | 32.38 | 52.08 | 48.20 |
| | InternVL3-38B | 40.55 | 16.49 | 59.25 | 64.74 | 64.80 | 32.64 | 53.85 | 46.00 |
| | R1-Omni-0.5B | 37.54 | 13.45 | 46.25 | 50.03 | 58.40 | 29.58 | 47.85 | 29.81 |
| | HumanOmni-7B | 35.59 | 12.55 | 49.50 | 53.50 | 59.89 | 32.98 | 47.90 | 36.20 |
| | Qwen2.5-Omni-7B | 31.63 | 11.42 | 53.00 | 55.79 | 61.93 | 31.09 | 44.65 | 37.68 |
| | Emotion-LLaMA-7B | 39.54 | 15.46 | 57.92 | 52.08 | 60.13 | 34.18 | 44.15 | 47.59 |
| | AffectGPT-7B | 34.17 | 12.27 | 56.50 | 40.07 | 60.48 | 30.95 | 42.40 | 37.92 |
| Closed Source | GPT-4o | 42.33 | 17.45 | 60.00 | 66.13 | 64.76 | 35.32 | 55.76 | 49.68 |
| | + ToM prompt | 45.90 (+3.57) | 25.70 (+8.25) | 66.63 (+6.63) | 71.19 (+5.06) | 68.30 (+3.54) | 36.45 (+1.13) | 61.04 (+5.28) | 53.72 (+4.04) |
| | GPT-4.1 | 47.50 | 18.62 | 70.19 | 67.68 | 70.81 | 37.98 | 65.76 | 53.82 |
| | + ToM prompt | 49.47 (+1.97) | 27.65 (+9.03) | 78.00 (+7.81) | 72.91 (+5.23) | 77.70 (+6.89) | 40.91 (+2.93) | 66.18 (+0.42) | 57.85 (+4.03) |
| | Gemini-2.5-Flash | 47.18 | 16.34 | 64.66 | 64.14 | 66.71 | 36.55 | 57.65 | 51.41 |
| | + ToM prompt | 56.35 (+9.17) | 24.87 (+8.53) | 65.74 (+1.08) | 72.63 (+8.49) | 73.21 (+6.50) | 38.12 (+1.57) | 61.83 (+4.18) | 55.56 (+4.15) |
| | Gemini-2.5-Pro | 49.23 | 19.25 | 69.39 | 70.67 | 67.61 | 39.23 | 64.95 | 52.65 |
| | + ToM prompt | **59.21** (+9.98) | 28.68 (+9.43) | 71.83 (+2.44) | **77.20** (+6.53) | 75.43 (+7.82) | 44.47 (+5.24) | **73.79** (+8.84) | 58.90 (+6.25) |
| Ours | TMPO (SFT) | 46.42 | 23.11 | 68.40 | 64.18 | 69.47 | 37.24 | 60.45 | 51.83 |
| | + GRPO | 56.23 | **32.82** | **78.64** | 73.39 | **78.16** | **45.68** | 71.56 | **61.08** |

# 5 EXPERIMENTS

## 5.1 SETTINGS

We use Qwen2.5-Omni-7B as our base model, trained on 8 × NVIDIA A800 80 GB GPUs. For our reward function, the weights $\mu_1, \mu_2, \mu_3, \mu_4$ are set to 0.4, 1.0, 0.1, and 1.0, respectively. During training, videos are sampled into 16 frames. The model first undergoes SFT for two epochs with a learning rate of 1e-5, followed by our GRPO strategy with a learning rate of 1e-6. For evaluation, we select the checkpoint with the best validation performance and conduct a comprehensive assessment on both open-source models (0.5B to 38B parameters) and closed-source models (GPT and Gemini series). Further implementation details are provided in the Appendix D.3.

## 5.2 RESULTS AND ANALYSIS

The experimental results reveal significant limitations in the multimodal emotion analysis capabilities of current MLLMs. As shown in Tables 3–5, model performance is evaluated across the three

Table 5: Performance on **Emotion Cognition and Reasoning**. By default, ACC is used as the evaluation metric, while MECPE uses the MF metric and SFA uses the EMF metric.

| Category | Model | EER | EI | LR | MECPE | SD | SFA |
|---|---|---|---|---|---|---|---|
| Open Source | VideoLLaMA3-7B | 45.31 | 31.29 | 39.56 | 13.09 | 37.56 | 13.16 |
| | LLaVA-One-Vision-7B | 46.88 | 47.40 | 44.60 | 10.83 | 45.20 | 16.22 |
| | LLaVA-NeXT-Video-7B | 35.94 | 46.80 | 43.10 | 13.05 | 46.36 | 18.42 |
| | Qwen2.5-VL-7B | 40.62 | 50.53 | 48.20 | 15.07 | 49.00 | 14.64 |
| | InternVL3-8B | 50.00 | 47.00 | 46.40 | 16.41 | 51.40 | 17.61 |
| | MiniCPM-V-2.6-8B | 39.68 | 33.93 | 50.40 | 16.44 | 51.40 | 21.83 |
| | Qwen2.5-VL-32B | 54.69 | 53.40 | 53.40 | 19.60 | 55.60 | 23.79 |
| | InternVL3-38B | 50.31 | 50.67 | 51.40 | 19.28 | 55.80 | 25.73 |
| | R1-Omni-0.5B | 39.67 | 43.73 | 43.00 | 16.13 | 53.00 | 19.93 |
| | HumanOmni-7B | 38.85 | 47.93 | 28.40 | 13.19 | 49.40 | 16.43 |
| | Qwen2.5-Omni-7B | 51.25 | 48.67 | 49.20 | 13.83 | 53.40 | 17.76 |
| | Emotion-LLaMA-7B | 42.81 | 49.53 | 53.00 | 19.28 | 52.60 | 19.02 |
| | AffectGPT-7B | 43.75 | 46.27 | 50.40 | 10.81 | 52.73 | 15.12 |
| Closed Source | GPT-4o | 57.81 | 54.13 | 55.83 | 20.93 | 56.60 | 25.77 |
| | + ToM prompt | 60.00 (+2.19) | 64.33 (+10.20) | 66.00 (+10.17) | 22.48 (+1.55) | 64.80 (+8.20) | 42.29 (+16.52) |
| | GPT-4.1 | 60.31 | 57.67 | 61.04 | 26.86 | 66.20 | 36.73 |
| | + ToM prompt | 65.86 (+5.55) | 69.00 (+11.33) | 71.79 (+10.75) | 28.11 (+1.25) | 68.67 (+2.47) | 47.75 (+11.02) |
| | Gemini-2.5-Flash | 58.33 | 54.47 | 58.20 | 27.11 | 61.49 | 28.02 |
| | + ToM prompt | 64.13 (+5.80) | 63.93 (+9.46) | 66.60 (+8.40) | 31.43 (+4.32) | 64.10 (+2.61) | 45.22 (+17.20) |
| | Gemini-2.5-Pro | 66.13 | 65.13 | 59.23 | 33.33 | 66.61 | 41.22 |
| | + ToM prompt | 71.94 (+5.81) | 70.27 (+5.14) | 68.20 (+8.97) | 37.70 (+4.37) | 69.00 (+2.39) | 52.78 (+11.56) |
| Ours | TMPO (SFT) | 60.10 | 62.36 | 59.75 | 26.33 | 59.92 | 40.50 |
| | + GRPO | **73.13** | **72.27** | **72.45** | **39.34** | **70.13** | **54.16** |

Figure 4: **Average performance across our HitEmotion benchmark levels.** Comparison of 17 multimodal models on our HitEmotion benchmark, showing average scores for each level per model.

hierarchical task categories. At the foundational level of EPR, only three of the ten tasks—FESD, MSA, and OSA—yield average scores above 60. Even the best-performing model, Gemini-2.5-Pro, achieves 78.39 on FESD, 74.20 on MSA, and 75.71 on OSA, while most other models remain around the 50-point range, reflecting limited robustness. As task complexity increases, performance declines markedly. In EUA level, only two tasks surpass the 60-point threshold. Most critically, within the cognitively demanding ECR level, no task achieves an average score above 60. This clear performance hierarchy underscores our benchmark's ability to differentiate models across distinct levels of reasoning. Taken together, the findings show that current MLLMs possess only rudimentary emotional intelligence and continue to struggle with higher-order emotional reasoning, highlighting an urgent need for advances in both model architectures and training methodologies.

**Closed-Source, Tuned, and Scaled Models Lead in Emotional Intelligence.** As shown in Figure 4, proprietary models such as the GPT and Gemini series consistently outperform open-source counterparts, owing to their large parameter scales and extensive pretraining on diverse datasets. Even in the zero-shot setting, Gemini-2.5-Pro achieves 78.39 on the FESD task, while GPT-4.1 reaches 71.46, both substantially ahead of most open-source models. The Gemini series further surpass GPT in multimodal emotion recognition due to its native capacity for processing video and audio inputs. Nevertheless, open-source models, though constrained by scale, can achieve competitive results through task-specific fine-tuning. Emotion-LLaMA-7B attains 34.18 on the MQE task, outperforming most untuned baselines. While Emotion-LLaMA benefits from domain-specific fine-tuning, it still lags behind zero-shot proprietary models, indicating that a significant capability gap persists between existing open-source solutions and top-tier proprietary systems. Likewise,

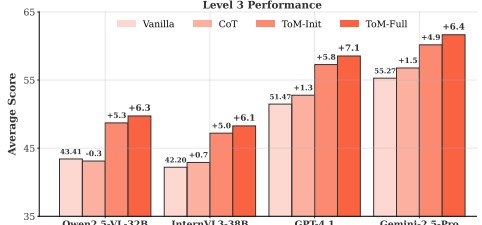

Figure 5: Ablation study on ToM-style prompting.

Table 6: Ablation study on reward components.

| Reward Components | | | | Average Score | | |
|---|---|---|---|---|---|---|
| $R_{structure}$ | $R_{content}$ | $R_{consistency}$ | $R_{process}$ | L1 | L2 | L3 |
| ✓ | - | - | - | 56.61 | 54.52 | 52.34 |
| ✓ | ✓ | - | - | 62.63 | 61.07 | 59.31 |
| ✓ | ✓ | ✓ | - | 65.12 | 64.03 | 62.82 |
| - | ✓ | ✓ | ✓ | 56.20 | 55.10 | 55.05 |
| ✓ | ✓ | ✓ | ✓ | **65.82** | **64.70** | **63.58** |

Qwen2.5-VL-32B achieves 53.40 on the LR task, closely approaching GPT-4o's 55.83. These findings show that large-scale pretraining provides the foundation for advanced emotion intelligence, but targeted fine-tuning offers open-source models a practical pathway to close the gap with proprietary systems and achieve broader accessibility.

**Effects of ToM Prompting on Emotional Intelligence.** Closed-source models such as GPT-4.1 and Gemini-2.5-Pro gain clear advantages when ToM prompting is applied, especially on the most challenging tasks. High-capacity open-source models, including Qwen2.5-VL-32B and InternVL3-38B, also benefit, with improvements observed across most tasks and effects becoming more pronounced at higher levels. This suggests that models with stronger baseline reasoning are better able to leverage intermediate reasoning chains provided by ToM. By contrast, weaker models don't exhibit consistent gains and in some cases deteriorate. For instance, VideoLLaMA3-7B drops from 61.78 to 54.18 on FESD under ToM prompting, and LLaVA-NeXT-Video-7B shows minimal improvement on IAVE, increasing only from 40.50 to 40.81. These outcomes imply that models with insufficient representational and reasoning capacity cannot stably exploit ToM, and are more prone to hallucinations that compromise the reasoning chain.

**TMPO Unlocks Advanced Reasoning Capabilities.** The experimental results consistently demonstrate the remarkable effectiveness of our TMPO framework, which provides a substantial performance uplift to the backbone model across all task categories. Both the SFT stage and the GRPO stage contribute to this success, with GRPO delivering a particularly significant boost in performance. Crucially, on more complex tasks requiring nuanced reasoning, our fully-optimized model not only closes the gap but often surpasses the performance of top-tier proprietary models, emerging as the top-performing model on 16 of the 24 tasks. This highlights TMPO's exceptional capability in teaching models how to reason. Conversely, for some direct, perception-driven tasks, such as inferring emotions mainly from facial expressions, our model still lags behind some leading systems. This is likely due to the inherent limitations in the base model's raw multimodal perception capabilities, which the reasoning-focused optimization cannot fully overcome.

## 5.3 Ablation Studies

**ToM-style Prompting.** To validate our prompt engineering, we conduct an ablation study on its key design choices. We compare three strategies: (1) **CoT**, which instructs the model to "please think step-by-step"; (2) **ToM-Init**, which establishes a cognitive reasoning path without specific terminological guidance; and (3) **ToM-Full**, which enhances *ToM-Init* by explicitly integrating task-relevant ToM keywords. The results in Figure 5 show that *ToM-Init* consistently outperforms the generic *CoT*, confirming the inherent benefit of a ToM-aligned framework over unguided reasoning. In addition, *ToM-Full* yields a further substantial performance gain over *ToM-Init*, validating that the explicit integration of key ToM concepts is crucial for unlocking the model's full reasoning potential.

**Reward Components.** To validate our reward function, we conduct a complementary ablation study on its individual components. We progressively add each reward to the GRPO objective, with results summarized in Table 6. Using only $R_{structure}$ establishes a baseline by enforcing a coherent format. The most substantial performance gain is observed with the introduction of $R_{content}$, underscoring the necessity of directly optimizing for the correct final answer. Furthermore, integrating $R_{consistency}$ yields another significant boost, validating its crucial role in eliminating logical fallacies and grounding the reasoning. Finally, $R_{process}$ provides a complementary refinement by encouraging the use of ToM-specific terminology. This progressive enhancement demonstrates that all four components work in synergy to produce high-quality, reliable, and cognitively aligned reasoning. Additionally, we investigate the necessity of $R_{structure}$ by removing it from the full configuration. This exclusion

leads to a significant performance drop. Without the explicit structural penalty, the model exhibits Format Collapse, failing to adhere to the XML schema required for extracting answers. This confirms that $R_{\text{structure}}$ acts as the foundational prerequisite that enables the effective optimization of other reward components.

## 6 CONCLUSION

In this work, we introduce **HitEmotion**, a hierarchical benchmark that systematically diagnoses MLLM's capability breakpoints across increasing cognitive depths. To improve reasoning, we develop **TMPO**, a novel preference optimization method that uses intermediate mental states as process-level supervision. Our experiments confirm that HitEmotion exposes deep reasoning deficits even in top-tier models, while TMPO substantially boosts the backbone model's performance. The optimized model surpasses leading proprietary systems on many cognitively demanding tasks by improving end-task accuracy, faithfulness, and the coherence of its reasoning. Together, HitEmotion and TMPO form a robust toolkit for evaluating and enhancing cognitive-based affective intelligence. This approach pushes MLLMs beyond superficial recognition toward a deeper, more human-like mental state simulation, facilitating the development of more empathetic AI.

## 7 ETHICS STATEMENT

This work relies exclusively on publicly available datasets released by prior publications. We did not collect new human-subject data, and no personally identifiable information was used. All datasets were used in accordance with their original licenses. No institutional ethics review was required. We adhere to the ICLR Code of Ethics.

## 8 REPRODUCIBILITY STATEMENT

To facilitate reproducibility, we have released all data preprocessing scripts, model training and inference code through an anonymous repository, with the link provided in the Abstract. In addition, we have uploaded the dataset samples used in our experiments, together with detailed configuration files. We further provide a comprehensive description of our experimental setup, including model architecture, training methodology, and hyperparameter settings in Section 5.1 and Appendix D.3. These resources ensure that the experimental results in this paper can be faithfully reproduced.

## ACKNOWLEDGMENTS

This work is supported by the Ministry of Education, Singapore, under its MOE AcRF TIER 3 Grant (MOE-MOET32022-0001).

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

APPENDIX OVERVIEW

- Appendix §A outlines how LLMs are utilized in the paper.
- Appendix §B summarizes the limitations and future work of this work.
- Appendix §C presents the taxonomy of the tasks and dataset details.
- Appendix §D provides additional implementation details.
- Appendix §E presents the extended experimental results.
- Appendix §F describes the design of ToM-style prompts.
- Appendix §G presents representative samples from each dataset used.
- Appendix §H presents representative case studies from multiple perspectives.

## A    THE USE OF LLMS

In this work, LLMs are employed as an auxiliary tool for language editing. Their use is limited to grammar correction, refinement of sentence structure, and improvements in textual fluency and readability. LLMs are not used for any substantive aspects of the research, including the design of methodology, execution of experiments, data analysis, or interpretation of results.

## B    LIMITATIONS AND FUTURE WORK

While our TMPO framework demonstrates significant advancements in multimodal emotion reasoning, we acknowledge certain limitations that outline directions for future research.

**Scope of Applicability.** Our framework is grounded in Theory of Mind (ToM), which refers to the ability to attribute mental states—such as emotions, intentions, and beliefs—to oneself and others. In essence, ToM involves "putting yourself in someone else's shoes" to infer hidden mental states. This stands in stark contrast to domains like mathematics, coding, and logical puzzles, where well-defined ground-truth answers are readily available, enabling objective verification. Social reasoning, however, is characterized by its information-asymmetric nature and increased uncertainty, where objective answers are not easily obtainable (Chang et al., 2025; Yuan et al., 2025b;a; Liao et al., 2026; Zheng et al., 2026; An et al., 2026; Lin et al., 2025a; An et al., 2025; 2024). Consequently, TMPO is tailored specifically for these social complexity challenges. It holds strong promise for broader Social Intelligence domains (e.g., negotiation, intent analysis) that rely on the same underlying cognitive mechanisms.

**Base Model and Modality Constraints.** Our choice of the 7B parameter backbone was driven by the necessity for native omni-modal processing (Video+Audio+Text), as audio is critical for emotion perception. Currently, few open-source models larger than 7B support native audio-visual integration. While TMPO significantly boosts cognitive reasoning (Level 2 & 3 tasks), the performance on direct perception tasks (Level 1) remains bounded by the inherent sensory quality of the base encoders. As larger omni-modal models become available, we anticipate that scaling up TMPO will yield further gains, leveraging the stronger reasoning priors of large-scale backbones.

**Computational Efficiency.** Despite using a compact 7B model, our approach achieves performance competitive with proprietary systems (e.g., Gemini-2.5-pro). This highlights the efficiency of our method: by optimizing the reasoning process via RL, we extract maximal cognitive intelligence from a lightweight architecture, offering a practical solution for resource-constrained deployment.

## C    TASK TAXONOMY

**Level 1: Emotion Perception and Recognition.** This level forms the foundation of emotion intelligence; its core function is to directly identify and classify explicit emotional states across modalities. This layer evaluates MLLMs' ability to accurately extract and integrate emotional information from multimodal inputs and map it to predefined emotion categories. This capability is a prerequisite for higher-level EI competencies. Specific evaluation tasks draw on several specialized datasets. For Facial Expression Sentiment Detection, CH-SIMS (Yu et al., 2020) provides Chinese video clips

with fine-grained multimodal annotations to assess models' capacity for integrated perception of visual and linguistic emotions in realistic scenarios. EmoSet (Yang et al., 2023) is a large-scale image-sentiment dataset that focuses on recognizing emotions from static visual content. To capture internet-specific expressions, Meme Sentiment Analysis employs the Memotion dataset (Mishra et al., 2023), which contains a large collection of memes annotated for sentiment and humor and challenges models' ability to comprehend text–image interplay. The Multimodal Emotion Recognition task uses MER2023 (Lian et al., 2023) to evaluate models' generalization in broader multimodal contexts. Opinion Sentiment Analysis uses CMU-MOSI (Zadeh et al., 2016), a corpus of monologue videos that targets sentiment polarity from speech and facial expressions. Emotion Intensity Analysis extends this by using the larger CMU-MOSEI dataset (Zadeh et al., 2018), which requires models not only to identify emotion categories but also to quantify their intensity. Song and Speech Emotion Recognition employ RAVDESS (Livingstone & Russo, 2018), comprising speech and song clips by professional actors with matched lexical content and emotions presented at varying intensities. Finally, to assess domain-specific performance, Stock Comment Emotion Analysis uses FMSA-SC (Song et al., 2024a) to analyze emotions in financial-domain comments.

**Level 2: Emotion Understanding and Analysis.** This level constitutes an advanced layer of emotion intelligence, which goes beyond basic classification, requiring models to analyze emotions within complex contexts. This capability entails not only identifying emotions but also modeling their complexity and interpreting their function and intent in specific situations. Accordingly, models must exhibit robust contextual awareness and relational reasoning. To evaluate these abilities, this layer incorporates several challenging tasks. For internet culture, Detection of Persuasion Techniques in Memes employs the SemEval-2021 Task 6 dataset (Dimitrov et al., 2021) to identify persuasive intent in memes. Emotion-Based Intent Analysis uses the MC-EIU dataset (Liu et al., 2024) to examine links between emotional expression and users' underlying intent. Humor Understanding employs UR-FUNNY (Hasan et al., 2019), a corpus of TED-talk clips that requires integrating linguistic, visual, and acoustic cues to determine whether content is humorous. Implicit Attribute Value Extraction uses ImplicitAVE (Zou et al., 2024), in which attribute values are not stated explicitly. Multimodal Stance Detection leverages the MWTWT datasets (Liang et al., 2024). The Multimodal Quintuple Extraction task, based on the PanoSent dataset (Luo et al., 2024a), aims to parse five core elements of sentiment—holder, target, aspect, opinion and sentiment. Multimodal Aspect-Based Sentiment Analysis uses Twitter2015/2017 (Yu & Jiang, 2019) to evaluate models' ability to identify fine-grained sentiment toward specific entities or aspects in text and images. Lastly, to approximate real-world social interaction, Multiparty Dialogue Emotion Recognition uses MELD (Poria et al., 2018), a corpus of multiparty conversational clips from Friends, and requires tracking the emotional dynamics of each character in multi-person interactions.

**Level 3: Emotion Cognition and Reasoning.** This level constitutes the highest tier of emotion intelligence, which requires models not only to perceive and understand emotions but also to reason about their causal relationships, temporal dynamics, and underlying cognitive processes. This level approximates a computational account of human emotional cognition, encompassing tasks such as explaining emotion causes, predicting consequent behaviors, and interpreting complex expressions. Evaluation at this layer focuses on models' cognitive and reasoning abilities. Emotion Elicitation Reasoning uses FilmStim (Schaefer et al., 2010) to assess whether models can infer emotions likely to be elicited in audiences from film-clip content. Emotion Interpretation leverages EIBench (Lin et al., 2025b), requiring models to explain the deeper meaning and motivation behind emotional expressions. Laughter Reasoning uses the SMILE dataset (Hyun et al., 2023), which requires models to explain the specific reason for a person's laughter in a video, demanding a nuanced understanding of social context. Multimodal Emotion Cause Pair Extraction employs the ECF dataset (Wang et al., 2022), focusing on precisely identifying the event or cause that leads to a particular emotion from multimodal signals. Sarcasm detection uses the MUSTARD dataset (Castro et al., 2019), which contains sarcastic dialogue clips from TV shows. Models must integrate contextual, prosodic, and facial cues to identify the incongruity between an utterance's literal meaning and its intended meaning. Finally, Sentiment Flip Analysis also uses the PanoSent (Luo et al., 2024a), requiring models to detect shifts in emotional state during conversation and to identify the key causes of such flips.

## C.1 DATASET DETAILS

We benchmark a total of 22 publicly available multimodal affective computing datasets, which together constitute 24 distinct tasks. The following section details each dataset included in our benchmark, and representative samples are provided in Appendix G.

- **CH-SIMS**: CH-SIMS is constructed from 60 Chinese movies, TV series, and variety shows involving 474 unique speakers. The dataset comprises 2,281 curated video clips with an average duration of 3.67 seconds. Its key characteristic is the provision of both unimodal and multimodal annotations across text, audio, and vision under in-the-wild conditions. Labels are assigned to five sentiment categories: negative, weakly negative, neutral, weakly positive, and positive.

- **CH-SIMSv2**: CH-SIMS v2.0 extends CH-SIMS with Mandarin video data from films, TV, talk shows, interviews, vlogs, and other sources. It offers 4,402 supervised segments (2,281 relabeled and 2,121 new) alongside 10,161 unsupervised clips. Designed for text–acoustic–visual analysis, it employs 720p+ sources, active speaker detection, and strict modality separation. Labels comprise unimodal and multimodal scores mapped to five sentiment categories ranging from negative to positive.

- **EmoSet**: EmoSet is curated using 810 emotion-related keywords from social media and artistic image platforms. It comprises 3.3 million images, of which 118,102 are carefully annotated by humans. Distinguished by attribute diversity, the dataset records brightness, colorfulness, scene type, object class, facial expression, and human action. Labels follow Mikels' model, encompassing eight categories: amusement, awe, contentment, excitement, anger, disgust, fear, and sadness.

- **Memotion**: Memotion is compiled from Reddit and Google Images through automated crawling and enriched with OCR text via Google Vision. The dataset consists of 10,000 Hinglish memes, divided into 8,500 for training, 1,500 for validation, and 1,500 for testing, with annotations verified by bilingual raters. It distinctively integrates multimodal content with code-mixed language, providing sentiment labels (positive, neutral, negative), four emotion types (humorous, sarcastic, offensive, motivational), and graded intensity levels.

- **MER2023**: MER 2023 extends CHEAVD (Li et al., 2017) by automatically collecting expression-centric video clips and releasing rigorously curated splits for community benchmarking. The corpus comprises 3,373 Train & Val samples and three test partitions—MER-MULTI, MER-NOISE, and MER-SEMI—amounting to about 68 hours of audiovisual data. Emphasizing robustness, it provides three tracks (multi-label learning, modality-noise robustness, and semi-supervised learning) and supplies annotations for six discrete emotions (neutral, anger, happiness, sadness, worry, surprise) together with a continuous valence dimension.

- **CMU-MOSI**: MOSI is a multimodal opinion-level corpus for sentiment intensity and subjectivity in online vlogs. It comprises 93 videos (89 speakers), 3,702 segments, and 2,199 opinion clips labeled on a -3...+3 scale by five AMT raters. Releases include word- and phoneme-aligned transcripts, millisecond acoustic features, frame-level visual cues, and gesture tags, with fine-grained subjectivity boundaries and high inter-annotator agreement. Baselines demonstrate that multimodal fusion—especially a word–gesture "multimodal dictionary"—outperforms text-only models.

- **CMU-MOSEI**: CMU-MOSEI is one of the largest multimodal sentiment analysis corpora, derived from 3,228 YouTube videos featuring about 1,000 speakers across 250 topics. It offers 23,453 sentence-level segments with synchronized text, audio, and visual modalities. Distinguished by its scale and fine-grained alignment, the dataset facilitates cross-modal learning. Labels cover a 7-point sentiment scale from -3 (strongly negative) to +3 (strongly positive), supporting both polarity and intensity prediction.

- **RAVDESS**: RAVDESS is a validated multimodal corpus of speech and song by 24 professional actors in North American English. Speech covers neutral/calm, happy, sad, angry, fearful, surprise, and disgust, while song includes neutral/calm, happy, sad, angry, and fearful, each at two intensity levels. The 7,356 recordings are available in audio-visual, audio-only, and video-only formats. Each clip is rated 10 times by 247 raters, demonstrating strong validity and test–retest reliability.

- **FMSA-SC**: FMSA-SC consists of 1,247 stock comment videos. Its novelty lies in offering fine-grained sentiment annotations, aligning textual phrases with corresponding visual and acoustic cues. Labels span five sentiment levels from strong negative to strong positive, establishing the first multimodal benchmark for financial sentiment analysis.

- **SemEval-2021 Task 6**: SemEval-2021 Task 6 introduces a multimodal benchmark for detecting persuasion techniques in memes collected from 26 public Facebook groups. The dataset comprises 950 memes, divided into 687 training, 63 development, and 200 testing samples. Its novelty lies in addressing propaganda in a multimodal context, offering three subtasks on text, spans, and complete memes. Annotations cover 22 persuasion techniques, encompassing both textual and visual strategies.

- **MC-EIU**: The MC-EIU dataset offers a large-scale open-source resource for multimodal emotion and intent understanding in conversation. It includes 4,970 clips with 56,012 utterances from English and Mandarin TV series, totaling 53 hours of dialogue. Distinguished by its bilingual coverage and tri-modal design (text, audio, and video), it provides seven emotion and nine intent categories, establishing the first comprehensive benchmark for joint affective analysis.

- **UR-FUNNY**: UR-FUNNY is derived from TED talks, using transcripts and laughter markers to label punchlines and contexts. The corpus covers 1,866 videos with 1,741 speakers and 417 topics, containing 8,257 humorous and 8,257 non-humorous instances. It features tri-modal alignment (text, audio, vision), speaker-independent splits, and word-level synchronization to support robust multimodal modeling, with balanced negatives sampled from the same videos. Tags: humor detection, multimodal language, TED, punchline–context modeling, laughter cues, speech–vision–text fusion.

- **ImplicitAVE**: ImplicitAVE is the first publicly available multimodal dataset for implicit attribute value extraction in e-commerce. It includes 68,604 training and 1,610 human-verified test instances across five domains and 25 attributes. Its novelty lies in curating implicit values absent from text but inferable from images or context, supplemented with product photos and rigorous human re-annotation. Labels span 158 attribute values across clothing, footwear, jewelry, food, and home products.

- **Twitter2015/2017**: The Twitter2015 and Twitter2017 datasets are benchmark corpora for target-oriented multimodal sentiment classification. Together they comprise over 5,000 tweets paired with images, annotated for sentiment polarity toward specific opinion targets. Their main contribution is enabling fine-grained alignment between textual and visual content to model sentiment at the target level. Labels span three categories—positive, negative, and neutral—supporting multimodal sentiment research.

- **PanoSent**: The PanoSent dataset establishes a large-scale benchmark for Multimodal Conversational Aspect-based Sentiment Analysis. It comprises 10,000 dialogues and over 47,000 sextuples across English, Chinese, and Spanish, integrating text, image, audio, and video. Its novelty lies in panoptic sentiment sextuple extraction and dynamic sentiment flipping analysis, capturing holders, targets, aspects, opinions, sentiments, and rationales. Annotated through both human experts and GPT-4 synthesis, it supports multi-scenario, and implicit sentiment reasoning, with labels covering fine-grained and causal dynamics.

- **MWTWT**: The MWTWT dataset originates from the Multi-modal Stance Detection project. It extends the textual Will-They-Won't-They dataset (Conforti et al., 2020) into a multimodal form by incorporating both tweets and images. The dataset comprises 1,747 annotated examples focused on corporate merger debates, where each instance is labeled as Support, Refute, Comment, or Unrelated. Its uniqueness lies in capturing stance expression across text–image pairs, enabling research on multimodal opinion dynamics. Labels highlight nuanced stance categories relevant to corporate decision-making.

- **MELD**: The MELD dataset extends the EmotionLines corpus(Chen et al., 2018) into a multimodal benchmark for emotion recognition in conversations. It contains over 13,000 utterances from 1,433 dialogues in the TV series Friends, each annotated with emotion and sentiment labels across audio, visual, and textual modalities. By emphasizing multi-party interactions, MELD captures complex phenomena such as emotion shifts and inter-speaker dependencies, providing a challenging resource for multimodal conversational emotion recognition. Tags: multimodal, emotion, conversation, multi-party.

- **FilmStim**: The FilmStim dataset is developed to provide a validated collection of emotion-eliciting film excerpts for experimental research. It comprises 70 clips selected through expert surveys and validated on 364 participants, offering a rich tool for controlled emotion induction. Its distinctive feature lies in covering both basic emotions and mixed feelings, with validated criteria including arousal, valence, and emotional discreteness. Labels span anger, fear, sadness, disgust, amusement, tenderness, and neutral states, making it a comprehensive benchmark for affective studies.

- **EIBench**: The EIBench dataset is constructed from CAER-S (Lee et al., 2019) and EmoSet to advance the task of Emotion Interpretation, which asks why an emotion arises rather than merely identifying which emotion is present. It contains 1,615 basic samples and 50 complex cases, requiring models to generate causal explanations across explicit and implicit triggers. Its key contribution is the Coarse-to-Fine Self-Ask (CFSA) annotation pipeline, which combines Vision-Language Models with human refinement to capture nuanced, context-dependent emotional reasoning. Labels span four primary emotions—angry, sad, happy, and excited—with complex subsets featuring overlapping emotional states.

- **SMILE**: The SMILE dataset is curated from TED talks and sitcoms to explore the task of Video Laugh Reasoning. It comprises 887 clips with 4,434 annotated segments, each paired with textual explanations of why laughter occurs. Its unique focus lies in audience laughter, reducing subjectivity and highlighting multimodal cues across visual, acoustic, and semantic channels. Labels provide free-form explanations rather than fixed classes, enabling deeper analysis of social intelligence.

- **ECF**: The ECF dataset is constructed from the sitcom Friends to support the task of Multimodal Emotion-Cause Pair Extraction in Conversations. It contains 1,344 conversations with 13,509 utterances and 9,272 annotated emotion–cause pairs. Its distinguishing feature lies in integrating text, audio, and video modalities to capture diverse causal triggers, categorized as events, opinions, emotional influence, and greetings. Emotion labels follow Ekman's six basic categories: anger, disgust, fear, joy, sadness, and surprise.

- **MUStARD**: The MUStARD dataset is constructed from TV shows such as Friends, The Big Bang Theory, The Golden Girls, and Sarcasmaholics Anonymous to advance multimodal sarcasm detection. It comprises 690 balanced video clips evenly divided between sarcastic and non-sarcastic utterances, each paired with transcripts and conversational context. Its distinctive contribution lies in integrating text, audio, and visual cues with dialogue history, enabling nuanced analysis of incongruity across modalities. Labels are binary: sarcastic versus non-sarcastic.

## D    IMPLEMENTATION DETAILS

### D.1    REWARD COMPONENT DETAILS

**Weight Assignment Rationale.** The weights for the reward function are set based on a principled hierarchy reflecting each component's importance. We set $\mu_2 = 1.0$ (content) and $\mu_4 = 1.0$ (consistency) to assign the highest priority to the foundational requirements of correctness and logical-factual grounding. A moderate weight of $\mu_1 = 0.4$ (structure) ensures compliance with the reasoning format without overriding correctness. Lastly, a minimal weight of $\mu_3 = 0.1$ (process) serves as a gentle stylistic nudge, guiding the model towards domain-specific language while mitigating the risk of superficial keyword stuffing. The four components of our comprehensive reward function are detailed as follows:

- **Structure Reward ($R_{\text{structure}}$):** This reward fosters the generalization of structured reasoning by enforcing the unique cognitive framework required for each training task. The reward system is task-aware: it first identifies the task from the input prompt to select the corresponding reasoning template. The reward $R_{\text{structure}}$ is then calculated as the proportion of required step headers that are correctly present and sequenced within the reasoning chain $\tau$.

- **Content Reward ($R_{\text{content}}$):** This reward evaluates the correctness of the final output $o$ by comparing it against the ground-truth label using the standard evaluation metric appropriate

for each task's specific format. This ensures the model's reasoning ultimately leads to a factually accurate conclusion.

- **Process Reward** ($R_{\text{process}}$): This reward promotes the articulation of reasoning using ToM-specific language. We curate a lexicon of ToM-related keywords (e.g., "belief," "intention," "desire"). $R_{\text{process}}$ is calculated as the normalized count of unique keywords from this lexicon found within $\tau$. This encourages the model not just to follow a structural template, but to fill it with rich language reflecting genuine cognitive reasoning.

- **Consistency Reward** ($R_{\text{consistency}}$): To penalize logical fallacies, this reward assesses the consistency of the reasoning chain $\tau$. We employ a large language model to detect two types of inconsistencies: (1) *Internal Contradictions*, where the chain contradicts itself, and (2) *External Contradictions*, where the chain describes a fact inconsistent with the input multimodal context. $R_{\text{consistency}}$ is a penalty-based reward, yielding a high value (1.0) for consistent chains and a significantly lower value (0.1) if any contradictions are found.

The computational formulas for the four reward components are defined as follows:

**Structure Reward** ($R_{\text{structure}}$).   This reward calculates the proportion of required structural elements that are correctly present and sequenced. Let $\mathcal{S}_{\text{req}}$ be the ordered sequence comprising the mandatory XML delimiters and the task-specific step headers $\{h_k\}$ derived from the prompt:

$$\mathcal{S}_{\text{req}} = [\texttt{<think>}, h_1, \ldots, h_K, \texttt{</think>}, \texttt{<answer>}, \texttt{</answer>}] \tag{4}$$

Let $\text{idx}(s, y)$ denote the index of token $s$ in $y$. We define a validity indicator $v_i \in \{0, 1\}$ for the $i$-th token in $\mathcal{S}_{\text{req}}$:

$$v_i = \mathbb{I}\left[\text{idx}(s_i, y) \neq \infty \quad \wedge \quad \text{idx}(s_i, y) > \max(\{\text{idx}(s_j, y) \mid j < i, v_j = 1\} \cup \{-1\})\right] \tag{5}$$

This recursive condition strictly enforces the topological order. Let $N = |\mathcal{S}_{\text{req}}|$ be the total number of required elements. The reward is the proportion:

$$R_{\text{structure}}(y) = \frac{1}{N} \sum_{i=1}^{N} v_i \tag{6}$$

**Content Reward** ($R_{\text{content}}$).   This evaluates the correctness using the standard metric $\mathcal{M}_{\text{task}}$ (e.g., Accuracy, F1) comparing the extracted answer $o$ to the ground truth $o^*$:

$$R_{\text{content}}(y) = \mathcal{M}_{\text{task}}(o, o^*) \tag{7}$$

**Process Reward** ($R_{\text{process}}$).   Consistent with the description of a normalized count, let $\mathcal{V}_{\text{ToM}}$ be the ToM lexicon and $S_\tau$ be the set of unique tokens in $\tau$. We use a normalization factor $\eta$:

$$R_{\text{process}}(y) = \min\left(1.0, \frac{|S_\tau \cap \mathcal{V}_{\text{ToM}}|}{\eta}\right) \tag{8}$$

**Consistency Reward** ($R_{\text{consistency}}$).   To rigorously enforce logical soundness, we employ an LLM Judge to detect inconsistencies, the Judge evaluates two logical predicates:

- $J_{\text{int}}(\tau)$: Returns *True* if the reasoning chain is free of internal contradictions.
- $J_{\text{ext}}(\tau, \text{Input})$: Returns *True* if the reasoning chain is consistent with the inputs $(T, A, V)$.

The final reward applies a penalty if either condition fails (i.e., if any contradiction is found):

$$R_{\text{consistency}}(y) = \begin{cases} 1.0 & \text{if } J_{\text{int}}(\tau) \wedge J_{\text{ext}}(\tau, \text{Input}) \\ 0.1 & \text{otherwise} \end{cases} \tag{9}$$

D.2 HYPERPARAMETER SENSITIVITY ANALYSIS

To empirically validate the optimality of our reward weight configuration ($\mu_1 = 0.4, \mu_2 = 1.0, \mu_3 = 0.1, \mu_4 = 1.0$), we conducted a fine-grained grid search. We report the Average Score (mean of L1, L2, and L3 tasks) to quantify the optimization objective.

**Sensitivity to Process Reward ($\mu_{\mathbf{process}}$).** We fixed $\mu_{\text{struct}} = 0.4$ and varied $\mu_{\text{process}}$. The results are shown in Table 7.

Table 7: Ablation study on Process Reward Weight.

| $\mu_{\text{process}}$ | 0.0 | 0.1 | 0.3 | 0.5 |
|---|---|---|---|---|
| Score | 63.99 | **64.70** | 63.65 | 62.31 |

The configuration $\mu = 0.0$ serves as the baseline without stylistic constraints. Introducing a minimal weight ($\mu = 0.1$) yields the optimal performance. However, increasing the weight beyond this point ($\mu \geq 0.3$) leads to a degradation in reasoning quality, as the model prioritizes keyword frequency over logical correctness.

**Sensitivity to Structure Reward ($\mu_{\mathbf{struct}}$).** We fixed $\mu_{\text{process}} = 0.1$ and varied $\mu_{\text{struct}}$. Table 8 illustrates the impact of structural constraints.

Table 8: Ablation on Structure Reward Weight.

| $\mu_{\text{struct}}$ | 0.1 | 0.4 | 0.7 | 1.0 |
|---|---|---|---|---|
| Score | 61.25 | **64.70** | 64.10 | 63.80 |

At $\mu = 0.1$, the penalty is insufficient to enforce the XML schema, leading to Format Collapse and parsing failures. Conversely, high weights ($\mu \geq 0.7$) cause Structural Rigidity, where the model strictly adheres to templates at the cost of the reasoning flexibility required for complex multimodal inputs, resulting in diminished accuracy.

D.3 EXPERIMENT SETTINGS

During inference, we allow 16–64 frames with a resolution of up to $256 \times 28 \times 28$ pixels per frame. Training hyperparameters for both SFT and GRPO stages, including learning rate, scheduler, batch size, and rollout settings, are summarized in Table 9. The model uses a context window of 8,192 tokens and a maximum generation length of 4,096. The closed-source models included in our evaluation are accessed through their official APIs.

We evaluate a total of 17 MLLMs, comprising 4 closed-source and 13 open-source models. A brief introduction to each model is provided below:

- **VideoLLaMA3** (Zhang et al., 2025a) is the third-generation VideoLLaMA series designed for both image and video understanding. It introduces flexible resolution tokenization and efficient frame pruning to reduce redundancy while preserving temporal context. With a progressive training pipeline, VideoLLaMA3 achieves strong performance on diverse video reasoning and description benchmarks, particularly in long-horizon and fine-grained temporal comprehension.
- **LLaVA-One-Vision** (Li et al., 2024a) is a unified vision–language model built to handle images, documents, and charts under one interface. Its transfer framework distills knowledge from multiple pretrained encoders into a single instruction-tuned backbone. LLaVA-One-Vision enables broad task coverage and practical deployment in real-world multimodal applications, ranging from general VQA to structured document analysis.
- **LLaVA-NeXT-Video** (Li et al., 2024b) extends the LLaVA-NeXT series to video understanding. It employs interleaved frame encoding with preference-optimized alignment to

Table 9: Key hyperparameters for the SFT and GRPO training stages.

| Hyperparameter | SFT Stage | GRPO Stage |
|---|---|---|
| Base Model | Qwen2.5-Omni-7B | SFT-tuned Model |
| Learning Rate | $1.0 \times 10^{-5}$ | $1.0 \times 10^{-6}$ |
| LR Scheduler | Cosine | Constant |
| Warmup Ratio | 0.1 | N/A |
| Epochs | 2 | 1 |
| Batch Size | 16 | 8 |
| Precision | bfloat16 | bfloat16 |
| Rollout Samples ($N$) | N/A | 8 |
| KL Coefficient | N/A | 0.001 |

enhance temporal reasoning and dialogue quality. LLaVA-NeXT-Video proves effective for video QA and conversational analysis in long-horizon scenarios, delivering more coherent and context-aware responses.

- **Qwen2.5-VL** (Bai et al., 2025) is a vision–language model family developed by Alibaba. It combines dynamic-resolution processing, fine-grained localization, and progressive alignment to support documents, diagrams, and long videos. Qwen2.5-VL delivers reliable perception and reasoning across diverse multimodal benchmarks, excelling in tasks that require detailed and structured visual understanding.

- **InternVL3** (Zhu et al., 2025) is the third-generation InternVL family, integrating stronger vision encoders with Qwen2.5 backbones. It introduces efficient token reduction and improved preference optimization for reasoning-heavy tasks. InternVL3 achieves notable improvements in OCR, document analysis, and complex visual understanding, offering a more balanced trade-off between efficiency and accuracy.

- **MiniCPM-V** (Yao et al., 2024) is a lightweight multimodal LLM optimized for on-device use, including phones and edge platforms. With efficient visual encoding, multilingual tuning, and system-level optimizations, it supports privacy-preserving and energy-efficient interaction. MiniCPM-V enables practical multimodal deployment in resource-constrained environments, making advanced perception and reasoning accessible on everyday devices.

- **R1-Omni** (Zhao et al., 2025a) is an omni-modal model focused on emotion reasoning. Building on HumanOmni, it integrates reinforcement learning with verifiable rewards to enhance interpretability. R1-Omni generates step-by-step explanations that clarify how visual and acoustic cues shape predictions, leading to improved generalization in challenging emotional tasks.

- **HumanOmni** (Zhao et al., 2025b) is a human-centric omni-multimodal model trained for emotion and interaction understanding. It employs dedicated perception branches for faces, bodies, and interactions, fused with audio signals. HumanOmni excels in human-related applications such as emotion recognition and social behavior analysis, enabling more fine-grained comprehension of real-world scenarios.

- **Qwen2.5-Omni** (Xu et al., 2025) is a flagship omnimodal model that unifies text, images, audio, and video while generating both text and speech. Its Thinker–Talker architecture and efficient streaming design support real-time interaction. Qwen2.5-Omni enables speech-in/speech-out multimodal assistants for continuous audiovisual tasks, combining responsiveness with versatile cross-modal reasoning.

- **Emotion-LLaMA** (Cheng et al., 2024) is a multimodal model tailored for affective computing. It fuses audio, visual, and text encoders through a two-stage training pipeline for recognition and explanation. Emotion-LLaMA advances emotion-aware understanding across diverse modalities, supporting both accurate recognition and interpretable rationale generation.

- **AffectGPT** (Lian et al., 2025) is a multimodal emotion model that introduces MER-Caption, the largest fine-grained emotion dataset gathered via a novel model-based crowd-sourcing strategy. It also embeds pre-fusion operations for enhanced cross-modal alignment and proposes MER-UniBench, a unified evaluation benchmark tailored to natural

language emotion understanding. AffectGPT optimizes emotion-aware reasoning and descriptive understanding in multimodal LLMs.

- **GPT-4o** (Hurst et al., 2024) is one of OpenAI's latest multimodal large language models, offering APIs that can seamlessly handle text, vision, and audio. It shows strong performance across numerous benchmarks, with notable progress in perception, comprehension, and multimodal reasoning. Built on a unified design that supports smooth cross-modal integration, GPT-4o is efficient and versatile, making it well-suited for practical multimodal applications.

- **GPT-4.1** (OpenAI, 2025) is a recently released multimodal model that emphasizes both cost-effectiveness and reliability. It enhances programming capabilities and instruction following, while also introducing an extended context window of up to one million tokens. With this improvement, GPT-4.1 is able to deliver more robust long-context reasoning and significantly improve task efficiency.

- **Gemini-2.5-Flash** (Comanici et al., 2025) is a multimodal reasoning model designed to balance speed, performance, and resource usage. It introduces selective reasoning modes, enabling users to trade off accuracy and efficiency depending on their needs. With its fine-grained control of reasoning steps, Gemini-2.5-Flash achieves competitive results across a broad range of multimodal understanding benchmarks.

- **Gemini-2.5-Pro** (Comanici et al., 2025) is the flagship model in the Gemini family, advancing multimodal understanding with stronger perception and reasoning. It supports longer contexts and delivers improved cross-modal alignment, while excelling in domains such as programming, mathematics, and scientific analysis. Equipped with more capable reasoning abilities, Gemini-2.5-Pro is optimized for demanding, knowledge-intensive tasks.

### D.4 TRAINING DATA GENERATION

Our training data generation methodology follows a multi-stage pipeline designed to produce high-fidelity reasoning chains. The pipeline integrates the generative capacity of advanced MLLMs with automated filtering and human-in-the-loop verification, ensuring both efficiency and reliability. It comprises four key stages, as outlined below.

**Step 1: Reasoning Chain Generation.** For each sample, we generate initial reasoning pathways by providing GPT-5 with the video input and our tailored prompt, augmented by an auxiliary report from Qwen2-Audio that identifies and supplies additional information present in the soundtrack. This module extracts salient audio cues—such as crying, laughter, changes in tone, speech rate, pauses, emphasis and stress, or voice trembling—which are integrated with the visual and textual context according to the task. GPT-5 then produces three independent candidate reasoning chains from these multimodal inputs.

**Step 2: Label-Aligned Filtering & Correction.** The generated reasoning chains are automatically evaluated by comparing their predicted labels against the ground-truth answers. Based on this comparison, each sample is categorized into one of three groups. If at least one of the three candidate chains produces a final output that exactly matches the ground truth, it is preserved as correct and advanced to the next stage. If all three chains fail, the sample is classified as completely incorrect and flagged for intensive correction. In multi-label tasks, chains that correctly predict part of the labels but miss others are deemed partially incorrect; for these cases, the full original reasoning path together with the gold-standard labels are provided to the correction model, which is instructed to revise only the erroneous parts while keeping the valid portions intact.

**Step 3: Self-Reflection Normalization.** The correction process is carried out using Gemini-2.5-pro, which generates either a fully new reasoning chain for completely incorrect samples or targeted revisions for partially incorrect ones. Once all corrected and initially correct chains are consolidated, they undergo a final self-reflection step with GPT-4o. In this phase, the model standardizes formatting, ensures logical clarity, and refines the articulation of ToM concepts, resulting in coherent and high-quality reasoning outputs.

**Step 4: Human Verification.** As the final stage, the complete set of refined reasoning chains undergoes human-in-the-loop verification. Two computer science PhD students manually review each chain with the primary goal of cross-validating the reasoning against the source multimodal

information. If any factual inaccuracies, logical inconsistencies, or misinterpretations of the visual context are detected, the annotators intervene to edit and finalize the chain, thereby ensuring its correctness and reliability.

Our data generation pipeline yields a two-stage training corpus tailored for reasoning alignment. First, in the SFT stage, we collect approximately 10,000 high-quality prompt–response pairs to bootstrap the model's output format, reasoning style, and baseline behavior. Then, in the GRPO stage, we select another 10,000 prompt instances emphasizing tasks with complex reasoning structure and high diversity; these prompts serve as seeds for multiple rollouts and pairwise preference comparisons to train a policy aligned to human judgments.

### D.5 EVALUATION SPECIFICATIONS

Our evaluation framework is designed to rigorously assess model performance across a spectrum of emotion-related tasks. We employ a set of four primary metrics: ACC (Accuracy), WAF (Weighted Average F1-score), MF (Micro F1 score), and EMF (Exact Match F1). The evaluation is stratified into three hierarchical levels, with the complexity of tasks and the sophistication of metrics increasing at each level.

**Level 1: Emotion Perception and Recognition.** This foundational level focuses on the direct perception and classification of emotional and sentimental states from various data modalities. It encompasses 10 tasks: FESD, ISA, MESA, MER, MSA, OSA, SIA, SOER, SPER, and SCEA. These tasks are measured using ACC and WAF.

- **ACC** offers a direct measure of overall correctness by calculating the ratio of correct predictions to the total number of samples.

$$\text{ACC} = \frac{\text{Number of Correct Predictions}}{\text{Total Number of Samples}} \quad (10)$$

- **WAF** addresses class imbalance by computing the F1 score for each class and averaging them, weighted by the number of true instances per class ($|S_c|$). This yields a more balanced assessment, especially when certain emotion labels are underrepresented. For a set of classes $C$ and a total sample size of $|S|$, it is defined as:

$$\text{WAF} = \sum_{c \in C} \frac{|S_c|}{|S|} \times F1_c \quad (11)$$

where the F1 score for an individual class, $F1_c$, is the harmonic mean of its precision and recall:

$$F1_c = 2 \times \frac{\text{Precision}_c \times \text{Recall}_c}{\text{Precision}_c + \text{Recall}_c} \quad (12)$$

**Level 2: Emotion Understanding and Analysis.** This intermediate level requires a deeper analytical capability, moving from simple recognition to understanding context and implicit attributes. It includes eight tasks: DPTM, EBIA, HU, IAVE, MABSA, MQE, MSD, MDER. For the classification tasks at this level, we continue to utilize ACC and WAF. Additionally, to provide a holistic performance view on more granular tasks, we also use the MF score.

- **MF** assesses performance by aggregating the counts of true positives (TP), false positives (FP), and false negatives (FN) across all classes before computing the final score. This makes it equivalent to overall accuracy in single-label classification but provides a robust metric for more complex scenarios.

$$\text{Precision}_\mu = \frac{\sum_{c \in C} \text{TP}_c}{\sum_{c \in C} (\text{TP}_c + \text{FP}_c)} \quad (13)$$

$$\text{Recall}_\mu = \frac{\sum_{c \in C} \text{TP}_c}{\sum_{c \in C} (\text{TP}_c + \text{FN}_c)} \quad (14)$$

$$\text{MF} = 2 \times \frac{\text{Precision}_\mu \times \text{Recall}_\mu}{\text{Precision}_\mu + \text{Recall}_\mu} \quad (15)$$

**Level 3: Emotion Cognition and Reasoning.** This highest level probes the model's ability to perform complex reasoning and generate human-like explanations. It comprises six tasks: EER, EI, LR, MECPE, SD, and SFA. The evaluation methodology at this level is diversified to match the task requirements. For classification-oriented tasks like SD, we continue to employ ACC and WAF. For tasks that demand the generation of free-form text, we also use two specialized evaluation strategies: EMF for answers with a high degree of expected lexical overlap, and a sophisticated LLM-based evaluation for open-ended, creative responses.

- **EMF** is designed for generative tasks where the desired output is a specific, factual explanation, such as in LR. It quantifies the word-level overlap between the predicted and ground-truth texts after normalization. The texts are treated as a bag of words, and the F1 score is computed based on the common words. Let $\text{Words}_{\text{pred}}$ and $\text{Words}_{\text{gt}}$ be the set of words in the prediction and ground truth, respectively.

$$\text{Precision}_{\text{word}} = \frac{|\text{Words}_{\text{pred}} \cap \text{Words}_{\text{gt}}|}{|\text{Words}_{\text{pred}}|} \tag{16}$$

$$\text{Recall}_{\text{word}} = \frac{|\text{Words}_{\text{pred}} \cap \text{Words}_{\text{gt}}|}{|\text{Words}_{\text{gt}}|} \tag{17}$$

$$\text{EMF} = 2 \times \frac{\text{Precision}_{\text{word}} \times \text{Recall}_{\text{word}}}{\text{Precision}_{\text{word}} + \text{Recall}_{\text{word}}} \tag{18}$$

- **LLM-based Semantic Evaluation** is employed for open-ended tasks like EI, where multiple, distinct answers can be valid and word-level overlap metrics like EMF are inadequate. In this paradigm, we leverage GPT-4.1 as a semantic judge. The LLM is prompted to compare the meaning of the generated response against the ground-truth answer(s), assessing its semantic relevance, plausibility, and correctness. This approach transcends word-level matching to capture the true quality of nuanced and diverse generative outputs.

In practice, some models fail to strictly follow the required output format due to differences in instruction-following ability. In such cases, we also employ GPT-4.1 to normalize and extract the intended answers, ensuring consistent and fair evaluation across all models.

# E    EXTENDED EXPERIMENTS RESULTS

In this section, we extend our evaluation across all three hierarchical levels and introduce metrics beyond accuracy for a more in-depth analysis. We report results in two parts: Tables 10-12 establish baseline performance of vanilla models, while Tables 13-15 show the effects of integrating ToM prompting. Complementing these tables, Figure 6 provides per-task radar visualizations that directly contrast each representative model *with* vs. *without* ToM prompting across all benchmark tasks, revealing heterogeneous impacts—substantial improvements on some tasks and occasional regressions on others. In addition, we provide task-level comparisons on labeled tasks, where fine-grained F1 scores across $n$ emotion categories are reported for all MLLMs; visual summaries appear in Figures 8–21. We also include confusion matrices for Gemini-2.5-Pro (Figures 22 and 23).

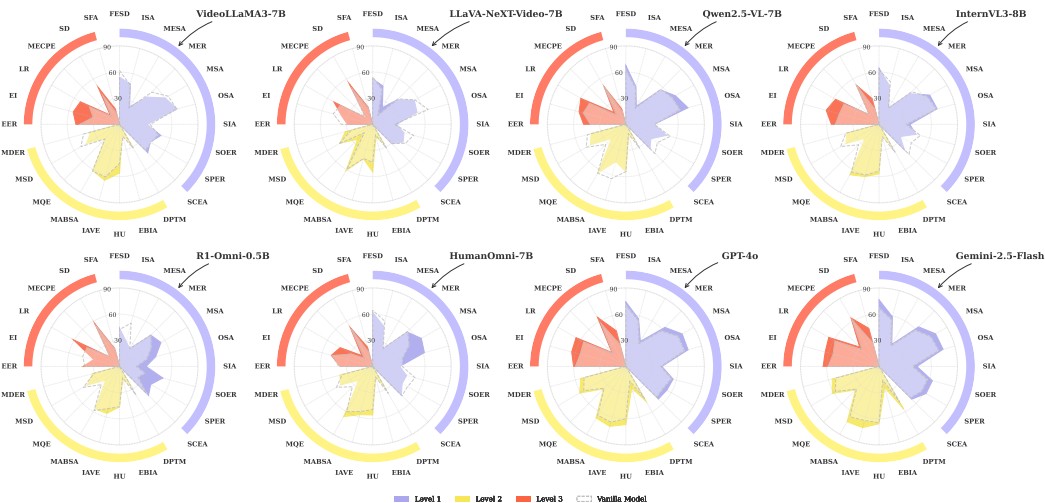

Figure 6: **Per-task radar performance for representative models.** Each polar chart groups tasks by benchmark level and juxtaposes vanilla model vs. ToM-prompted results.

## E.1    TASK-LEVEL PERFORMANCE CHARACTERISTICS OF CURRENT MLLMS.

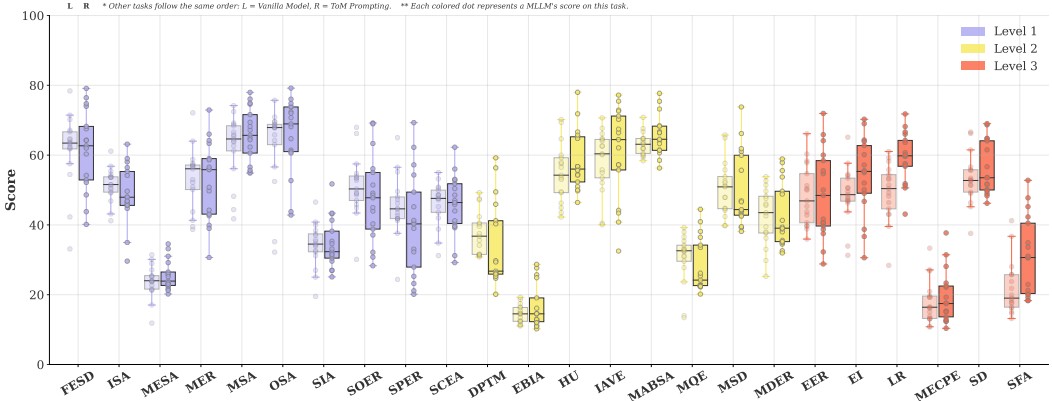

Figure 7: **Score distributions by task and level.** For each task, the left box corresponds to the vanilla model and the right box to the ToM Prompting.

Figure 7 summarizes the performance of MLLMs across tasks spanning three hierarchical levels. Current models perform relatively well on explicit emotion recognition: for example, Gemini-2.5-Pro scores 78.39 on FESD, and GPT-4.1 reaches 71.46, demonstrating that large-scale MLLMs can classify direct emotional cues with reasonable accuracy. However, performance declines sharply on tasks requiring implicit inference or complex contextual reasoning. In EBIA, even Gemini-2.5-Pro attains only 19.25, underscoring the difficulty of intent recognition. Likewise, in structured extraction tasks such as MQE and MECPE, most models achieved MF scores below 30. These results reveal persistent weaknesses in handling implicit cues, causal reasoning, structured extraction, and higher-level pragmatic understanding such as dialogue dynamics, sarcasm, and humor.

## E.2 VERIFICATION OF LLM-BASED EVALUATION RELIABILITY

To validate the reliability of GPT-4.1 in our evaluation pipeline, we consider two representative scenarios. First, for free-form generation tasks in Level 3, we randomly sample one-third of the dataset and compare GPT-4.1's judgments with human annotations. Agreement is measured using accuracy and Cohen's Kappa. GPT-4.1 achieves 98.5% agreement with a Cohen's Kappa of 0.98, demonstrating near-human reliability. Second, for classification-style tasks where models often fail to follow the required output format, we employ GPT-4.1 to normalize predictions and extract the intended labels. On a stratified sample of 2,000 such cases, GPT-4.1's extracted labels match human interpretations with 96.5% agreement and a Cohen's Kappa of 0.96. These results confirm that GPT-4.1 provides a consistent and trustworthy mechanism both for semantic judgment and for standardizing model outputs, ensuring fair and reliable evaluation across diverse task types.

Table 10: Performance on **Emotion Perception and Recognition** using vanilla model. **Bold** and underlined indicate the best and the worst results among all models, respectively.

| Method | FESD | | ISA | | MESA | | MER | | MSA | | OSA | | SIA | | SOER | | SPER | | SCEA | |
|---|---|---|---|---|---|---|---|---|---|---|---|---|---|---|---|---|---|---|---|---|
| | ACC | WAF | ACC | WAF | ACC | WAF | ACC | WAF | ACC | WAF | ACC | WAF | ACC | WAF | ACC | WAF | ACC | WAF | ACC | WAF |
| *Open-Source Model* | | | | | | | | | | | | | | | | | | | | |
| VideoLLaMA3-7B | 61.78 | 61.51 | 46.85 | 45.68 | 21.60 | 18.28 | 52.18 | 50.91 | 64.62 | 63.91 | 67.89 | 69.98 | 35.20 | 34.04 | 45.80 | 41.03 | 41.80 | 36.02 | 42.00 | 41.50 |
| LLaVA-One-Vision-7B | 63.44 | 64.84 | 49.19 | 48.14 | 17.05 | 16.35 | 39.50 | 36.84 | 65.40 | 66.87 | 63.00 | 66.74 | 27.00 | 25.30 | 53.40 | 46.14 | 44.60 | 34.04 | 34.80 | 37.07 |
| LLaVA-NeXT-Video-7B | 54.44 | 58.53 | 41.20 | 37.44 | 11.85 | 9.63 | 41.31 | 38.42 | 56.11 | 61.65 | 65.80 | 70.53 | 25.03 | 20.28 | 48.60 | 46.10 | 43.40 | 40.54 | 31.20 | 32.72 |
| Qwen2.5-VL-7B | 62.00 | 64.90 | 43.15 | 43.13 | 21.25 | 16.97 | 56.75 | 57.24 | 61.21 | 65.79 | 64.20 | 71.71 | 32.60 | 29.54 | 52.80 | 44.18 | 41.80 | 32.13 | 47.20 | 45.77 |
| InternVL3-8B | 62.33 | 64.45 | 50.65 | 49.25 | 21.40 | 18.69 | 53.00 | 53.92 | 63.80 | 67.73 | 68.00 | 72.90 | 31.20 | 24.38 | 49.00 | 46.55 | 42.60 | 35.86 | 48.60 | 40.02 |
| MiniCPM-V-2.6-8B | 57.53 | 61.12 | 49.39 | 47.78 | 25.15 | 18.85 | 50.13 | 50.01 | 62.65 | 66.85 | 52.45 | 62.35 | 37.42 | 34.11 | 44.90 | 38.73 | 37.59 | 30.17 | 45.21 | 38.93 |
| Qwen2.5-VL-32B | 63.78 | 67.18 | 53.70 | 52.18 | 25.28 | 22.43 | 57.14 | 57.10 | 65.80 | 69.85 | 68.80 | 74.02 | 34.20 | 27.38 | 43.40 | 39.66 | 41.80 | 32.13 | 47.60 | 47.73 |
| InternVL3-38B | 63.22 | 66.39 | 53.58 | 52.53 | 24.00 | 22.63 | 57.16 | 68.57 | 68.80 | 71.62 | 68.80 | 74.98 | 35.73 | 31.92 | 53.46 | 48.58 | 48.00 | 40.18 | 50.60 | 42.28 |
| R1-Omni-0.5B | 42.28 | 46.78 | 51.55 | 52.18 | 23.72 | 24.68 | 50.88 | 49.29 | 41.74 | 47.95 | 32.20 | 42.27 | 19.50 | 16.04 | 30.12 | 26.87 | 24.38 | 22.10 | 43.60 | 33.28 |
| HumanOmni-7B | 64.44 | 66.68 | 53.77 | 53.81 | 23.82 | 24.97 | 56.75 | 53.46 | 48.20 | 49.47 | 35.20 | 44.54 | 33.90 | 34.28 | 50.31 | 40.74 | 46.20 | 37.37 | 47.60 | 31.13 |
| Qwen2.5-Omni-7B | 64.67 | 68.03 | 51.56 | 52.85 | 22.71 | 23.55 | 56.08 | 56.67 | 64.00 | 68.74 | 68.00 | 74.78 | 32.30 | 28.03 | 54.72 | 49.10 | 44.60 | 34.69 | 48.60 | 49.56 |
| Emotion-LLaMA-7B | 33.11 | 34.74 | 53.63 | 53.31 | 24.00 | 10.58 | 43.75 | 43.70 | 44.40 | 49.78 | 56.60 | 63.08 | 37.00 | 38.27 | 47.00 | 45.75 | 47.27 | 36.13 | 48.44 | 49.69 |
| AffectGPT-7B | 66.67 | 65.47 | 50.33 | 50.93 | 25.46 | 12.78 | 38.69 | 39.16 | 66.60 | 66.56 | 67.76 | 69.66 | 34.50 | 34.67 | 49.19 | 50.87 | 41.25 | 34.89 | 38.80 | 40.63 |
| *Closed-Source Model* | | | | | | | | | | | | | | | | | | | | |
| GPT-4o | 70.22 | 72.58 | 54.48 | 54.50 | 30.12 | 23.11 | 57.64 | 59.10 | 69.20 | 72.57 | 69.53 | 76.78 | 40.00 | 38.98 | 54.00 | 53.94 | 49.60 | 47.01 | 49.96 | 49.87 |
| GPT-4.1 | 71.46 | 73.75 | 56.80 | 57.15 | **31.43** | **27.20** | 64.00 | 64.40 | 72.46 | 75.01 | 69.60 | 77.14 | 40.81 | 41.34 | 66.19 | 65.64 | 55.20 | 50.91 | 53.20 | **54.24** |
| Gemini-2.5-Flash | 67.11 | 70.03 | 55.41 | 54.89 | 27.12 | 25.23 | 58.73 | 60.66 | 68.40 | 67.48 | 70.91 | 76.35 | 38.44 | 36.58 | 57.47 | 55.47 | 56.51 | 57.27 | 50.83 | 52.47 |
| Gemini-2.5-Pro | **78.39** | **78.22** | **61.12** | **61.29** | 28.96 | 25.52 | **72.11** | **72.78** | **74.20** | **76.24** | **75.71** | **79.66** | **46.53** | **43.93** | **67.96** | **67.27** | **65.00** | **63.08** | **55.02** | 51.40 |

Table 11: Performance on **Emotion Understanding and Analysis** using vanilla model.

| Method | DPTM | EBIA | HU | | IAVE | | MABSA | MQE | MSD | MDER | |
|---|---|---|---|---|---|---|---|---|---|---|---|
| | MF | ACC | ACC | WAF | ACC | WAF | MF | MF | ACC | ACC | WAF |
| *Open-Source Model* | | | | | | | | | | | |
| VideoLLaMA3-7B | 31.17 | 14.42 | 44.89 | 34.80 | 62.50 | 61.46 | 61.96 | 23.67 | 51.15 | 42.61 | 40.71 |
| LLaVA-One-Vision-7B | 31.54 | 11.33 | 42.25 | 29.96 | 60.37 | 58.70 | 63.89 | 14.02 | 39.75 | 33.00 | 26.52 |
| LLaVA-NeXT-Video-7B | 31.28 | 12.37 | 43.50 | 39.54 | 40.50 | 35.38 | 59.40 | 13.45 | 44.75 | 25.25 | 19.08 |
| Qwen2.5-VL-7B | 31.41 | 11.02 | 54.25 | 54.19 | 64.49 | 63.16 | 64.65 | 32.59 | 52.55 | 45.60 | 44.18 |
| InternVL3-8B | 36.77 | 14.79 | 53.50 | 53.25 | 60.13 | 57.03 | 63.11 | 33.13 | 50.90 | 39.48 | 34.80 |
| MiniCPM-V-2.6-8B | 30.80 | 14.51 | 49.25 | 47.32 | 55.03 | 54.04 | 61.82 | 27.77 | 39.85 | 43.51 | 41.38 |
| Qwen2.5-VL-32B | 40.22 | 14.62 | 57.50 | 57.43 | 62.67 | 62.20 | 63.29 | 32.38 | 52.08 | 48.20 | 48.64 |
| InternVL3-38B | 40.55 | 16.49 | 59.25 | 59.19 | 64.74 | 63.82 | 64.80 | 32.64 | 53.85 | 46.00 | 44.88 |
| R1-Omni-0.5B | 37.54 | 13.45 | 46.25 | 45.41 | 50.03 | 50.92 | 58.40 | 29.58 | 47.85 | 29.81 | 28.24 |
| HumanOmni-7B | 35.59 | 12.55 | 49.50 | 49.50 | 53.50 | 53.67 | 59.89 | 32.98 | 47.90 | 36.20 | 33.30 |
| Qwen2.5-Omni-7B | 31.63 | 11.42 | 53.00 | 52.77 | 55.79 | 53.85 | 61.93 | 31.09 | 44.65 | 37.68 | 35.09 |
| Emotion-LLaMA-7B | 39.54 | 15.46 | 57.92 | 57.92 | 52.08 | 52.04 | 60.13 | 34.18 | 44.15 | 47.59 | 48.04 |
| AffectGPT-7B | 34.17 | 12.27 | 56.50 | 56.47 | 40.07 | 37.99 | 60.48 | 30.95 | 42.40 | 37.92 | 37.28 |
| *Closed-Source Model* | | | | | | | | | | | |
| GPT-4o | 42.33 | 17.45 | 60.00 | 59.92 | 66.13 | 65.87 | 64.76 | 35.32 | 55.76 | 49.68 | 50.64 |
| GPT-4.1 | 47.50 | 18.62 | **70.19** | **70.17** | 67.68 | 67.78 | **70.81** | 37.98 | **65.76** | 53.82 | **53.82** |
| Gemini-2.5-Flash | 47.18 | 16.34 | 64.66 | 64.38 | 64.14 | 64.45 | 66.71 | 36.55 | 57.65 | 51.41 | 53.07 |
| Gemini-2.5-Pro | **49.23** | **19.25** | 69.39 | 66.98 | **70.67** | **70.99** | 67.61 | **39.23** | 64.95 | 52.65 | 53.59 |

Table 12: Performance on **Emotion Cognition and Reasoning** using vanilla model.

| Method | EER | | EI | LR | MECPE | SD | | SFA |
|---|---|---|---|---|---|---|---|---|
| | ACC | WAF | LLM | LLM | MF | ACC | WAF | EMF |
| *Open-Source Model* | | | | | | | | |
| VideoLLaMA3-7B | 45.31 | 37.85 | 31.29 | 39.56 | 13.09 | 37.56 | 36.90 | 13.16 |
| LLaVA-One-Vision-7B | 46.88 | 42.43 | 47.40 | 44.60 | 10.83 | 45.20 | 40.37 | 16.22 |
| LLaVA-NeXT-Video-7B | 35.94 | 28.33 | 46.80 | 43.10 | 13.05 | 46.36 | 36.37 | 18.42 |
| Qwen2.5-VL-7B | 40.62 | 29.14 | 50.53 | 48.20 | 15.07 | 49.00 | 45.43 | 14.64 |
| InternVL3-8B | 50.00 | 44.72 | 47.00 | 46.40 | 16.41 | 51.40 | 51.37 | 17.61 |
| MiniCPM-V-2.6-8B | 39.68 | 34.54 | 33.93 | 50.40 | 16.44 | 51.40 | 39.60 | 21.83 |
| Qwen2.5-VL-32B | 54.69 | 51.41 | 53.40 | 53.40 | 19.60 | 55.60 | 45.73 | 23.79 |
| InternVL3-38B | 50.31 | 46.63 | 50.67 | 51.40 | 19.28 | 55.80 | 55.58 | 25.73 |
| R1-Omni-0.5B | 39.67 | 38.65 | 43.73 | 43.00 | 16.13 | 53.00 | 52.91 | 19.93 |
| HumanOmni-7B | 38.85 | 30.76 | 47.93 | 28.40 | 13.19 | 49.40 | 49.15 | 16.43 |
| Qwen2.5-Omni-7B | 51.25 | 51.98 | 48.67 | 49.20 | 13.83 | 53.40 | 53.25 | 17.76 |
| Emotion-LLaMA-7B | 42.81 | 40.53 | 49.53 | 53.00 | 19.28 | 52.60 | 52.45 | 19.02 |
| AffectGPT-7B | 43.75 | 45.75 | 46.27 | 50.40 | 10.81 | 52.73 | 52.64 | 15.12 |
| *Closed-Source Model* | | | | | | | | |
| GPT-4o | 57.81 | 59.48 | 54.13 | 55.83 | 20.93 | 56.60 | 53.61 | 25.77 |
| GPT-4.1 | 60.31 | **65.49** | 57.67 | **61.04** | 26.86 | 66.20 | 65.09 | 36.73 |
| Gemini-2.5-Flash | 58.33 | 60.29 | 54.47 | 58.20 | 27.11 | 61.49 | 59.76 | 28.02 |
| Gemini-2.5-Pro | **66.13** | 65.41 | **65.13** | 59.23 | **33.33** | **66.61** | **66.62** | **41.22** |

Table 13: Performance on **Emotion Perception and Recognition** with proposed ToM prompting.

| Method | FESD | | ISA | | MESA | | MER | | MSA | | OSA | | SIA | | SOER | | SPER | | SCEA | |
|---|---|---|---|---|---|---|---|---|---|---|---|---|---|---|---|---|---|---|---|---|
| | ACC | WAF | ACC | WAF | ACC | WAF | ACC | WAF | ACC | WAF | ACC | WAF | ACC | WAF | ACC | WAF | ACC | WAF | ACC | WAF |
| *Open-Source Model* | | | | | | | | | | | | | | | | | | | | |
| VideoLLaMA3-7B | 54.18 | 58.59 | 47.93 | 45.43 | 23.01 | 21.83 | 44.22 | 45.47 | 60.58 | 63.72 | 68.15 | 72.62 | 31.80 | 32.38 | 49.09 | 43.23 | 40.60 | 34.29 | 45.83 | 44.63 |
| LLaVA-One-Vision-7B | 52.88 | 57.03 | 46.61 | 46.41 | 22.65 | 22.33 | 42.64 | 42.08 | 56.56 | 61.87 | 64.71 | 69.60 | 25.10 | 23.89 | 32.15 | 29.15 | 25.26 | 20.03 | 40.40 | 38.10 |
| LLaVA-NeXT-Video-7B | 52.16 | 54.83 | 45.55 | 43.84 | 22.23 | 20.12 | 39.88 | 35.57 | 55.10 | 59.02 | 52.71 | 56.98 | 27.00 | 26.42 | 37.11 | 35.55 | 40.29 | 38.48 | 31.20 | 33.77 |
| Qwen2.5-VL-7B | 67.50 | 68.58 | 44.71 | 42.47 | 23.90 | 21.34 | 55.95 | 56.54 | 65.66 | 68.52 | 73.80 | 75.77 | 32.33 | 32.36 | 48.48 | 42.59 | 33.06 | 26.29 | 39.20 | 39.99 |
| InternVL3-8B | 64.29 | 66.87 | 45.96 | 43.51 | 21.57 | 20.05 | 49.75 | 51.66 | 67.15 | 70.02 | 68.94 | 74.29 | 35.29 | 35.65 | 43.95 | 44.15 | 27.91 | 25.08 | 41.20 | 43.08 |
| MiniCPM-V-2.6-8B | 48.60 | 53.23 | 45.59 | 45.13 | 25.88 | 18.44 | 43.08 | 43.25 | 64.85 | 67.42 | 70.75 | 72.82 | 29.61 | 28.71 | 30.77 | 28.32 | 20.12 | 15.60 | 44.00 | 44.88 |
| Qwen2.5-VL-32B | 66.67 | 72.38 | 55.30 | 54.12 | 28.50 | 23.79 | 59.00 | 59.17 | 69.40 | 72.31 | 72.40 | 75.26 | 38.62 | 38.21 | 47.78 | 47.31 | 45.20 | 41.16 | 48.55 | 48.63 |
| InternVL3-38B | 68.22 | 70.61 | 54.15 | 51.77 | 25.88 | 19.88 | 58.65 | 59.66 | 71.60 | 73.99 | 72.00 | 75.09 | 38.21 | 37.76 | 55.02 | 53.73 | 49.40 | 45.18 | 51.80 | 52.20 |
| R1-Omni-0.5B | 43.80 | 47.35 | 39.62 | 38.14 | 23.48 | 21.19 | 50.77 | 50.34 | 54.85 | 57.39 | 42.83 | 48.94 | 30.48 | 30.76 | 41.43 | 40.81 | 41.28 | 40.90 | 47.00 | 46.06 |
| HumanOmni-7B | 62.00 | 65.21 | 47.97 | 53.99 | 22.98 | 19.77 | 55.83 | 51.92 | 64.40 | 67.76 | 61.40 | 67.93 | 31.10 | 31.04 | 40.31 | 30.63 | 39.00 | 30.12 | 46.41 | 46.51 |
| Qwen2.5-Omni-7B | 64.22 | 68.01 | 49.64 | 47.46 | 25.20 | 22.80 | 56.53 | 56.07 | 68.20 | 71.34 | 61.00 | 65.98 | 33.10 | 32.41 | 44.97 | 43.69 | 32.26 | 27.29 | 50.40 | 43.26 |
| Emotion-LLaMA-7B | 60.31 | 63.47 | 34.96 | 39.78 | 20.20 | 14.72 | 39.00 | 31.89 | 62.32 | 66.65 | 43.80 | 52.53 | 26.94 | 21.60 | 38.80 | 31.05 | 23.20 | 14.98 | 29.20 | 37.52 |
| AffectGPT-7B | 40.14 | 42.71 | 46.84 | 46.28 | 21.23 | 19.75 | 30.65 | 29.77 | 55.62 | 59.10 | 60.60 | 61.30 | 31.55 | 31.64 | 28.30 | 24.62 | 21.12 | 21.10 | 39.60 | 39.53 |
| *Closed-Source Model* | | | | | | | | | | | | | | | | | | | | |
| GPT-4o | 74.00 | 75.74 | 56.44 | 56.41 | 33.18 | 30.36 | 63.32 | 63.73 | 74.60 | 75.75 | 73.87 | 75.89 | 41.34 | 41.05 | 56.10 | 55.15 | 54.31 | 54.12 | 52.80 | 50.49 |
| GPT-4.1 | 74.74 | 76.20 | 58.06 | 58.59 | **34.55** | 28.36 | 66.00 | 65.96 | 76.06 | 77.31 | 74.80 | 77.65 | 43.17 | 44.07 | 69.19 | 69.71 | 57.20 | 52.92 | 56.01 | 52.37 |
| Gemini-2.5-Flash | 76.44 | 74.73 | 59.01 | 57.85 | 29.20 | 28.86 | 64.19 | 65.05 | 74.84 | 74.97 | 76.03 | 77.54 | 43.36 | 42.33 | 63.33 | 63.69 | 62.22 | 64.06 | 53.04 | 53.72 |
| Gemini-2.5-Pro | **79.11** | **79.42** | **63.13** | **66.89** | 31.02 | 28.86 | **72.92** | **73.05** | **77.97** | **78.91** | **79.19** | **80.85** | **51.74** | **51.75** | 69.00 | **69.99** | **69.31** | **69.62** | **62.25** | **62.85** |

Table 14: Performance on **Emotion Understanding and Analysis** with proposed ToM prompting.

| Method | DPTM | EBIA | HU | | IAVE | | MABSA | MQE | MSD | MDER | |
|---|---|---|---|---|---|---|---|---|---|---|---|
| | MF | ACC | ACC | WAF | ACC | WAF | MF | MF | ACC | ACC | WAF |
| *Open-Source Model* | | | | | | | | | | | |
| VideoLLaMA3-7B | 26.75 | 12.42 | 56.00 | 53.07 | 66.41 | 65.15 | 61.37 | 22.86 | 43.60 | 34.49 | 31.83 |
| LLaVA-One-Vision-7B | 24.72 | 10.21 | 51.01 | 47.85 | 64.44 | 62.80 | 60.66 | 20.18 | 39.15 | 34.90 | 31.62 |
| LLaVA-NeXT-Video-7B | 25.89 | 14.53 | 54.50 | 53.84 | 40.81 | 38.32 | 61.32 | 24.18 | 43.12 | 32.58 | 31.66 |
| Qwen2.5-VL-7B | 20.12 | 12.27 | 54.25 | 46.41 | 44.17 | 41.84 | 64.30 | 22.79 | 46.90 | 41.30 | 40.16 |
| InternVL3-8B | 29.76 | 12.88 | 56.75 | 53.00 | 62.08 | 59.09 | 66.67 | 22.42 | 44.00 | 38.00 | 34.01 |
| MiniCPM-V-2.6-8B | 29.36 | 14.52 | 56.75 | 55.85 | 70.26 | 69.13 | 61.42 | 22.63 | 38.15 | 39.37 | 38.28 |
| Qwen2.5-VL-32B | 41.20 | 16.12 | 65.25 | 64.04 | 75.46 | 74.33 | 68.37 | 33.56 | 59.95 | 49.64 | 49.17 |
| InternVL3-38B | 40.28 | 19.08 | 64.75 | 61.07 | 67.45 | 66.05 | 66.50 | 34.21 | 55.65 | 48.83 | 43.30 |
| R1-Omni-0.5B | 26.21 | 14.57 | 46.43 | 38.71 | 55.74 | 56.76 | 56.29 | 22.24 | 42.78 | 31.95 | 28.42 |
| HumanOmni-7B | 26.14 | 12.89 | 55.64 | 54.24 | 57.21 | 57.75 | 66.39 | 22.61 | 42.49 | 38.80 | 36.13 |
| Qwen2.5-Omni-7B | 25.67 | 11.20 | 49.75 | 34.08 | 57.23 | 55.08 | 64.54 | 25.38 | 46.60 | 35.55 | 32.56 |
| Emotion-LLaMA-7B | 25.25 | 10.91 | 50.00 | 33.33 | 32.51 | 25.84 | 58.52 | 22.37 | 39.51 | 39.06 | 32.11 |
| AffectGPT-7B | 25.93 | 11.98 | 52.31 | 43.61 | 43.40 | 42.90 | 63.29 | 26.16 | 44.39 | 35.21 | 34.94 |
| *Closed-Source Model* | | | | | | | | | | | |
| GPT-4o | 45.90 | 25.70 | 66.63 | 66.51 | 71.19 | 70.92 | 68.30 | 36.45 | 61.04 | 53.72 | 55.24 |
| GPT-4.1 | 49.47 | 27.65 | **78.00** | **77.68** | 72.91 | 72.75 | **77.70** | 40.91 | 66.18 | 57.85 | 59.83 |
| Gemini-2.5-Flash | 56.35 | 24.87 | 65.74 | 65.57 | 72.63 | 72.01 | 73.21 | 38.12 | 61.83 | 55.56 | 57.36 |
| Gemini-2.5-Pro | **59.21** | **28.68** | 71.83 | 71.42 | **77.20** | **77.78** | 75.43 | **44.47** | **73.79** | **58.90** | **60.13** |

Table 15: Performance on **Emotion Cognition and Reasoning** with proposed ToM prompting.

| Method | EER | | EI | LR | MECPE | SD | | SFA |
|---|---|---|---|---|---|---|---|---|
| | ACC | WAF | LLM | LLM | MF | ACC | WAF | EMF |
| *Open-Source Model* | | | | | | | | |
| VideoLLaMA3-7B | 50.00 | 43.30 | 55.00 | 51.80 | 15.18 | 48.88 | 32.09 | 18.29 |
| LLaVA-One-Vision-7B | 40.62 | 37.90 | 52.00 | 50.60 | 12.42 | 50.10 | 33.67 | 19.22 |
| LLaVA-NeXT-Video-7B | 28.81 | 25.97 | 36.67 | 51.28 | 10.37 | 53.60 | 53.36 | 20.33 |
| Qwen2.5-VL-7B | 48.44 | 43.90 | 55.53 | 59.59 | 19.30 | 50.00 | 44.91 | 18.33 |
| InternVL3-8B | 54.69 | 51.70 | 62.73 | 57.56 | 15.32 | 51.70 | 38.03 | 30.54 |
| MiniCPM-V-2.6-8B | 39.68 | 34.41 | 60.13 | 56.80 | 17.46 | 48.35 | 40.16 | 19.99 |
| Qwen2.5-VL-32B | 58.42 | 52.10 | 57.16 | 64.21 | 22.14 | 61.53 | 53.24 | 34.91 |
| InternVL3-38B | 55.50 | 51.33 | 55.32 | 57.13 | 23.33 | 64.72 | 57.54 | 33.61 |
| R1-Omni-0.5B | 41.67 | 43.23 | 30.59 | 60.72 | 15.02 | 56.68 | 56.21 | 21.22 |
| HumanOmni-7B | 37.50 | 36.90 | 49.07 | 43.09 | 17.62 | 51.60 | 38.61 | 22.94 |
| Qwen2.5-Omni-7B | 40.62 | 35.36 | 50.07 | 63.60 | 13.65 | 46.20 | 42.36 | 30.65 |
| Emotion-LLaMA-7B | 38.71 | 44.02 | 38.73 | 59.80 | 12.38 | 49.60 | 35.47 | 31.07 |
| AffectGPT-7B | 32.26 | 37.83 | 46.73 | 60.08 | 12.83 | 53.51 | 52.50 | 40.49 |
| *Closed-Source Model* | | | | | | | | |
| GPT-4o | 60.00 | 63.41 | 64.33 | 66.00 | 22.48 | 64.80 | 62.49 | 42.29 |
| GPT-4.1 | 65.86 | **77.98** | 69.00 | **71.79** | 28.11 | 68.67 | **68.63** | 47.75 |
| Gemini-2.5-Flash | 64.13 | 75.71 | 63.93 | 66.60 | 31.43 | 64.10 | 64.08 | 45.22 |
| Gemini-2.5-Pro | **71.94** | 75.73 | **70.27** | 68.20 | **37.70** | **69.00** | **68.63** | **52.78** |

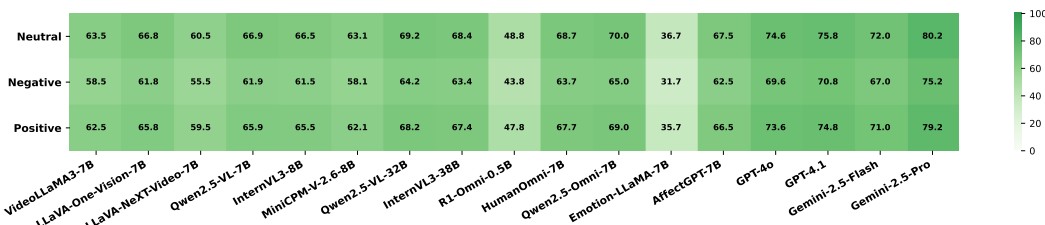

Figure 8: **Task-level Performance Comparison on FESD task.**

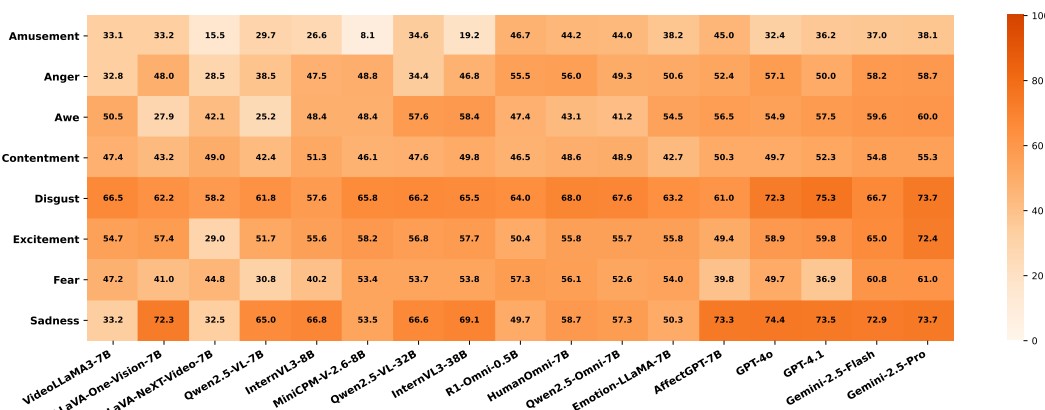

Figure 9: **Task-level Performance Comparison on ISA task.**

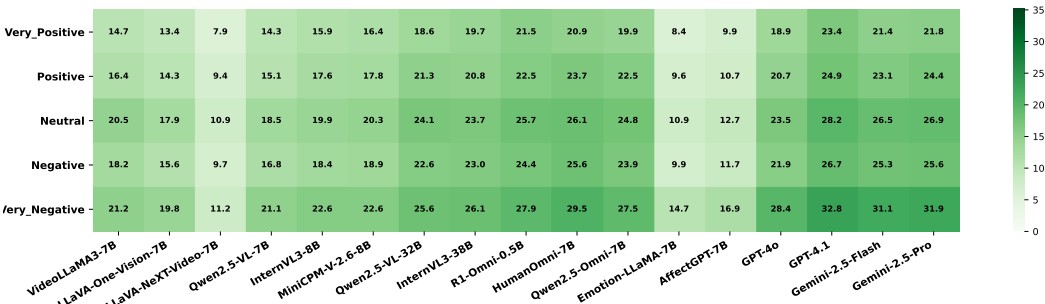

Figure 10: **Task-level Performance Comparison on MESA task.**

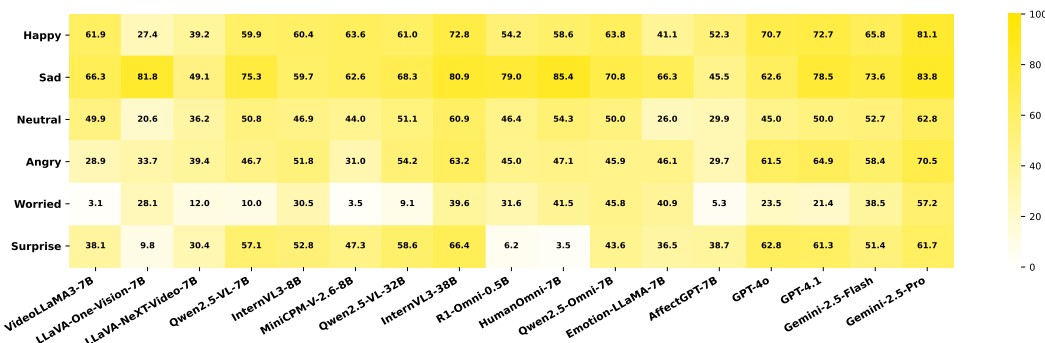

Figure 11: **Task-level Performance Comparison on MER task.**

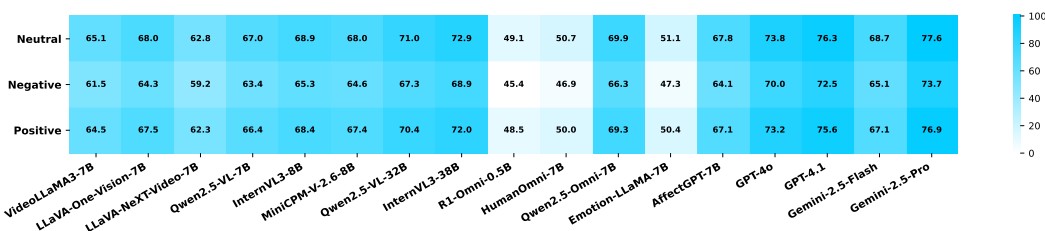

Figure 12: **Task-level Performance Comparison on MSA task.**

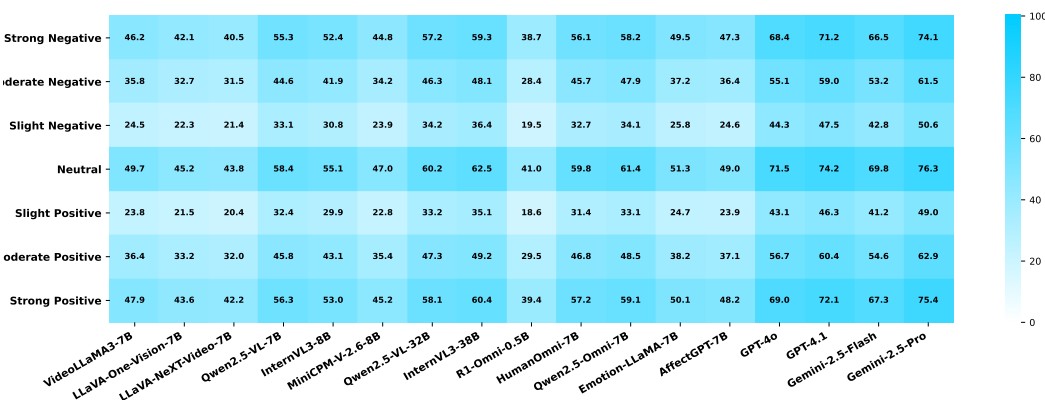

Figure 13: **Task-level Performance Comparison on SIA task.**

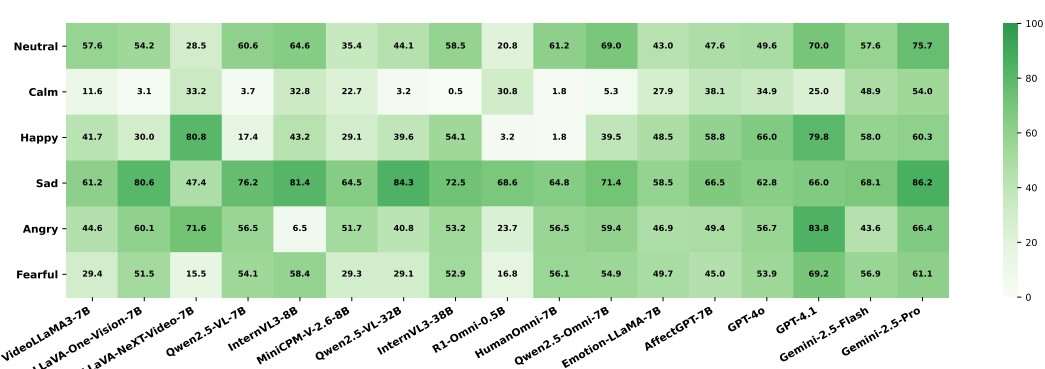

Figure 14: **Task-level Performance Comparison on SOER task.**

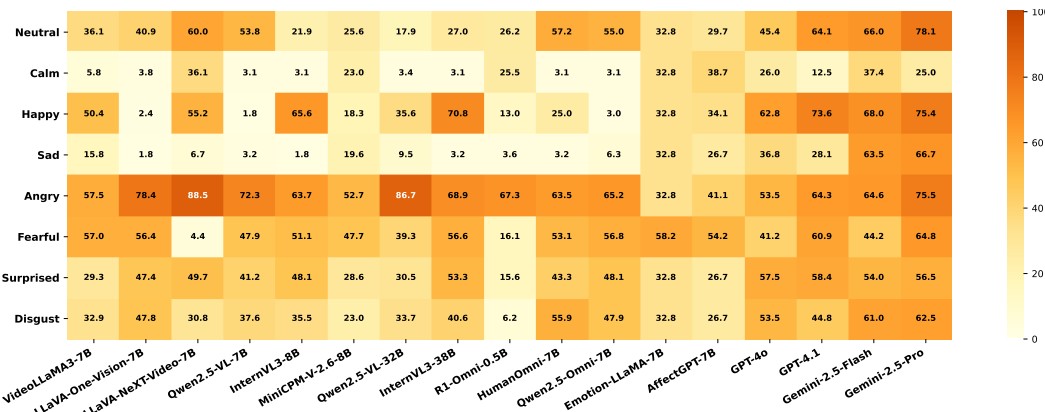

Figure 15: **Task-level Performance Comparison on SPER task.**

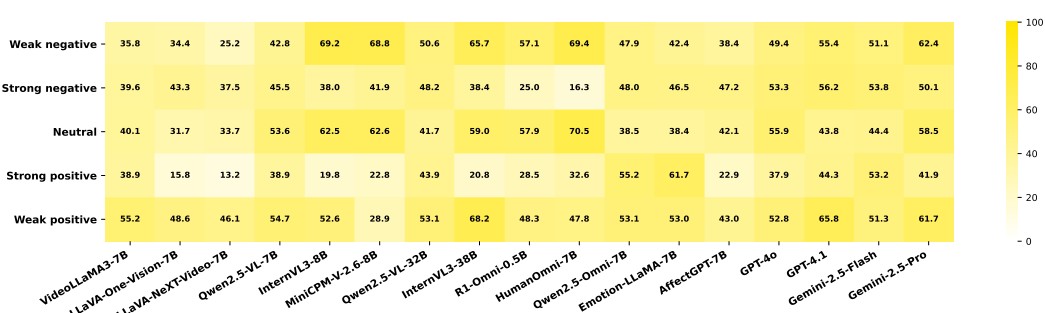

Figure 16: **Task-level Performance Comparison on SCEA task.**

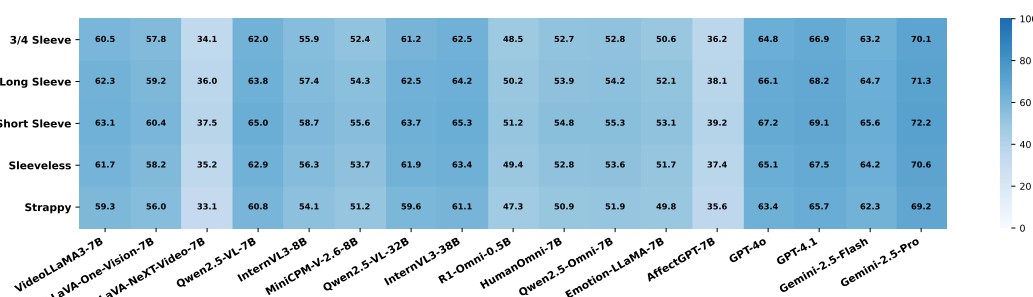

Figure 17: **Task-level Performance Comparison on IAVE task.**

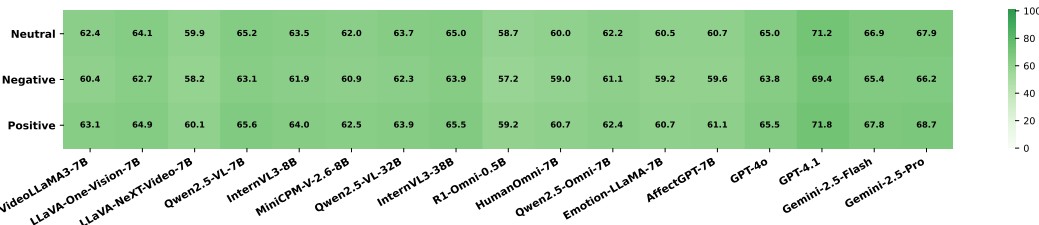

Figure 18: **Task-level Performance Comparison on MABSA task.**

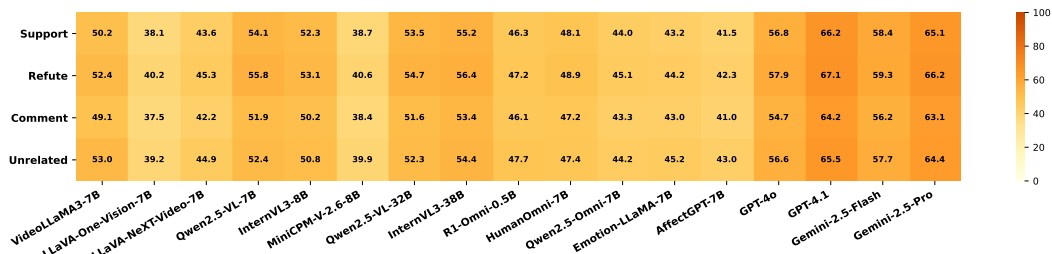

Figure 19: **Task-level Performance Comparison on MSD task.**

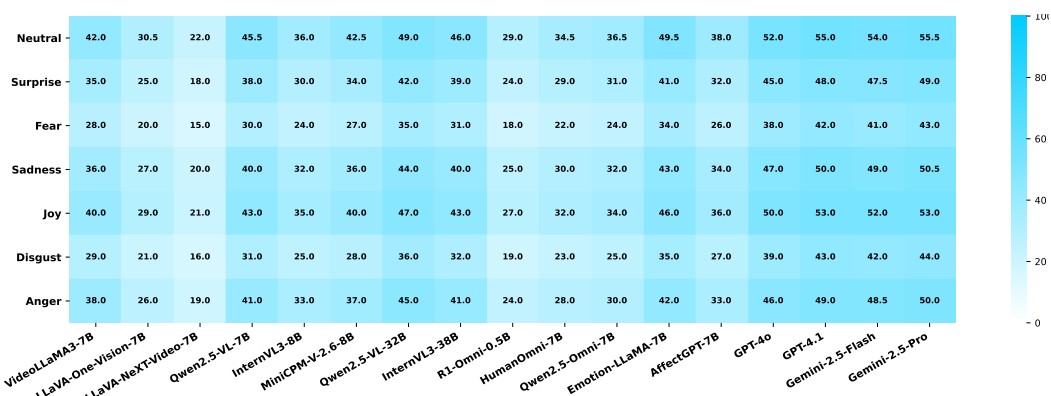

Figure 20: **Task-level Performance Comparison on MDER task.**

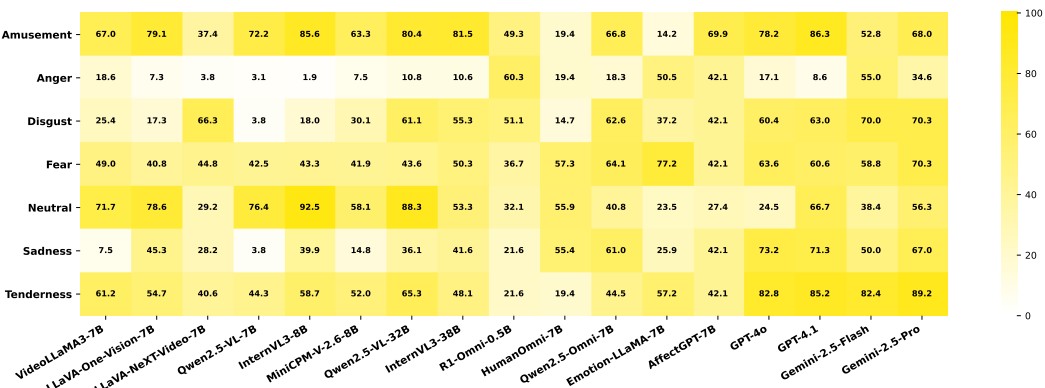

Figure 21: **Task-level Performance Comparison on EER task.**

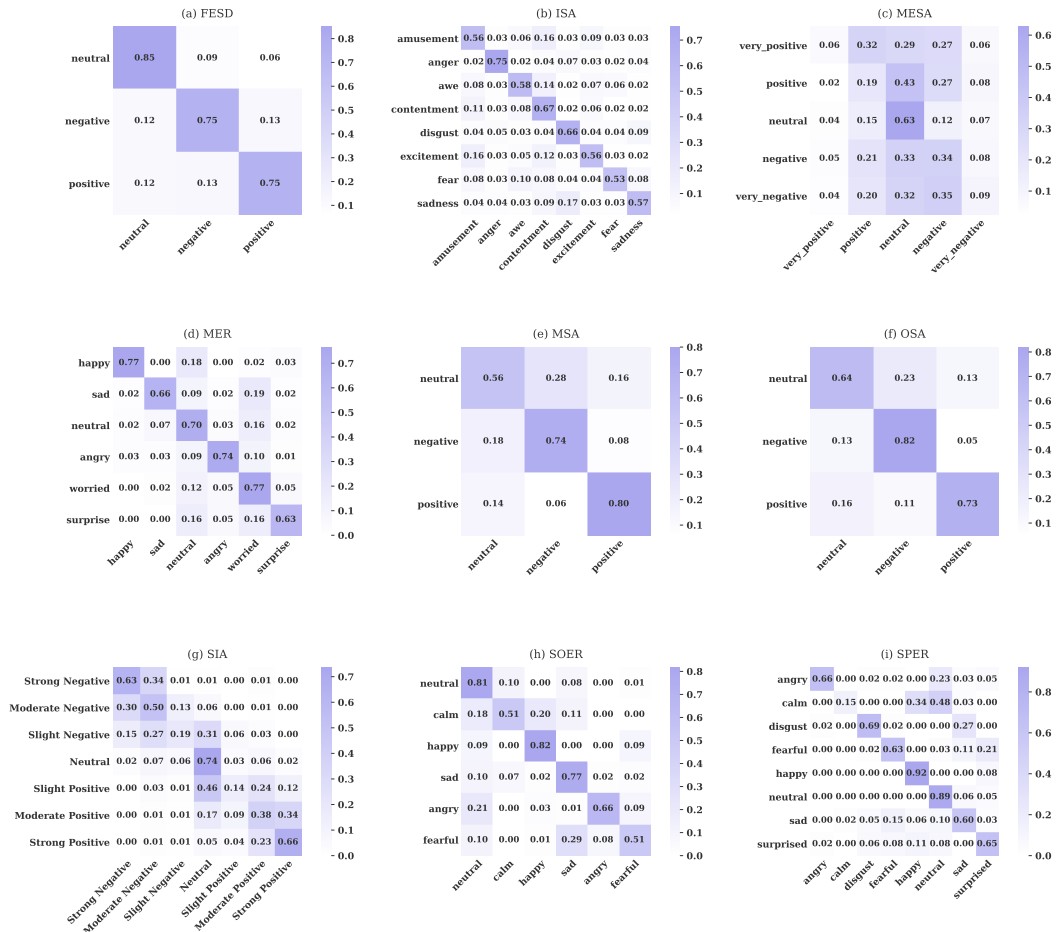

Figure 22: **Confusion matrices for Gemini-2.5-Pro on each task (Part 1).**

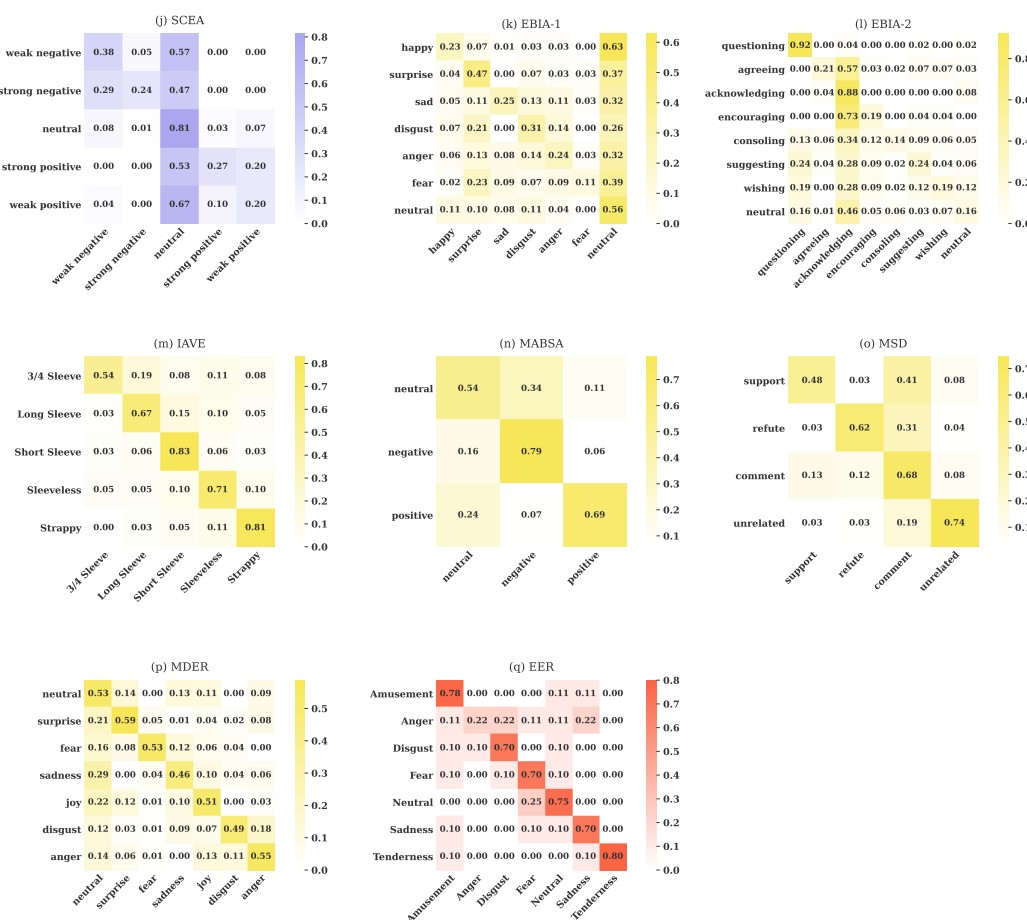

Figure 23: **Confusion matrices for Gemini-2.5-Pro on each task (Part 2).**

## F    DESIGN OF TOM-STYLE PROMPTS

We adopt a unified ToM-style prompting scaffold across three levels, aligned with the progression of our framework. Level 1 operationalizes first-order affect attribution through a four-stage chain—Perceptual Simulation, Cognitive Empathy, Perspective-Taking, and Conclude-and-Map—illustrated in Figure 24 to Figure 33. Level 2 extends this scaffold to relational and contextual mind modeling, where perceived states are linked to entities, aspects, and communicative goals (state → about(entity, context)). Representative templates are shown in Figure 34 to Figure 41. Level 3 advances to causal attribution and second-order reasoning, focusing on why emotions arise, how they shift, and how they are socially interpreted (cause attribution and recursive mind modeling). Illustrative templates are provided in Figure 42 to Figure 47.

### F.1    DETAILED PROMPT DESIGN RATIONALE

- **Face Expression Sentiment Detection.** This task instantiates basic ToM attribution by inferring a subject's immediate affect from facial micro-expressions, gaze, posture, and prosody. The model integrates convergent and divergent cues into a coherent here-and-now hypothesis, explicitly excluding observer bias or trait-based assumptions. Attribution remains grounded in the subject's perspective, ensuring that emotion labels reflect their mental state rather than external interpretation.

- **Image Sentiment Analysis.** This task extends ToM reasoning to full-scene interpretation, requiring attribution of an affective stance from either visible human subjects or environmental affordances such as threat, celebration, or serenity. The model infers what an experiencer—depicted or implied—would feel, grounding sentiment in context rather than in the observer's reaction. This design highlights scene-level ToM attribution by linking visual evidence to an imagined experiencer's mental state.

- **Meme Sentiment Analysis.** This task reframes sentiment detection as communicative intent attribution. The model integrates text and image cues, treating convergence as straightforward reinforcement and divergence as deliberate rhetorical strategy (e.g., sarcasm, irony, humor). Sentiment is attributed from the creator's perspective toward the intended audience, embedding ToM reasoning in the recognition of communicative goals.

- **Multimodal Emotion Recognition.** This task generalizes first-order ToM attribution across multiple input channels—visual behavior, prosody, and lexical content. The model integrates these cues into a coherent hypothesis of the speaker's immediate emotional state, resolving convergence and divergence strictly on observable evidence. Attribution reflects the speaker's perspective, ensuring recognition captures their current mental state.

- **Multimodal Sentiment Analysis.** This task shifts from emotion recognition to polarity attribution, asking whether the speaker expresses a positive, negative, or neutral stance. Multimodal cues are synthesized into a stance hypothesis, with attribution grounded in the speaker's evaluative perspective rather than external judgments. The design distinguishes evaluative positioning from emotion states while preserving ToM-based reasoning.

- **Opinion Sentiment Analysis.** This task emphasizes propositional attitudes, attributing polarity toward a stated proposition. The model decodes evaluative lexical markers alongside multimodal cues and synthesizes them into a hypothesis about the speaker's stance. Attribution is explicitly tied to the speaker's point of view, ensuring that polarity judgments are context-sensitive rather than generic affect labels.

- **Sentiment Intensity Analysis.** This task advances polarity attribution by incorporating graded strength. The model decodes lexical intensifiers, prosodic emphasis, and visual force/tension to distinguish slight, moderate, or strong polarity expressions. Attribution remains anchored in the speaker's immediate evaluative stance, enabling finer-grained distinctions within ToM-based sentiment reasoning.

- **Song Emotion Recognition.** This task applies ToM reasoning to performance contexts, attributing enacted emotional states conveyed by singers or performers. The model integrates facial, bodily, and acoustic-musical cues, with lyrics considered when present. Attribution is framed from the performer's expressive perspective, capturing intended affective enactment rather than audience response.

- **Speech Emotion Recognition.** This task applies ToM attribution to spoken interaction, decoding acoustic-prosodic features, articulatory-visual cues, and lexical content as evidence of inner state. The model integrates these signals into a hypothesis of the speaker's immediate emotion, ensuring attribution reflects the speaker's mental world rather than the listener's impression.

- **Stock Comment Emotion Analysis.** This task adapts ToM reasoning to financial discourse, attributing a commenter's evaluative stance toward a financial target with graded polarity strength. Lexical cues such as hedging, certainty, or numeric framing are central, with prosodic/visual markers incorporated when available. Attribution reflects the commenter's current evaluative orientation, distinguishing weak versus strong polarity in context.

- **Emotion-Based Intent Analysis.** This task extends ToM reasoning from first-order emotion attribution to relational modeling of communicative intent in dialogue. The model decodes lexical, prosodic, visual, and dialogic cues as traces of the speaker's state, integrates them into an emotion hypothesis, and then contextualizes this stance as intent toward the addressee (e.g., questioning, consoling, encouraging). Attribution reflects the transition from state to state→about(addressee, context), binding emotions to pragmatic communicative goals.

- **Humor Understanding.** This task applies second-order ToM reasoning, requiring the model to capture how a speaker amuses an audience by violating expectations. The model decodes setup, punchline, and delivery cues, constructs an audience expectation baseline, and checks for mismatches such as reversals or double meanings. Attribution is made from the speaker→audience perspective, framing humor as communicative intent based on incongruity resolution.

- **Implicit Attribute Value Extraction.** This task adapts ToM reasoning to product interpretation, treating product presentation as a communicative act between designer and observer. The model decodes visual design cues together with metadata, interprets them as intentional signals of hidden properties, and maps them onto valid attribute values. Attribution thus reframes classification as relational reasoning about design intent and observer inference.

- **Multimodal Aspect-Based Sentiment Analysis.** This task extends sentiment attribution to multiple targets and aspects, requiring structured stance separation. The model decodes multimodal and referential cues, integrates them into stance hypotheses for each target/aspect, and interprets divergences as possible rhetorical devices while grounding strictly in evidence. Attribution reflects the author's perspective toward each entity, producing distinct and contextually bound polarity labels.

- **Multimodal Quintuple Extraction.** This task formalizes relational stance mapping by extracting structured units of evaluation. The model decodes evaluative cues, resolves holder identity and coreference, and infers holder–target–aspect relations. Attribution is expressed as quintuples (holder, target, aspect, opinion, sentiment), ensuring attitudes are contextualized, evidence-based, and relationally organized beyond raw polarity classification.

- **Multimodal Stance Detection.** This task links an author's evaluative state to a specific claim or target, distinguishing stance from generic sentiment. The model decodes multimodal stance cues, attributes the author's immediate attitude toward the target, and maps it into support, refute, comment, or unrelated. Attribution explicitly conditions inference on target-specific positioning, aligning with ToM reasoning about communicative orientation.

- **Multiparty Dialogue Emotion Recognition.** This task situates emotion attribution within multi-party exchanges. The model decodes lexical, prosodic, and visual cues for the focal speaker, integrates them into an emotion hypothesis, and refines the attribution using roles, turn-taking, and interactional context. Attribution reflects ToM reasoning about how a speaker's state is shaped and signaled within dialogue structure, ensuring role- and context-sensitive recognition.

- **Emotion Elicitation Reasoning.** This task shifts ToM reasoning to second-order attribution, modeling how a generic viewer, rather than the characters, appraises events. The model decodes narrative and cinematic cues as potential affect triggers, constructs viewer appraisals along dimensions such as goal congruence, threat, or attachment, and maps them to a single elicited emotion. Attribution is grounded in the causal link between specific

events/devices and the audience's reaction, highlighting what the scene makes viewers feel and why.

- **Emotion Interpretation.** This task explains why a subject experiences a given emotion by reconstructing appraisal pathways. The model decodes observable cues (expressions, posture, events, objects), builds a subject-centered appraisal hypothesis (e.g., goal obstruction, threat, social evaluation), and attributes the emotion to proximate, visible causes. ToM reasoning is operationalized as event $\rightarrow$ appraisal $\rightarrow$ emotion mapping, producing concise, evidence-based explanations.

- **Laughter Reasoning.** This task applies second-order ToM reasoning to explain why laughter occurs. The model decodes setup, punchline, and delivery cues, models audience expectations, and identifies the humor trigger (e.g., incongruity, reversal, irony, norm violation). Attribution explains how the mismatch causes reinterpretation into amusement, framing laughter as the outcome of expectation management and speaker intent.

- **Multimodal Emotion Cause Pair Extraction.** This task extends emotion recognition to explicit cause–effect mapping in dialogue. The model decodes cues in a target utterance, builds a subject-centered appraisal hypothesis, and links it to the most proximate prior utterance that explains the emotion, enforcing temporal precedence. Attribution results in explicit emotion–cause pairs, embedding ToM reasoning about interpersonal dynamics and conversational elicitation.

- **Sarcasm Detection.** This task frames sarcasm as nonliteral intent attribution requiring second-order reasoning. The model decodes the literal proposition and surface polarity, models speaker–audience dynamics, and tests for incongruity–reversal where context signals the opposite of what is said. Attribution distinguishes sarcasm from humor or exaggeration by grounding meaning in the speaker's intent for the audience to infer a reversed stance.

- **Sentiment Flip Analysis.** This task tracks how sentiments evolve across dialogue, attributing changes to conversational causes. The model decodes polarity and discourse cues, builds sentiment timelines for each speaker, and detects flips from one stance to another. Attribution assigns trigger types (e.g., new information, argument, feedback, self-reflection) by enforcing temporal and causal reasoning. This highlights ToM's role in modeling dynamic shifts in evaluation rather than static judgments.

## 1) Face Expression Sentiment Detection

*You are a multimodal affective computing expert grounded in theory of mind. Your task is to infer the subject's current emotional state by reasoning about their mental world from observable evidence. Complete the face expression emotion detection task as follows:*

**(1) Decode Affective Signals (Perceptual Simulation):**
Systematically decode observable affect displays—facial features, gaze, body posture, prosody (if available), and verbal content/transcript (if provided)—as outward evidence of an inner state. Briefly describe each present cue and its likely valence.

**(2) Synthesize a Mental State Hypothesis (Cognitive Empathy):**
Integrate the decoded signals to form a coherent first-order hypothesis about what the subject is feeling now. If the signals converge, state that they support the same valence. If some signals diverge, acknowledge the discrepancy but ground your hypothesis strictly in the observable evidence without speculating beyond it.

**(3) Attribute from the Subject's Perspective (Perspective-Taking):**
Refine the hypothesis to attribute the most plausible here-and-now affective valence from the subject's perspective. Exclude interpretations based on the viewer's reaction or on the subject's stable personality traits.

**(4) Conclude and Map to Label:**
Provide your reasoning following the steps above, then choose exactly one label from: neutral, negative, positive, and output exactly {'emotion': 'label'}.

Figure 24: **ToM-style prompting for Face Expression Sentiment Detection.**

## 2) Image Sentiment Analysis

*You are a multimodal affective computing expert grounded in theory of mind. Your task is to infer the single dominant emotion conveyed by the image by reasoning about an (explicit or implied) human experiencer's mental state from observable visual evidence. Complete the image sentiment analysis task as follows:*

**(1) Decode Affective Signals (Perceptual Simulation):**
Systematically decode visible affective cues as outward evidence of an inner state. If human subjects are visible, note facial expression, body posture, and interpersonal actions. If no clear subject is visible, note the scene's affective affordances (e.g., threat/safety cues, contamination/decay vs cleanliness, celebration/-goal attainment, loss/damage, vastness/majesty, serenity/coziness) together with color/lighting and composition. Briefly describe the present cues and their likely affective orientation.

**(2) Synthesize a Mental State Hypothesis (Cognitive Empathy):**
Integrate the decoded cues into a coherent first-order hypothesis about what a person in or observing this scene would feel now. Indicate whether the cues converge on the same emotion or partially diverge; if they diverge, acknowledge the discrepancy but ground the hypothesis strictly in what is visible (avoid speculation beyond the image).

**(3) Attribute from the Experiencer's Perspective (Perspective-Taking):**
Attribute the most plausible here-and-now affect from the experiencer's perspective (the depicted person, if present; otherwise an implied observer embedded in the scene). Exclude interpretations based on your own idiosyncratic preferences or on stable personality traits.

**(4) Conclude and Map to Label:**
Provide your reasoning following the steps above, then choose exactly one label from: amusement, anger, awe, contentment, disgust, excitement, fear, sadness, and output exactly {'emotion': 'label'}.

Figure 25: **ToM-style prompting for Image Sentiment Analysis.**

😌 **3) Meme Sentiment Analysis**

*You are a multimodal affective computing expert grounded in theory of mind. Your task is to infer the meme's overall affective attitude by reasoning about the communicative intent behind it. Complete the meme sentiment analysis task as follows:*

**(1) Decode Affective Signals (Perceptual Simulation):**
Systematically decode observable signals from both the image (faces, actions, symbols) and the text (wording, emphasis, emojis). Briefly note each cue and its literal, surface-level polarity.

**(2) Analyze Multimodal Interaction and Infer Intent (Cognitive Empathy):**
Analyze the relationship between the image and text cues. If cues converge: State that they reinforce each other to express a straightforward attitude. If cues diverge: Treat this conflict as a critical clue. Consider if the discrepancy is intentional to create a non-literal meaning like sarcasm, irony, or humor. The meme's true attitude often arises from interpreting this very conflict. Formulate a hypothesis about the meme's intended communicative goal.

**(3) Attribute from the Meme's Communicative Perspective (Perspective-Taking):**
Based on your analysis of the multimodal interaction and inferred intent, attribute the single most plausible affective attitude being expressed by the meme as a communicative act. This is the attitude the meme's creator intends for a generic viewer to understand. Exclude your own reaction or judgments about people depicted.

**(4) Conclude and Map to Label:**
Provide your reasoning following the steps above, then choose exactly one label from: very_positive, positive, neutral, negative, very_negative, and output exactly {'emotion': 'label'}.

Figure 26: **ToM-style prompting for Meme Sentiment Analysis.**

**4) Multimodal Emotion Recognition**

*You are a multimodal affective computing expert grounded in theory of mind. Your task is to infer the speaker's current emotional state by reasoning about their mental world from multimodal evidence. Complete the multimodal emotion recognition task as follows:*

**(1) Decode Affective Signals (Perceptual Simulation):**
Systematically decode observable cues in each available channel—visual behavior (face, gaze, posture, gesture), speech prosody (tone, pitch, intensity, tempo) if available, and lexical content/transcript if provided—as outward evidence of an inner state. Briefly note the present cues and their likely emotion implications.

**(2) Synthesize a Mental State Hypothesis (Cognitive Empathy):**
Integrate the decoded cues into a coherent first-order hypothesis about what the speaker is feeling now. If the cues converge, state that they reinforce the same valence. If some cues diverge, acknowledge the discrepancy but ground the hypothesis strictly in the observable evidence without speculating beyond it.

**(3) Attribute from the Speaker's Perspective (Perspective-Taking):**
Refine the hypothesis to attribute the single most plausible here-and-now emotional state from the speaker's perspective. Exclude interpretations based on the viewer's reaction or on the speaker's stable traits.

**(4) Conclude and Map to Label:**
Provide your reasoning following the steps above, then choose exactly one label from: happy, sad, neutral, angry, worried, surprise, and output exactly {'emotion': 'label'}.

Figure 27: **ToM-style prompting for Multimodal Emotion Recognition.**

**5) Multimodal Sentiment Analysis**

*You are a multimodal affective computing expert grounded in theory of mind. Your task is to infer the speaker's overall evaluative polarity by reasoning about their current mental state from multimodal evidence. Complete the multimodal sentiment analysis task as follows:*

**(1) Decode Affective Signals (Perceptual Simulation):**
Systematically decode observable cues in each available channel—visual behavior (face, gaze, posture, gesture), speech prosody (tone, pitch, intensity, tempo) if available, and lexical content/transcript if provided—as outward evidence of an inner evaluative stance. Briefly note the present cues and their likely polarity implications.

**(2) Synthesize a Mental State Hypothesis (Cognitive Empathy):**
Integrate the decoded cues into a coherent first-order hypothesis about what overall polarity the speaker is expressing now. If the cues converge, state that they reinforce the same polarity. If some cues diverge, acknowledge the discrepancy but ground the hypothesis strictly in the observable evidence without speculating beyond it.

**(3) Attribute from the Speaker's Perspective (Perspective-Taking):**
Refine the hypothesis to attribute the single most plausible here-and-now evaluative polarity from the speaker's perspective (not the viewer's reaction and not a stable trait).

**(4) Conclude and Map to Label:**
Provide your reasoning following the steps above, then choose exactly one label from: neutral, negative, positive, and output exactly {'emotion': 'label'}.

Figure 28: **ToM-style prompting for Multimodal Sentiment Analysis.**

**6) Opinion Sentiment Analysis**

*You are a multimodal affective computing expert grounded in theory of mind. Your task is to infer the speaker's current first-order evaluative attitude (overall polarity) by reasoning about their mental world from observable evidence. Complete the opinion sentiment analysis task as follows:*

**(1) Decode Propositional & Affective Signals (Perceptual Simulation):**
Systematically decode the spoken/ textual content to identify opinionated predicates, evaluative terms, intensifiers/negations, and modality markers (lexical content/transcript if provided). Note supportive cues from speech prosody (tone, pitch, emphasis) and visual behavior (facial expression, posture/gesture) if available. Briefly describe each present cue and its likely polarity.

**(2) Synthesize a Propositional Attitude Hypothesis (Cognitive Empathy):**
Integrate the decoded cues into a coherent here-and-now hypothesis about the speaker's evaluative stance toward what is being talked about. If the cues converge, state that they reinforce the same polarity. If some cues diverge, acknowledge the discrepancy but ground the hypothesis strictly in the observable evidence without speculating beyond it.

**(3) Attribute from the Speaker's Perspective (Perspective-Taking):**
Refine the hypothesis to attribute the single most plausible immediate overall polarity from the speaker's perspective (not the viewer's reaction and not a stable trait).

**(4) Conclude and Map to Label:**
Provide your reasoning following the steps above, then choose exactly one label from: neutral, negative, positive, and output exactly {'emotion': 'label'}.

Figure 29: **ToM-style prompting for Opinion Sentiment Analysis.**

**7) Sentiment Intensity Analysis**

*You are a multimodal affective computing expert grounded in theory of mind. Your task is to infer the speaker's current first-order evaluative state with graded intensity by reasoning about their mental world from multimodal evidence. Complete the sentiment intensity analysis task as follows:*

**(1) Decode Polarity & Intensity Signals (Perceptual Simulation):**
Systematically decode observable cues as outward evidence of an inner state—lexical content/transcript (if provided: evaluative terms, intensifiers/diminishers, negation, capitalization, repetition/elongation, emojis), speech prosody (if available: pitch, loudness/energy, tempo, emphasis), and visual behavior (facial muscle activation, gesture force, posture tension). Briefly note each present cue with its polarity direction and intensity strength.

**(2) Synthesize a Graded Mental State Hypothesis (Cognitive Empathy):**
Integrate the decoded cues into a coherent here-and-now hypothesis consisting of a base polarity (positive/negative/neutral) and an intensity level (slight/moderate/strong). Indicate whether the cues converge; if some diverge, acknowledge the discrepancy but ground the hypothesis strictly in observable evidence without inventing unobserved causes.

**(3) Attribute from the Speaker's Perspective (Perspective-Taking):**
Refine the hypothesis to attribute the single most plausible immediate evaluative state with intensity from the speaker's perspective (not the viewer's reaction and not a stable trait).

**(4) Conclude and Map to Label:**
Provide your reasoning following the steps above, then choose exactly one label from: Strong Negative, Moderate Negative, Slight Negative, Neutral, Slight Positive, Moderate Positive, Strong Positive, and output exactly {'emotion': 'label'}.

Figure 30: **ToM-style prompting for Sentiment Intensity Analysis.**

**8) Song Emotion Recognition**

*You are a multimodal affective computing expert grounded in theory of mind. Your task is to infer the performance's current first-order emotional state (as enacted by the singer/performer) by reasoning about their mental world from multimodal evidence. Complete the song emotion recognition task as follows:*

**(1) Decode Affective Signals (Perceptual Simulation):**
Systematically decode observable performance cues as outward evidence of an inner state—facial expression and micro-expressions, gaze and body/breath posture and movement quality; musical-acoustic prosody if audible (tempo, intensity/energy, pitch range/contour, timbre, articulation); and lyrics/onscreen text if provided (evaluative terms, stance). Briefly note the present cues and their likely emotion implications.

**(2) Synthesize a Performance-State Hypothesis (Cognitive Empathy):**
Integrate the decoded cues into a coherent here-and-now hypothesis about what emotion the performer is expressing. Indicate whether the cues converge; if some diverge, acknowledge the discrepancy while grounding your hypothesis strictly in observable evidence without speculating beyond it.

**(3) Attribute from the Performer's Perspective (Perspective-Taking):**
Refine the hypothesis to attribute the single most plausible immediate emotion from the performer's enacted perspective (not the viewer's reaction and not a stable trait of the performer).

**(4) Conclude and Map to Label:**
Provide your reasoning following the steps above, then choose exactly one label from: neutral, calm, happy, sad, angry, fearful, and output exactly {'emotion': 'label'}.

Figure 31: **ToM-style prompting for Song Emotion Recognition.**

**9) Speech Emotion Recognition**

*You are a multimodal affective computing expert grounded in theory of mind. Your task is to infer the speaker's current first-order emotional state during speech by reasoning about their mental world from observable evidence. Complete the speech emotion recognition task as follows:*

**(1) Decode Affective Signals (Perceptual Simulation):**
Systematically decode speech-related cues as outward evidence of an inner state—acoustic-prosodic features (pitch level/range/contour, loudness/energy, tempo/rhythm, emphasis, voice quality such as breathy/tense/rough), articulatory-visual cues (mouth shaping, facial expression, gaze, head and upper-body movement, posture), and transcript content if provided. Briefly note the present cues and their likely emotion implications.

**(2) Synthesize a Mental State Hypothesis (Cognitive Empathy):**
Integrate the decoded cues into a coherent here-and-now emotion hypothesis. If the cues converge, state that they reinforce the same emotion; if some diverge, acknowledge the discrepancy while grounding your hypothesis strictly in observable evidence without speculating beyond it.

**(3) Attribute from the Speaker's Perspective (Perspective-Taking):**
Refine the hypothesis to attribute the single most plausible immediate emotion from the speaker's perspective (not the listener's reaction and not a stable trait).

**(4) Conclude and Map to Label:**
Provide your reasoning following the steps above, then choose exactly one label from: neutral, calm, happy, sad, angry, fearful, surprised, disgust, and output exactly {'emotion': 'label'}.

Figure 32: **ToM-style prompting for Speech Emotion Recognition.**

**10) Stock Comment Emotion Analysis**

*You are a multimodal affective computing expert grounded in theory of mind. Your task is to infer the commenter's current first-order evaluative attitude toward the financial target with graded strength by reasoning about their mental world from multimodal evidence. Complete the stock comment emotion analysis task as follows:*

**(1) Decode Evaluative Signals (Perceptual Simulation):**
Systematically decode observable cues as outward evidence of an inner stance—lexical content (finance-laden evaluations, polarity terms, certainty/hedging and modality markers, numeric framing, comparative/superlative wording), speech prosody if available (pitch/loudness/tempo/emphasis indicating confidence or doubt), and visual behavior if available (facial expression, nods/shakes, posture/gesture tension). Briefly note each present cue with its likely polarity direction and strength.

**(2) Synthesize a Graded Attitude Hypothesis (Cognitive Empathy):**
Integrate the decoded cues into a coherent here-and-now hypothesis consisting of a base polarity (positive/negative/neutral) and a strength level (weak/strong) toward the specific financial target implied in the comment. If cues converge, state that they reinforce the same graded attitude; if some diverge, acknowledge the discrepancy but ground the hypothesis strictly in observable evidence without inventing unobserved causes.

**(3) Attribute from the Commenter's Perspective (Perspective-Taking):**
Refine the hypothesis to attribute the single most plausible immediate evaluative attitude from the commenter's perspective toward the financial target. Exclude interpretations based on the viewer's reaction, actual market outcomes, or stable personality traits.

**(4) Conclude and Map to Label:**
Provide your reasoning following the steps above, then choose exactly one label from: weak negative, strong negative, neutral, weak positive, strong positive, and output exactly {'emotion': 'label'}.

Figure 33: **ToM-style prompting for Stock Comment Emotion Analysis.**

**1) Detection of Persuasion Techniques in Memes**

*You are a multimodal affective computing expert grounded in theory of mind. Your task is to detect persuasion techniques in the meme by modeling it as a communicative act from creator to audience. This requires a relational and contextual reasoning chain: first infer a mental state from observable cues (cues → state), then reason about how this state is directed toward entities and serves a communicative purpose (state → about(entity, context)).*

**(1) Decode Propositional & Affective Content (Perceptual Simulation):**
Systematically decode observable signals as outward evidence of a mental attitude. For the image (faces, actions, symbols, flags, crowds, threat/victim imagery) and the text (wording, slogans, emphasis such as ALL CAPS, repetition, emojis), identify key targets, in-group/out-group framing if present, and affective or value-laden terms. Note whether the channels converge or deliberately contrast.

**(2) Infer Communicative Intent toward the Audience (Cognitive Empathy & Perspective-Taking):**
Adopt the creator-as-speaker perspective and model the generic viewer as the audience. Infer what belief, attitude, or emotion the meme is designed to elicit. If text and image diverge, treat this as a potential rhetorical device (e.g., sarcasm, irony, exaggeration) rather than noise.

**(3) Attribute Persuasion Techniques (Relational Mapping):**
Map the inferred intent and its supporting cues to zero or more persuasion techniques from the given taxonomy. Link each chosen technique to the relevant cue–intent relation, but keep the reasoning concise and evidence-based. Select only techniques directly supported by the multimodal content.

**(4) Conclude and Output:**
Provide your reasoning following the steps above, then output exactly: {'techniques': ['label1','label2', ...]}.

Figure 34: **ToM-style prompting for Detection of Persuasion Techniques in Memes.**

**2) Emotion-Based Intent Analysis**

*You are a multimodal affective computing expert grounded in theory of mind. Model the utterance as a communicative act from speaker to addressee. Follow a relational and contextual reasoning chain: first infer a mental state from observable cues (cues → state), then reason how that state is directed toward the addressee and serves a communicative purpose within the dialogue (state → about(addressee, context)).*

**(1) Decode Multimodal & Dialogic Cues (Perceptual Simulation):**
Systematically decode observable signals as outward evidence of an inner state—lexical content, prosody (if available), visual behavior (if available), and immediate dialogue context. Note present cues and their emotion/intent implications.

**(2) Attribute the Speaker's Emotion (Cognitive Empathy, First-Order):**
Integrate the decoded cues to infer the speaker's here-and-now emotional state from the speaker's perspective (not the listener's reaction and not a stable trait). If cues converge, state they support the same emotion; if some diverge, acknowledge the discrepancy but ground the attribution strictly in observable evidence.

**(3) Infer Communicative Intent toward the Addressee (Perspective-Taking & Relational Mapping):**
Using dialogue context and speech-act indicators, infer the single most plausible intent the speaker directs at the addressee. Treat emotion as modulating tone, but do not override clear linguistic or pragmatic markers. Use "neutral" if no intent is indicated.

**(4) Conclude and Output:**
Provide your reasoning following the steps above, then choose exactly one emotion from: happy, surprise, sad, disgust, anger, fear, neutral, and exactly one intent from: questioning, agreeing, acknowledging, encouraging, consoling, suggesting, wishing, neutral. Output exactly:
{'emotion': 'label', 'intent': 'label'}.

Figure 35: **ToM-style prompting for Emotion-Based Intent Analysis.**

**3) Humor Understanding**

*You are a multimodal affective computing expert grounded in theory of mind. Model the utterance as a communicative act from speaker to audience. Follow a relational and contextual reasoning chain: first infer a mental state from observable cues and context (cues → state), then reason about how that state is used to amuse an audience within the dialogue (state → about(audience, context)).*

**(1) Decode Context & Delivery Cues (Perceptual Simulation):**
Systematically decode observable signals as outward evidence of an intended stance—context/setup content, the punchline sentence, and any available delivery cues (prosody, timing, facial/gestural markers). Briefly note the salient propositions and the baseline expectation established by the setup.

**(2) Construct an Audience Expectation Model (Cognitive Empathy):**
From the setup, infer what a reasonable audience would expect next. Specify the assumed belief/interpretation the audience holds before hearing the punchline (the contextual baseline).

**(3) Detect Incongruity and Seek Coherent Reinterpretation (Perspective-Taking & Intent Inference):**
Compare the punchline to the baseline. Determine whether there is an intentional mismatch (e.g., reversal, double meaning, absurd shift) that invites a coherent reinterpretation resolving the surprise in a benign/acceptable way. Ground the inference strictly in the provided content; do not import external facts.

**(4) Attribute Communicative Intent (Relational Mapping):**
Decide whether the speaker is using that incongruity/resolution to amuse the audience here-and-now (i.e., humor as the communicative purpose). If no plausible humorous reinterpretation emerges from the given content, conclude it is not humor.

**(5) Conclude and Output:**
Provide your reasoning following the steps above, then choose exactly one label from: true, false, and output the chosen label only.

Figure 36: **ToM-style prompting for Humor Understanding.**

**4) Implicit Attribute Value Extraction**

*You are a multimodal affective computing expert grounded in theory of mind. Your task is to infer an unstated product attribute value by reasoning about the designer's intent and the observer's interpretation. Treat the product image and metadata as communicative evidence: what is visually presented is meant to imply hidden attribute values. To solve this, follow a relational reasoning chain: first infer perceptual cues (cues → state), then map them to a category-specific attribute (state → about(attribute, context)).*

**(1) Decode Product Cues (Perceptual Simulation):**
Systematically analyze observable details in the product image—such as shape, cut, seams, length, neckline, sleeve, strap, collar, shoulder, shaft height, heel, toe style, or other relevant features—and combine them with the provided metadata (category and attribute type). Treat these as outward signals of the product's design properties.

**(2) Attribute Implied Design Property (Cognitive Empathy):**
Infer the most plausible here-and-now design property implied by the decoded cues, simulating the intent behind how the product is visually presented. Ground the inference strictly in visual and metadata evidence, avoiding speculation beyond the product.

**(3) Relational Mapping to Attribute Taxonomy (Perspective-Taking):**
Map the inferred design property to exactly one valid value within the provided taxonomy for that category and attribute. Ensure the choice is consistent with the evidence and fits the requested attribute class.

**(4) Conclude and Output:**
Provide your reasoning following the steps above, then choose exactly one attribute value from the given list of options for the target attribute. Output only the chosen label.

Figure 37: **ToM-style prompting for Implicit Attribute Value Extraction.**

## 5) Multimodal Aspect-Based Sentiment Analysis

*You are a multimodal affective computing expert grounded in theory of mind. Model the post as a communicative act from author to audience. Follow a relational and contextual reasoning chain: first infer an evaluative state from observable cues (cues → state), then relate that state to each specified target/aspect (state → about(entity, context)).*

**(1) Decode Multimodal & Referential Cues (Perceptual Simulation):**
Systematically decode observable signals as outward evidence of an inner evaluative stance—textual content (opinion predicates, polarity words, intensifiers/negations, stance verbs), image cues (faces, actions, symbols, scenes), and prosody if available. Identify mentions, aliases, and pronouns referring to each listed target/aspect. Note whether text and image converge or deliberately contrast.

**(2) Build Target-Level Attitude Hypotheses (Cognitive Empathy):**
For each specified target/aspect, integrate only the cues that pertain to it (direct mentions, co-references, or clearly implied links) to form a here-and-now stance hypothesis. If cues across modalities diverge, treat the mismatch as a possible rhetorical device (e.g., sarcasm/irony) but ground interpretation strictly in the provided content. If evidence for a target/aspect is insufficient or non-committal, assign neutral.

**(3) Attribute from the Author's Perspective (Perspective-Taking & Relational Mapping):**
Attribute the single most plausible immediate polarity the author holds toward each target/aspect (not the viewer's reaction, and not a stable trait). Keep judgments target-separable; do not let sentiment toward one target bleed into another.

**(4) Conclude and Output:**
Provide your reasoning following the steps above, then for every target output exactly one label from: positive, neutral, negative. Return exactly one JSON object mapping each target to its label, e.g. {'TargetA': 'positive', 'TargetB': 'neutral'}.

Figure 38: **ToM-style prompting for Multimodal Aspect-Based Sentiment Analysis.**

## 6) Multimodal Quintuple Extraction

*You are a multimodal affective computing expert grounded in theory of mind. Model each utterance/post as a communicative act from a specific holder toward targets in context. Follow a relational and contextual reasoning chain: first infer evaluative states from textual cues (cues → state), then bind each state to a concrete target and aspect within the discourse or scene (state → about(entity, context)) and express it as structured quintuples.*

**(1) Decode Propositional & Referencing Cues (Perceptual Simulation):**
Systematically parse textual content for affective/evaluative markers—opinion predicates, polarity words, intensifiers/negations, hedges/modality—and resolve speaker identity (holder). Perform basic coreference/alias resolution to link pronouns or aliases to explicit targets. Keep strictly to the given text; do not import external facts.

**(2) Attribute Target-Level Attitudes (Cognitive Empathy, First-Order):**
For each holder, infer the here-and-now evaluative state toward a specific target and aspect indicated or implied by the text. If cues diverge (e.g., positive term with negative intensifier), acknowledge and ground the inference in the most coherent observable reading. If stance is weak/uncertain, treat sentiment as neutral.

**(3) Relational Mapping to Structured Quintuples (Perspective-Taking & Context Binding):**
Compose quintuples of the form (holder, target, aspect, opinion, sentiment). holder: the speaking author of the utterance; target: the entity being evaluated (brand/place/person/item/topic); aspect: the facet/property of the target (explicit noun phrase; allow implicit facet only if clearly signaled); opinion: the minimal surface phrase expressing the evaluation; sentiment: one of {positive, neutral, negative}, consistent with cues. Create separate quintuples for different targets/aspects.

**(4) Conclude and Output:**
Provide your reasoning following the steps above, then output a single Python-style list of quintuples using single quotes and no extra commentary, e.g.:[('holder','target','aspect','opinion','sentiment'), ...].

Figure 39: **ToM-style prompting for Multimodal Quintuple Extraction.**

## 7) Multimodal Stance Detection

*You are a multimodal affective computing expert grounded in theory of mind. Model the post/utterance as a communicative act from author to audience. Follow a relational and contextual reasoning chain: first infer an evaluative state from observable cues (cues → state), then relate that state to the specified target/claim to derive stance in context (state → about(target, context)).*

**(1) Decode Multimodal & Referential Cues (Perceptual Simulation):**
Systematically decode textual content (opinion/stance markers, polarity, negation/modality), image cues (faces, actions, symbols, scenes), and prosody if available, as outward evidence of an inner attitude. Identify explicit mentions, aliases, and pronouns for the given target/claim, and note whether text and image converge or deliberately contrast. Do not import external facts.

**(2) Attribute the Author's Attitude toward the Target/Claim (Cognitive Empathy, First-Order):**
Integrate only the cues that pertain to the specified target/claim to form a here-and-now attitude hypothesis from the author's perspective. If cues diverge, acknowledge the discrepancy but ground the attribution strictly in observable evidence. If evidence is non-committal, keep the hypothesis neutral.

**(3) Map Attitude to Stance (Perspective-Taking & Relational Mapping): Derive exactly one stance label conditioned on the target/claim:**
• support — expresses endorsement/alignment with the target/claim.
• refute — expresses opposition/contradiction to the target/claim.
• comment — discusses the topic/context without clear support or refutation.
• unrelated — no meaningful relation to the specified target/claim.
Keep judgments target-conditioned; do not let general sentiment toward other entities bleed into this mapping.

**(4) Conclude and Output:**
Provide your reasoning following the steps above, then choose exactly one label from: support, refute, comment, unrelated. Output exactly: {'stance': 'label'}.

Figure 40: **ToM-style prompting for Multimodal Stance Detection.**

## 8) Multiparty Dialogue Emotion Recognition

*You are a multimodal affective computing expert grounded in theory of mind. Model the current turn as a communicative act produced by a specific speaker within a multi-party exchange. Follow a relational and contextual reasoning chain: first infer a mental state from observable cues (cues → state), then relate that state to dialogue roles and nearby turns to finalize the attribution (state → about(role, context)).*

**(1) Decode Multimodal & Dialogic Cues (Perceptual Simulation):**
Systematically decode observable signals as outward evidence of an inner state—lexical content/transcript (if provided), speech prosody (if available), and visual behavior (face/gaze/posture/gesture if available). Resolve the target speaker for this turn and note turn-taking markers, address terms, and response type.

**(2) Attribute the Speaker's Emotion (Cognitive Empathy, First-Order):**
Integrate the decoded cues to infer the speaker's here-and-now emotional state from the speaker's perspective (not the listener's reaction and not a stable trait). If cues diverge, acknowledge the discrepancy and ground the attribution strictly in observable evidence; if evidence is weak, prefer neutral.

**(3) Contextualize with Roles & Surrounding Turns (Perspective-Taking & Relational Mapping):**
Use immediate context (preceding/following turns and participant roles such as addressee/third party) to refine or disambiguate the attribution—e.g., whether the turn is a reply, challenge, tease, agreement, or correction. Let context modulate, but not override, clear in-turn affective signals. Do not import external facts.

**(4) Conclude and Map to Label:**
Provide your reasoning following the steps above, then choose exactly one label from: neutral, surprise, fear, sadness, joy, disgust, anger. Output exactly: {'emotion': 'label'}.

Figure 41: **ToM-style prompting for Multiparty Dialogue Emotion Recognition.**

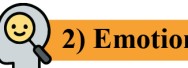

## 1) Emotion Elicitation Reasoning

*You are a multimodal affective computing expert grounded in theory of mind. Your task is to infer the emotion elicited in a typical viewer by this clip, using causal and second-order reasoning (modeling the viewer's appraisals rather than the characters' feelings). Proceed as follows:*

**(1) Decode Narrative & Affective Cues (Perceptual Simulation):**
Systematically extract salient events/outcomes, character expressions and behaviors, spoken tone/prosody (if audible), and cinematic signals (music, pacing, camera focus, lighting) as outward evidence of intended affect.

**(2) Construct a Viewer Appraisal Model (Second-Order Perspective-Taking):**
From the perspective of a generic adult viewer, appraise the depicted events along core dimensions: goal congruence/obstruction, threat/safety, agency/blame, norm or purity violation, attachment/care or loss. Keep this distinct from the characters' internal states.

**(3) Infer the Elicited Emotion (Causal Attribution):**
Map the dominant viewer appraisals to one here-and-now elicited emotion for the viewer. Resolve any cue conflicts by prioritizing the most consequential event-appraisal pattern for the viewer, not character mood.

**(4) Identify Proximate Causes & Modulators:**
Name the specific scene events/dynamics and any cinematic devices that most directly produce or intensify the elicited emotion you chose.

Figure 42: **ToM-style prompting for Emotion Elicitation Reasoning.**

## 2) Emotion Interpretation

*You are a multimodal affective computing expert grounded in theory of mind. Given a stated target emotion to explain, attribute plausible causes for the focal person(s) using causal and second-order reasoning. Proceed as follows:*

**(1) Decode Scene & Affective Cues (Perceptual Simulation):**
Systematically parse observable evidence—facial expression, gaze, body posture/movement, interpersonal spacing or touch, salient objects/events, setting cues, and any on-screen text. Treat these as outward evidence of internal appraisals.

**(2) Build a Subject-Centered Appraisal Hypothesis (Cognitive Empathy & Perspective-Taking):**
From the subject's perspective (not the viewer's), infer appraisals that fit the given emotion along core dimensions such as goal congruence/obstruction or loss, threat/safety, agency/blame, social evaluation, norm/purity violation, control/uncertainty. Keep this distinct from other characters' feelings.

**(3) Causal Attribution (Event → Appraisal → Emotion):**
Identify concrete, proximate causes in the scene—events, agents, objects, or contextual conditions—that would produce those appraisals and thus the target emotion. Prefer immediately visible causes; allow only minimal inferences directly suggested by the evidence. Do not import external facts.

**(4) Conclude and Output:**
Provide your reasoning following the steps above, then output a concise numbered list (1., 2., …) of plausible causes, phrased as short clauses grounded in the scene, with no extra commentary.

Figure 43: **ToM-style prompting for Emotion Interpretation.**

## 3) Laughter Reasoning

*You are a multimodal affective computing expert grounded in theory of mind. Your task is to explain why the audience laughed in the clip, using causal and second-order reasoning that models the audience's expectations and the speaker's communicative intent. Proceed as follows:*

**(1) Decode Setup & Delivery Cues (Perceptual Simulation):**
Systematically parse the setup content and the punchline region together with delivery cues—lexical propositions, prosody/timing (pauses, pitch/energy shifts), facial/gestural markers, and any on-screen audience reactions. Summarize the baseline interpretation established by the setup.

**(2) Model Audience Expectations (Second-Order Perspective-Taking):**
From a generic audience perspective, state what they are led to expect immediately before the trigger (the belief/interpretation they momentarily hold based on the setup and delivery).

**(3) Identify the Humor Trigger & Mechanism (Causal Attribution):**
Locate the specific element that violates or reinterprets the baseline (e.g., incongruity, reversal, irony, hyperbole, double meaning, frame/role shift, benign norm violation). Ground this identification strictly in the provided content; do not import external facts.

**(4) Explain the Causal Link & Intent (Relational Mapping):**
Explain how this trigger causes a quick reinterpretation that resolves surprise into amusement and signals playful intent from the speaker to the audience. Note any delivery features (timing/emphasis) that amplify the effect.

**(5) Conclude and Output:**
Provide your reasoning following the steps above, then output a single sentence starting with: The audience laughed because ...

Figure 44: **ToM-style prompting for Laughter Reasoning.**

## 4) Multimodal Emotion Cause Pair Extraction

*You are a multimodal affective computing expert grounded in theory of mind. For EACH target_utterance_id, infer the speaker's emotion and identify the single most plausible prior utterance that causes it, using causal and second-order reasoning. Proceed as follows:*

**(1) Decode the Target Turn (Perceptual Simulation):**
Systematically parse the target utterance's observable cues—lexical content, prosody if available (pitch/energy/tempo/emphasis), and visual behavior if available (face/gaze/posture/gesture)—as outward evidence of an inner state. Note the speaker, addressee, and turn function.

**(2) Build a Subject-Centered Appraisal Hypothesis (Cognitive Empathy & Perspective-Taking):**
From the target speaker's perspective, infer the appraisals that can explain the target emotion (e.g., goal obstruction/attainment, threat/safety, agency/blame, social evaluation, norm violation, loss). Keep this distinct from other participants' feelings. Ground all in the given dialogue/video; do not import external facts.

**(3) Causal Linking to a Prior Utterance (Causal Attribution & Second-Order Reasoning):**
Scan earlier turns to find the single prior utterance that most directly elicits the inferred appraisals for this target turn. Enforce temporal precedence (cause must precede effect). Prefer the most proximal utterance that is minimally sufficient to trigger the target emotion and is semantically aligned with the appraisals. If multiple candidates exist, choose the nearest one that best explains the target emotion; if evidence is weak or mixed, keep the emotion neutral but still select the most plausible prior cause supported by context.

**(4) Conclude and Output:**
Provide your reasoning following the steps above, then output exactly one JSON object mapping each target_utterance_id to {'emotion': 'label', 'cause_utterance_id': 'ID'} using single quotes. Choose emotion from: joy, sadness, anger, disgust, fear, surprise, neutral. Output nothing else.

Figure 45: **ToM-style prompting for Multimodal Emotion Cause Pair Extraction.**

## 5) Sarcasm Detection

*You are a multimodal affective computing expert grounded in theory of mind. Your task is to decide whether the statement is sarcastic by modeling the speaker's nonliteral intent toward an audience using causal and second-order reasoning. Proceed as follows:*

**(1) Decode Literal Proposition & Affective Cues (Perceptual Simulation):**
Systematically parse lexical content together with delivery cues—prosody if available (pitch/energy/tempo/emphasis) and visual behavior if available (face/gaze/gesture). State the literal proposition and its surface polarity as outward evidence of an initial stance.

**(2) Model Speaker Intent & Audience Uptake (Second-Order Perspective-Taking):**
From the speaker→audience perspective, infer whether the speaker intends the audience to recover a meaning that contrasts with the literal stance, based only on contextual evidence in the clip (preceding events, shared situational facts shown, discourse expectations). Do not import external facts.

**(3) Incongruity–Reversal Test (Causal Attribution):**
Check for a deliberate, coherent mismatch between (a) the literal words and (b) the situational appraisal signaled by context or by prosodic/visual cues. Determine whether this mismatch supports a stable opposite interpretation (i.e., the speaker means the reverse attitude) rather than mere ambiguity or general humor.

**(4) Decision Rule:**
If the evidence indicates an intended opposite attitude that a reasonable audience would recognize (nonliteral reversal with supportive contextual/delivery cues), label as sarcastic = true; otherwise label false.

**(5) Conclude and Output:**
Provide your reasoning following the steps above, then output exactly one token: true or false.

Figure 46: **ToM-style prompting for Sarcasm Detection.**

## 6) Sentiment Flip Analysis

*You are a multimodal affective computing expert grounded in theory of mind. Your task is to detect ALL sentiment flips in the dialogue and attribute a trigger type for each flip, using causal and second-order reasoning. Proceed as follows:*

**(1) Decode Turns & Textual Cues (Perceptual Simulation):**
Systematically parse each utterance's textual signals—lexical polarity words, negations, intensifiers, hedges, modality markers, discourse markers—as outward evidence of an inner evaluative stance. Record the holder (speaker) for every turn and note its relation to the ongoing dialogue.

**(2) Track Holder Sentiment Timeline (Cognitive Empathy, First-Order):**
For each holder, infer the here-and-now sentiment for each turn as one of {positive, negative, neutral}. Establish a baseline (the earliest clearly expressed stance) and update the timeline across turns. If cues are ambiguous, assign neutral.

**(3) Detect Flips & Attribute Triggers (Causal Attribution & Second-Order Reasoning):**
Scan each holder's timeline to detect changes of sentiment category. For each flip, attribute the most plausible trigger type based ONLY on the dialogue context, enforcing temporal precedence: Introduction of New Information, Logical Argumentation, Participant Feedback and Interaction, Personal Experience and Self-reflection.

**(4) Conclude and Output:**
Provide your reasoning following the steps above, then output ONLY a single JSON-like list where each item describes one flip: [{'holder': '...', 'initial_sentiment': 'positive|negative|neutral', 'flipped_sentiment': 'positive|negative|neutral', 'trigger_type': 'Introduction of New Information|Logical Argumentation|Participant Feedback and Interaction|Personal Experience and Self-reflection'}]. Output nothing else.

Figure 47: **ToM-style prompting for Sentiment Flip Analysis.**

## G  DATASET CASES

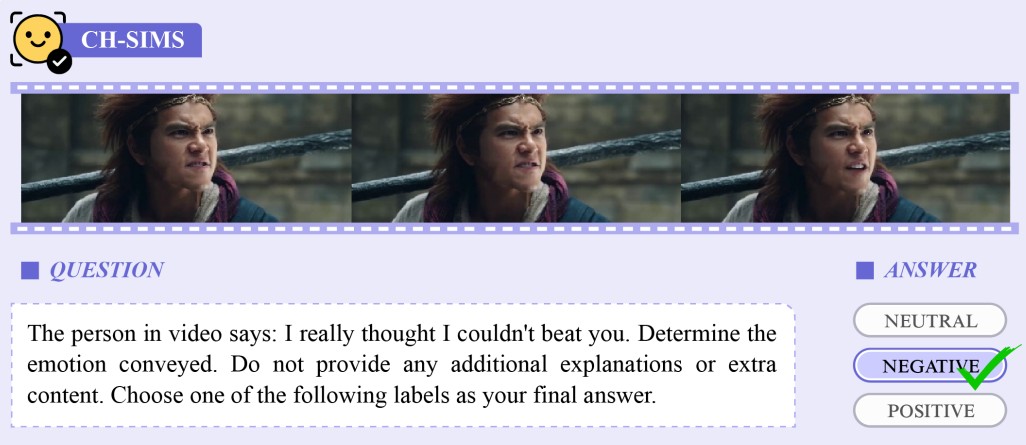

Figure 48: **Representative sample of CH-SIMS dataset.**

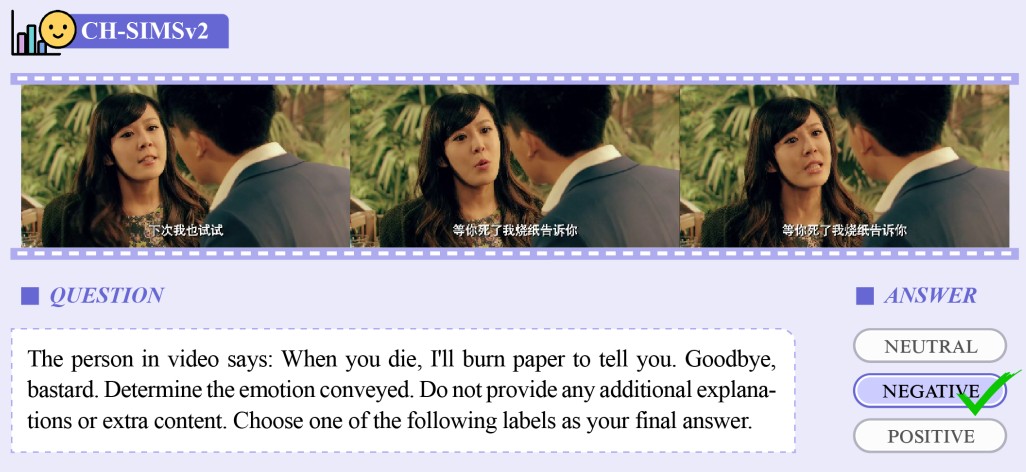

Figure 49: **Representative sample of CH-SIMSv2 dataset.**

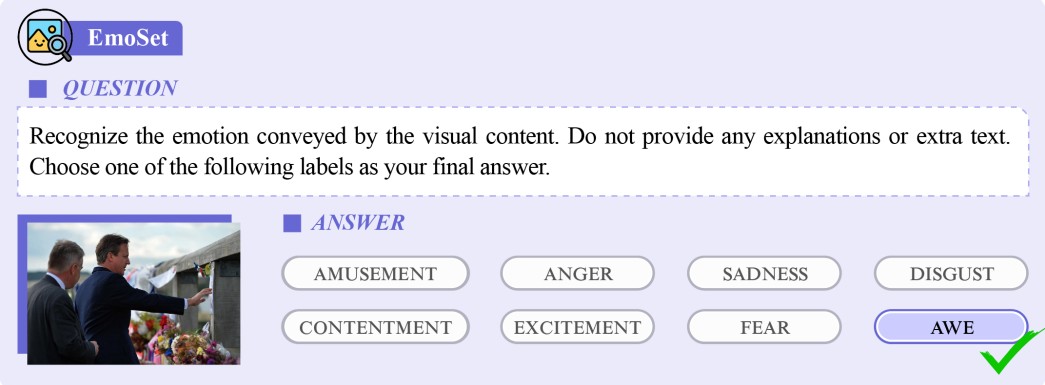

Figure 50: **Representative sample of EmoSet dataset.**

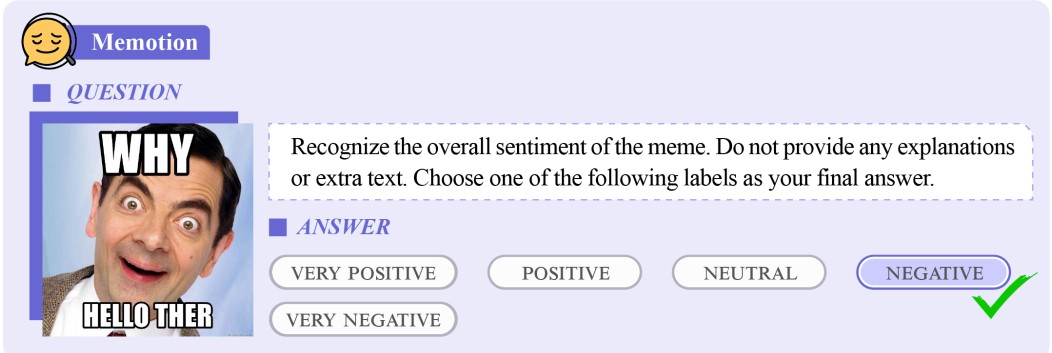

Figure 51: **Representative sample of Memotion dataset.**

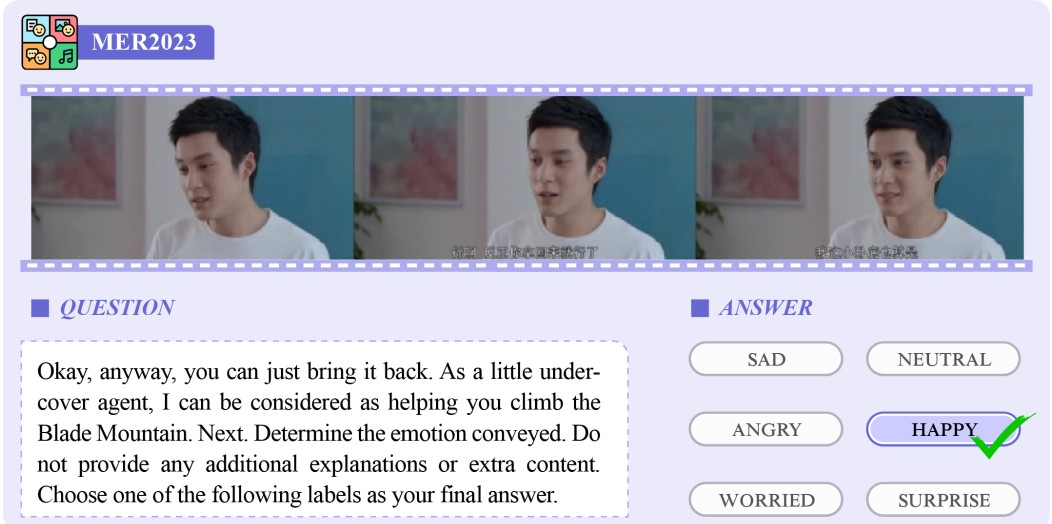

Figure 52: **Representative sample of MER2023 dataset.**

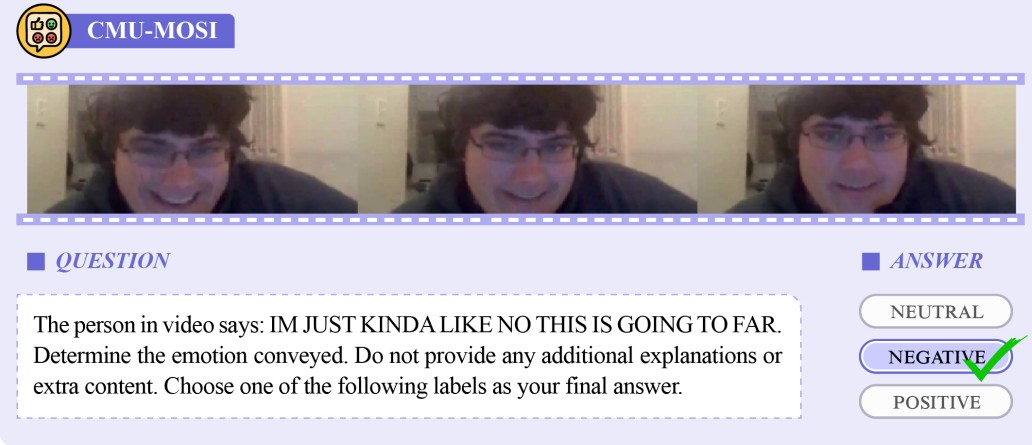

Figure 53: **Representative sample of CMU-MOSI dataset.**

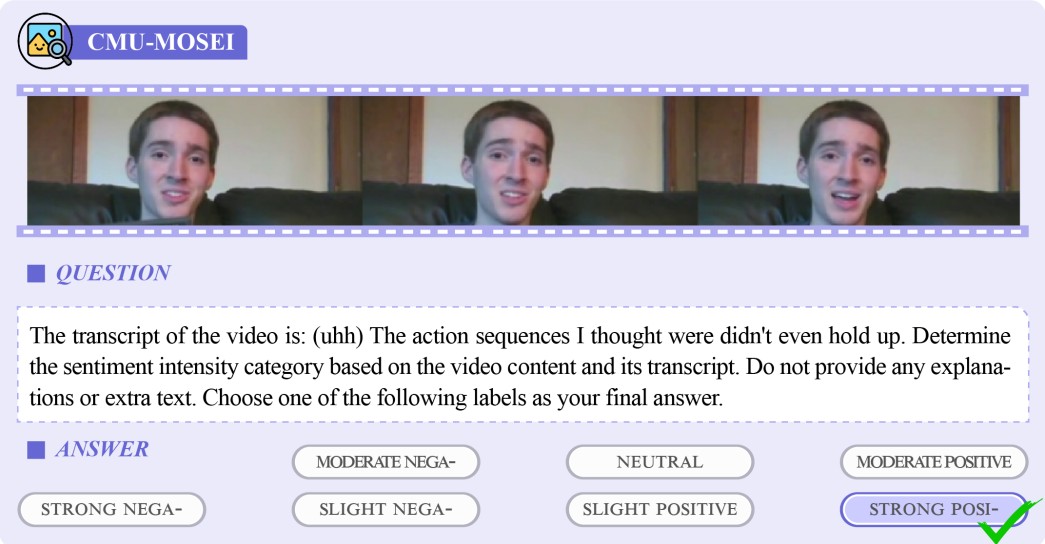

Figure 54: **Representative sample of CMU-MOSEI dataset.**

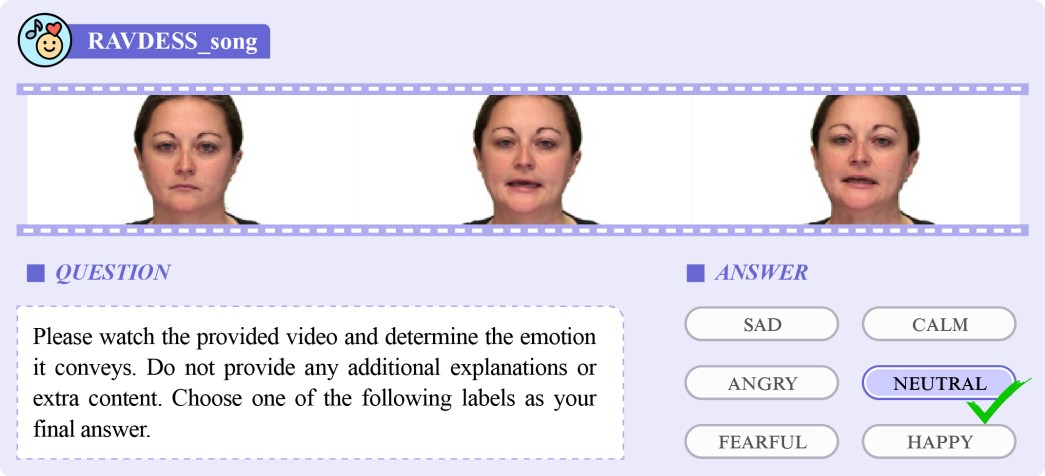

Figure 55: **Representative sample of RAVDESS (song) dataset.**

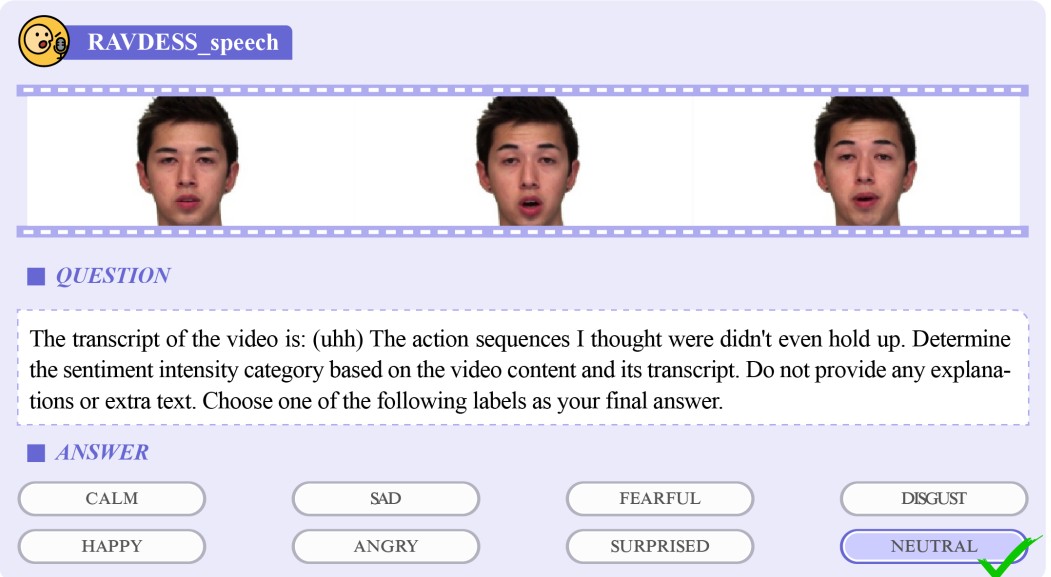

Figure 56: **Representative sample of RAVDESS (speech) dataset.**

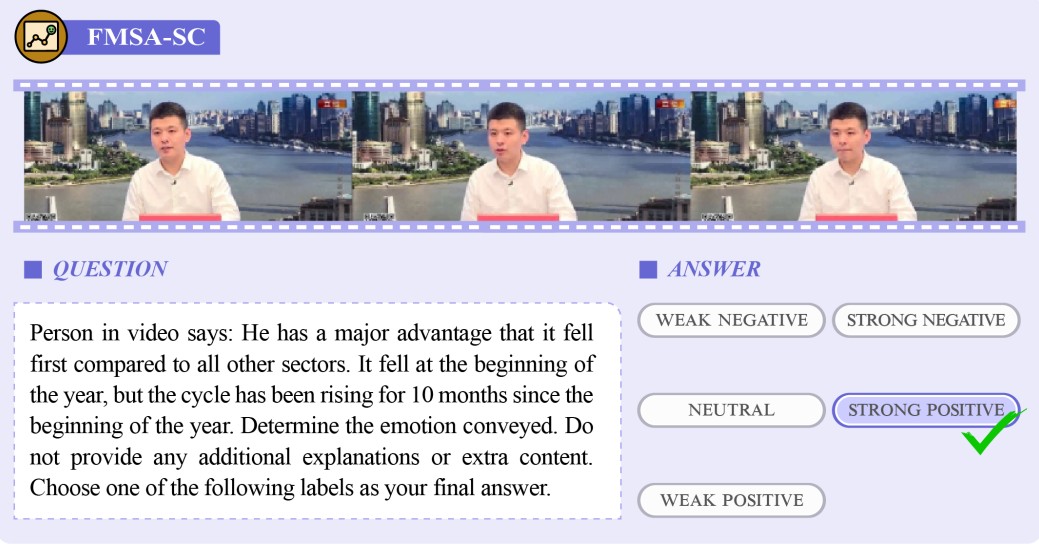

Figure 57: **Representative sample of FMSA-SC.**

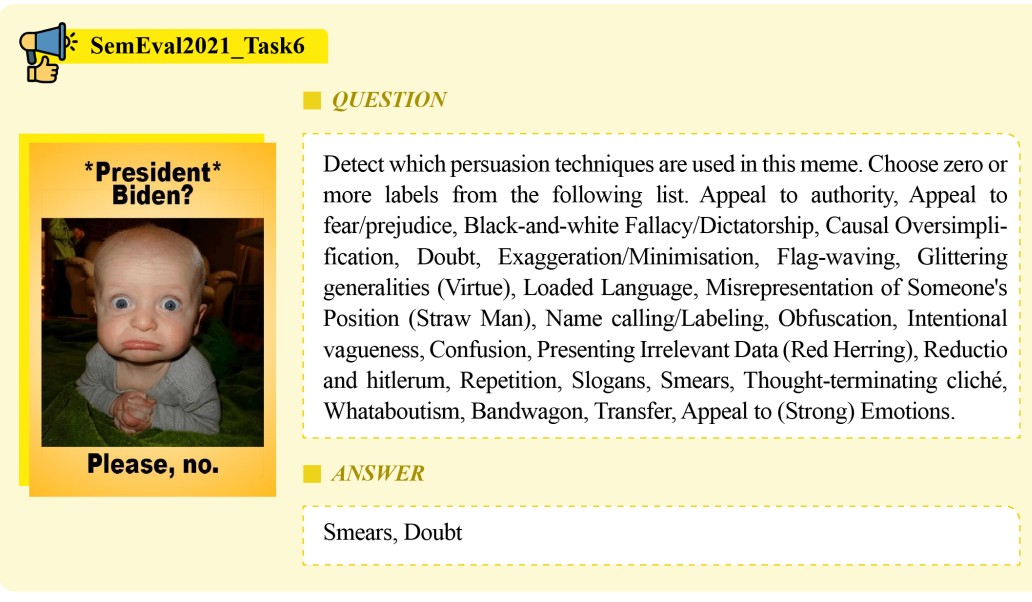

Figure 58: **Representative sample of SemEval2021_Task6 dataset.**

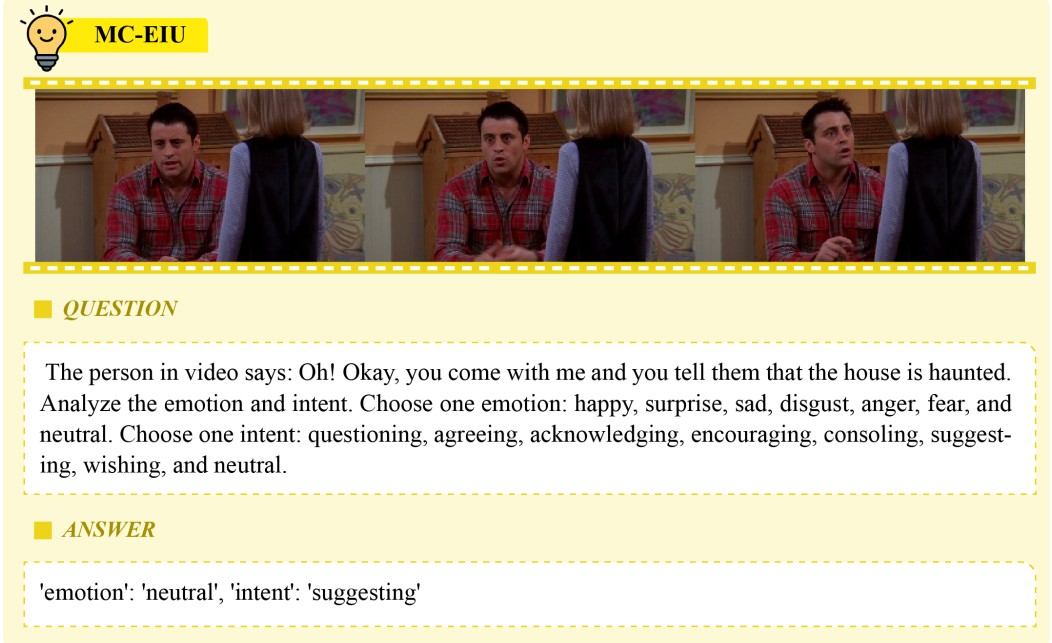

Figure 59: **Representative sample of MC-EIU dataset.**

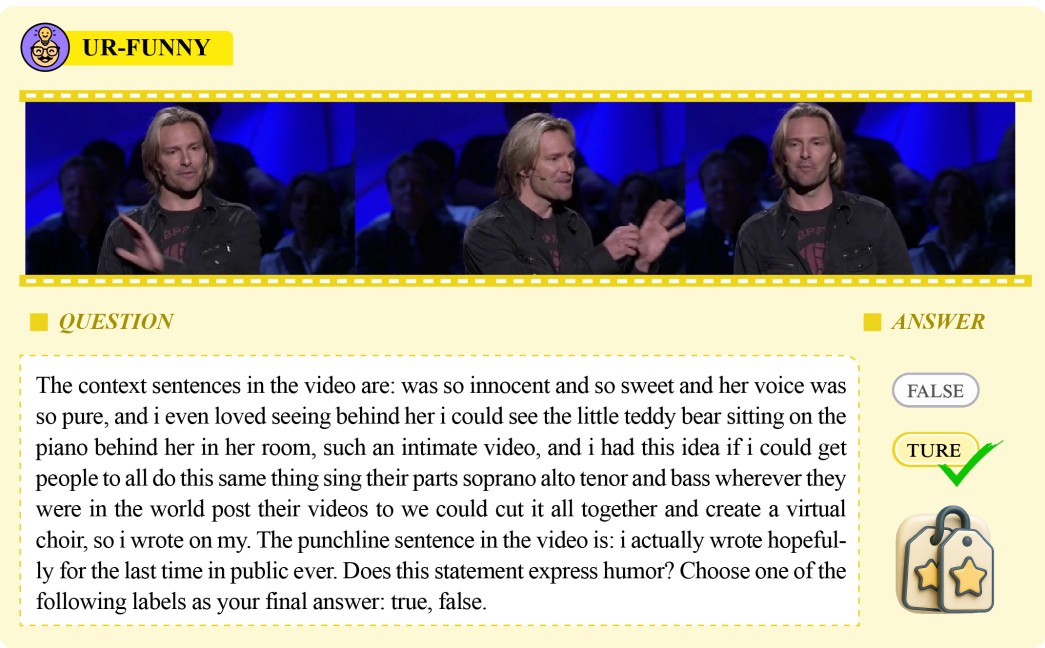

Figure 60: **Representative sample of UR-FUNNY dataset.**

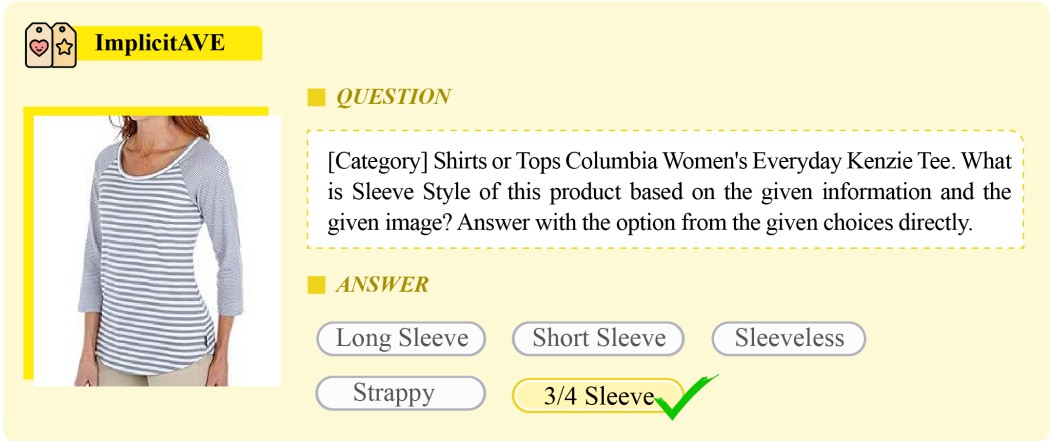

Figure 61: **Representative sample of ImplicitAVE dataset.**

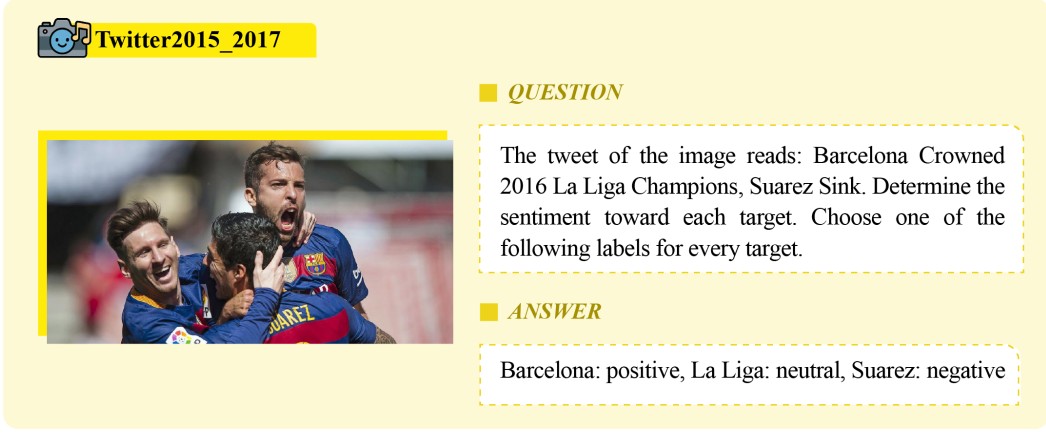

Figure 62: **Representative sample of Twitter2015/2017 dataset.**

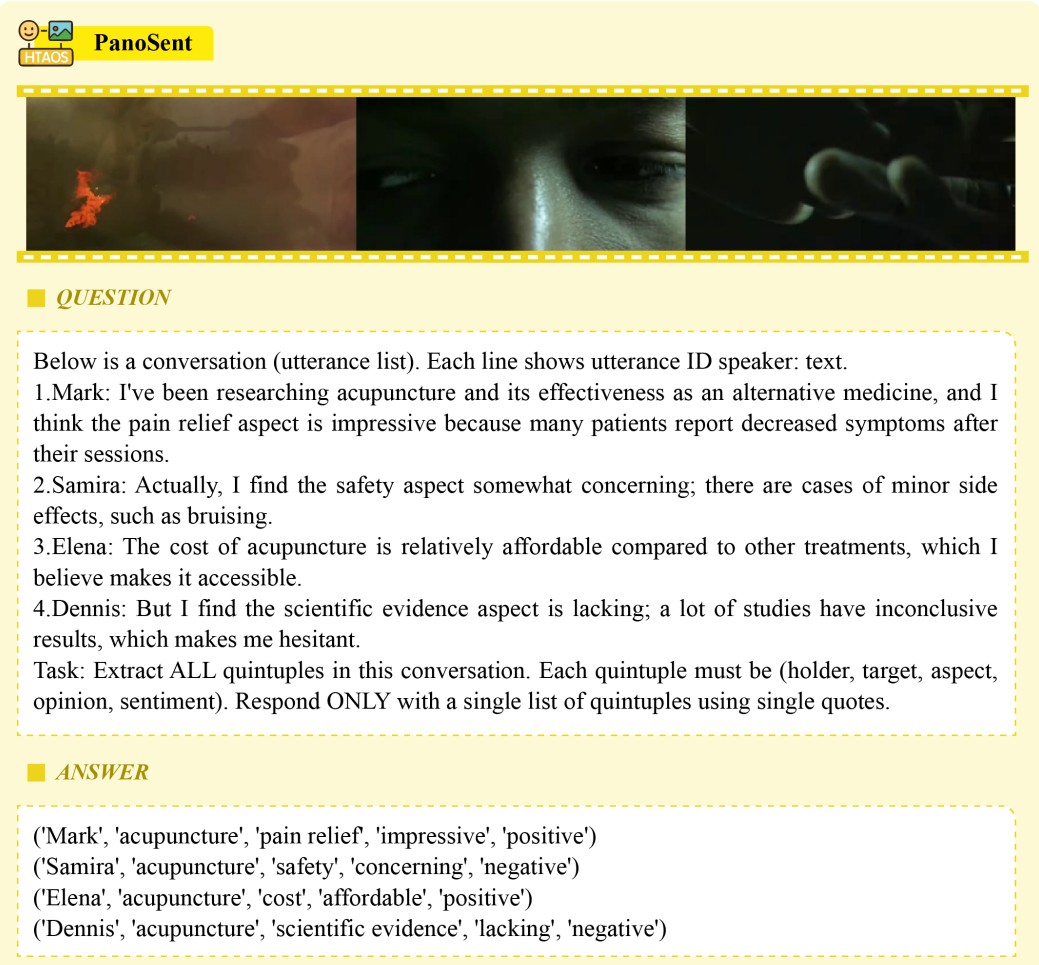

Figure 63: **Representative sample of PanoSent dataset.**

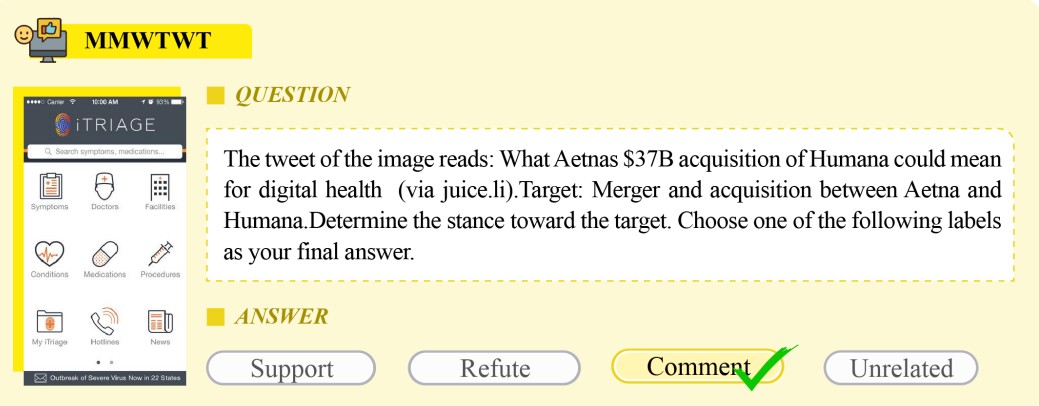

Figure 64: **Representative sample of MMWTWT dataset.**

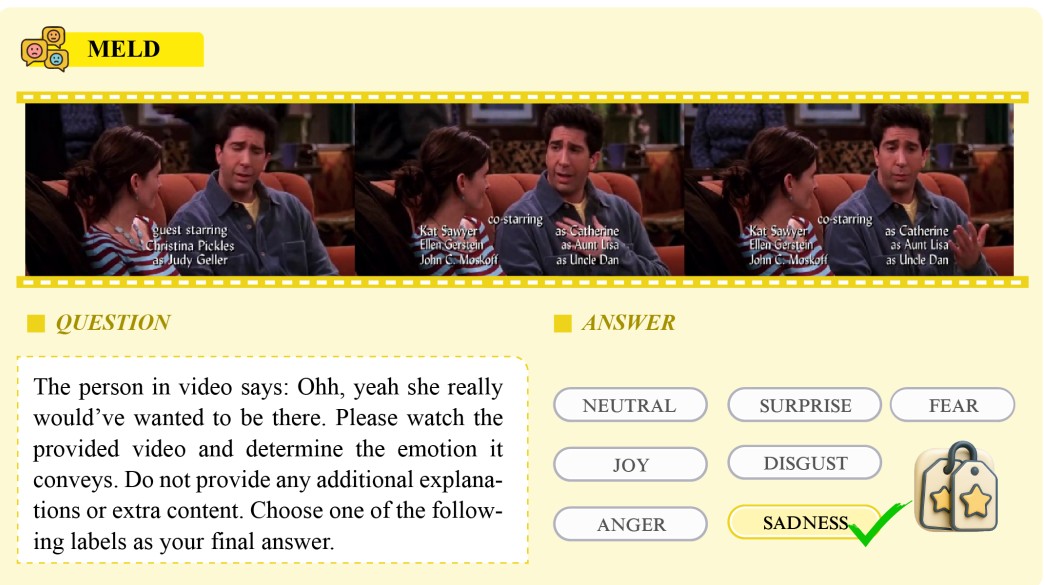

Figure 65: **Representative sample of MELD dataset.**

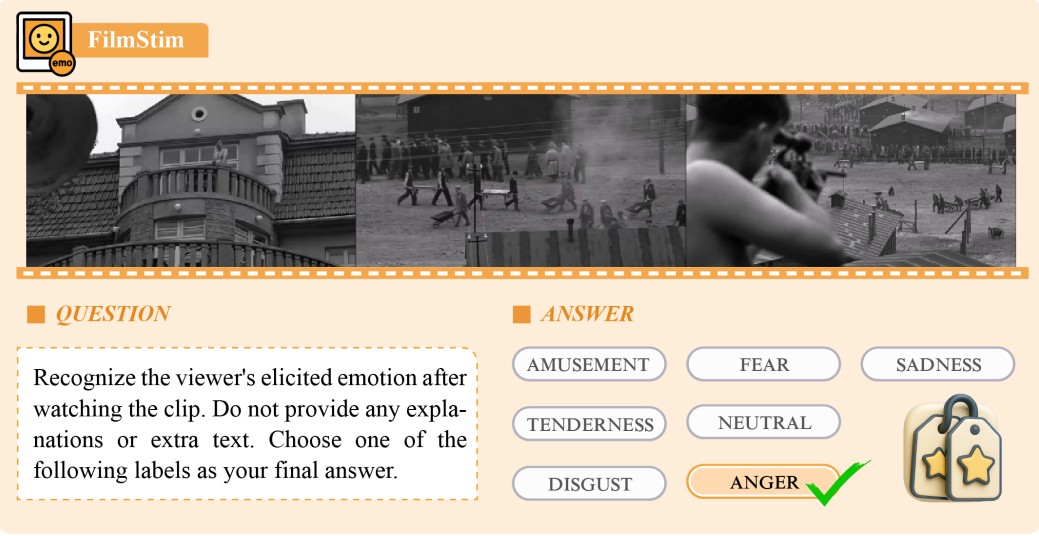

Figure 66: **Representative sample of FilmStim dataset.**

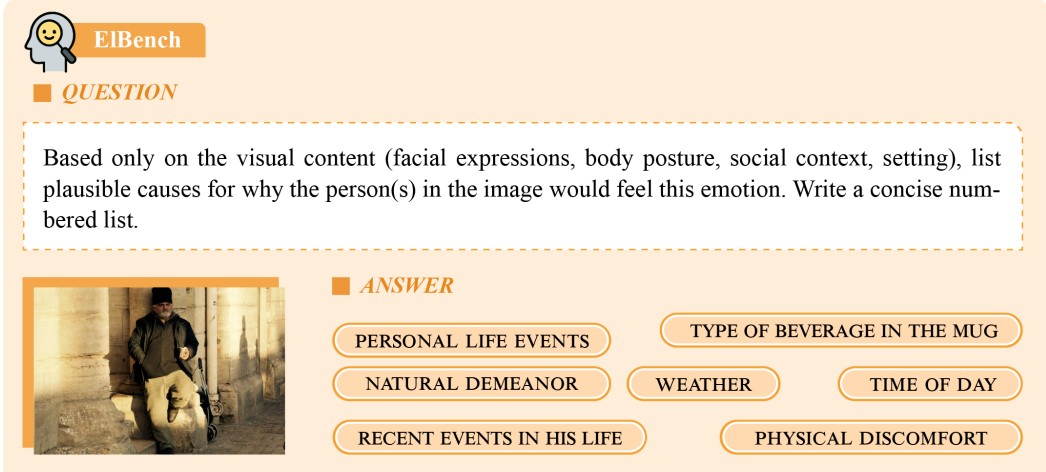

Figure 67: **Representative sample of EIBench dataset.**

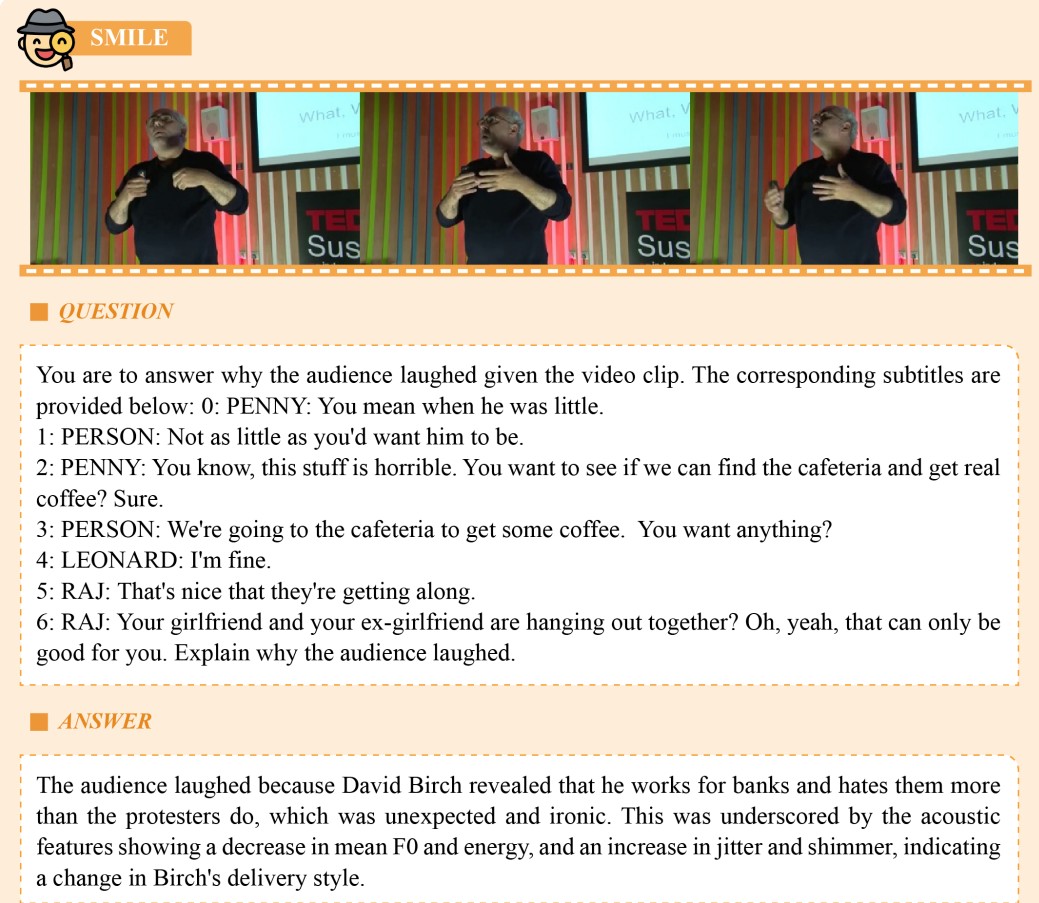

Figure 68: **Representative sample of SMILE dataset.**

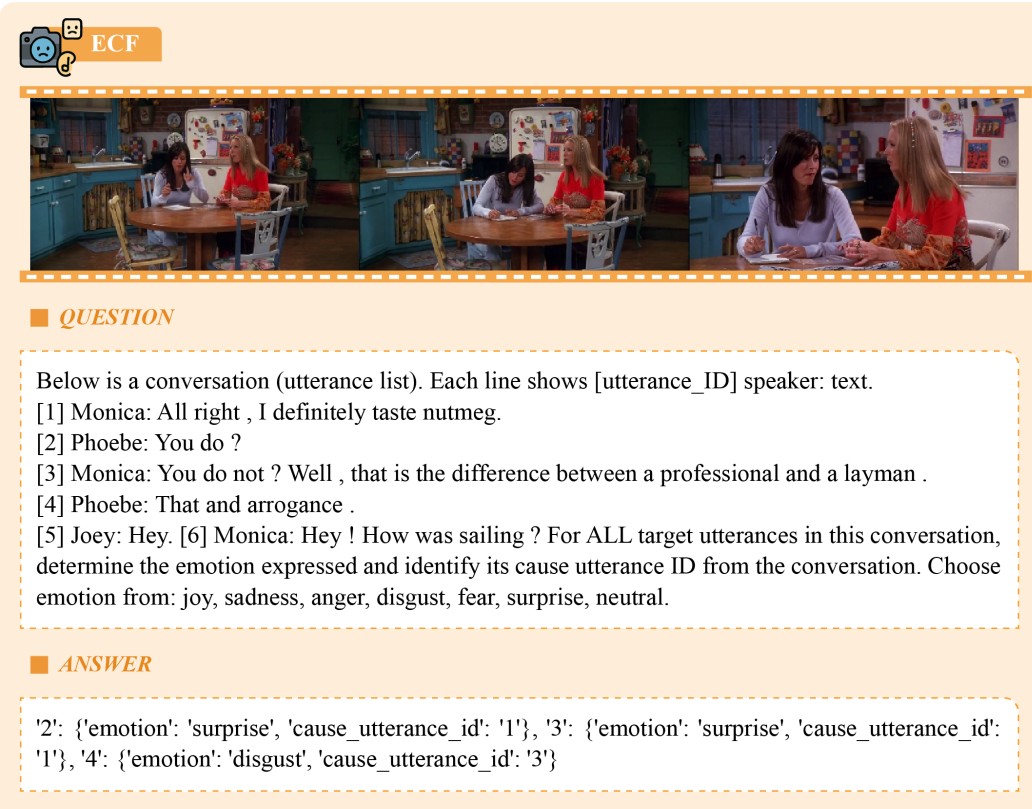

**ECF**

**QUESTION**

Below is a conversation (utterance list). Each line shows [utterance_ID] speaker: text.
[1] Monica: All right , I definitely taste nutmeg.
[2] Phoebe: You do ?
[3] Monica: You do not ? Well , that is the difference between a professional and a layman .
[4] Phoebe: That and arrogance .
[5] Joey: Hey. [6] Monica: Hey ! How was sailing ? For ALL target utterances in this conversation, determine the emotion expressed and identify its cause utterance ID from the conversation. Choose emotion from: joy, sadness, anger, disgust, fear, surprise, neutral.

**ANSWER**

'2': {'emotion': 'surprise', 'cause_utterance_id': '1'}, '3': {'emotion': 'surprise', 'cause_utterance_id': '1'}, '4': {'emotion': 'disgust', 'cause_utterance_id': '3'}

Figure 69: **Representative sample of ECF dataset.**

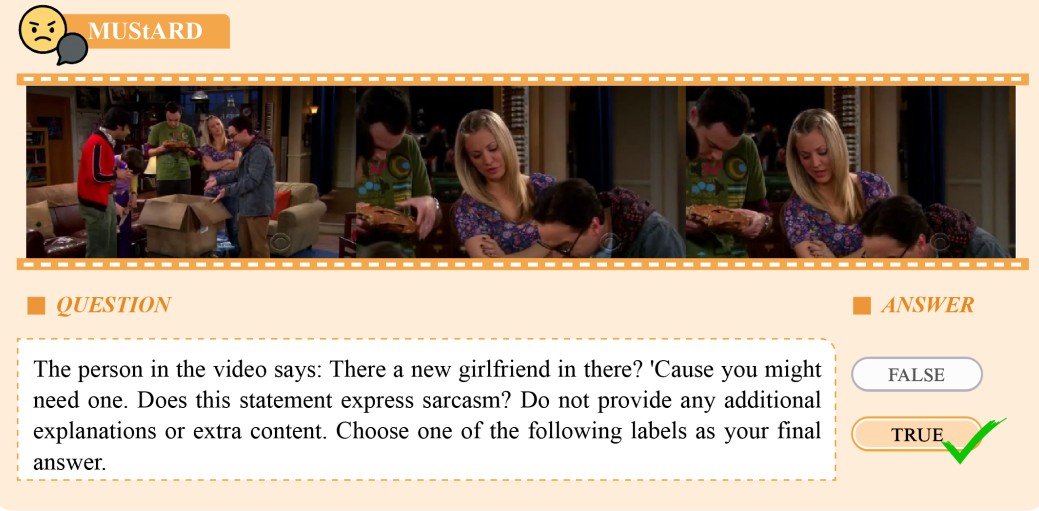

**MUStARD**

**QUESTION**

**ANSWER**

The person in the video says: There a new girlfriend in there? 'Cause you might need one. Does this statement express sarcasm? Do not provide any additional explanations or extra content. Choose one of the following labels as your final answer.

FALSE

TRUE ✔

Figure 70: **Representative sample of MUStARD dataset.**

**PanoSent**

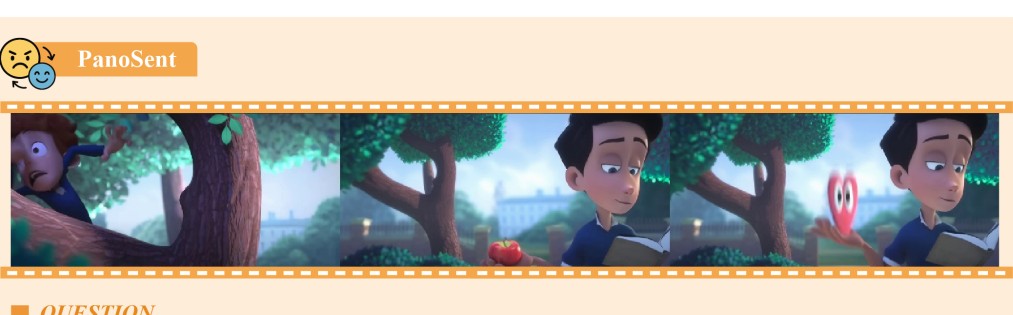

**■ *QUESTION***

Below is a conversation (utterance list). Each line shows [utterance\_ID] speaker: text.
[0] Chen: I think the color palette in 'Sunrise Valley' is vibrant; every scene feels alive because of the bold saturations.
[1] Amina: The voice acting in the show, however, sounds monotone; it misses emotional highs and lows for me.
[2] Sophia: Actually, I'd argue that the animation style is sophisticated; the subtle character movements reflect attention to detail you rarely see in other series.
[3] Marcus: That's a fair point about the animation style, Sophia. At first, I felt the animation style was plain, but given how natural the gestures appear in that dinner scene, I have to say it's actually refined.
Identify ALL sentiment flips in this conversation. For each flip, output \{'holder': 'People's names', 'initial\_sentiment': 'positive, negative, neutral', 'flipped\_sentiment': 'positive, negative, neutral', 'trigger\_type': 'one of the predefined types'\}.

**■ *ANSWER***

{'holder': 'Marcus', 'initial_sentiment': 'neutral', 'flipped_sentiment': 'positive', 'trigger_type': 'Logical Argumentation'}

Figure 71: **Representative sample of PanoSent dataset.**

## H    CASE STUDY

We present representative case studies to complement the quantitative analyses. In each case, the QA specification is fixed so that comparisons are controlled along three axes. First, Figures 72–74 compare different models under identical prompts for the same QA, revealing substantial variability in final predictions. Second, Figures 75–77 fix the model but switch the response mode between a direct answer and ToM prompting, showing how explicit reasoning reshapes intermediate justifications and can alter predicted emotions or intents. Third, Figures 78–80 return to the across-model setting, applying standardized ToM prompting for the same QA and examining step-by-step traces; despite explicit reasoning, divergences remain and some systems still err. Together, these qualitative results highlight both the strengths and the limitations of current reasoning procedures.

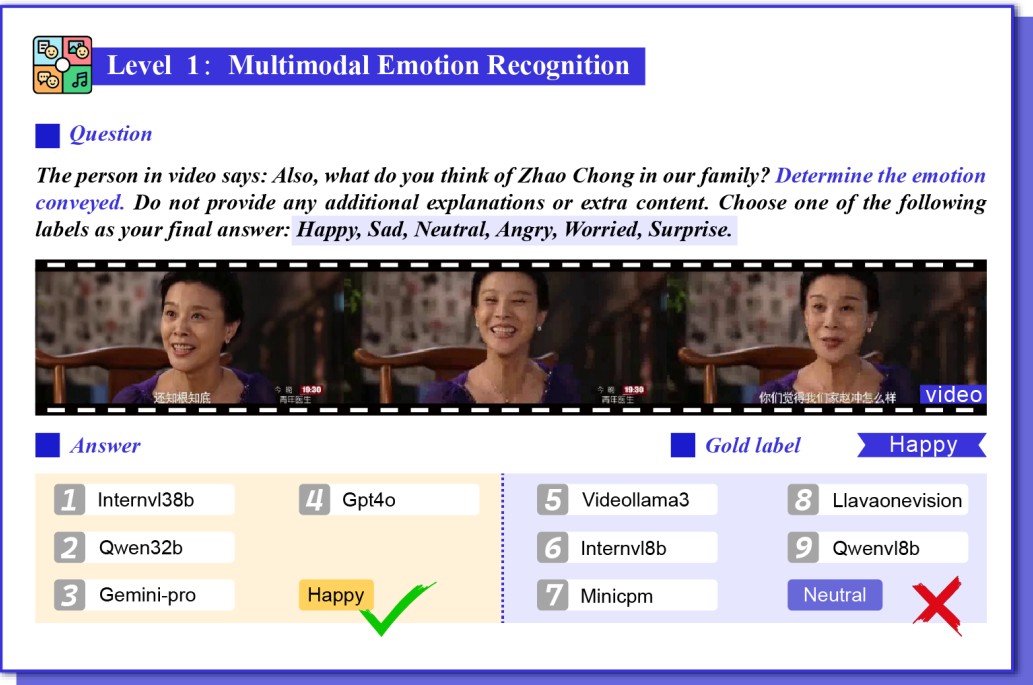

Figure 72: **Model answers on the same QA.** A side-by-side comparison of different models' predictions for the same QA, illustrating variability in responses across models.

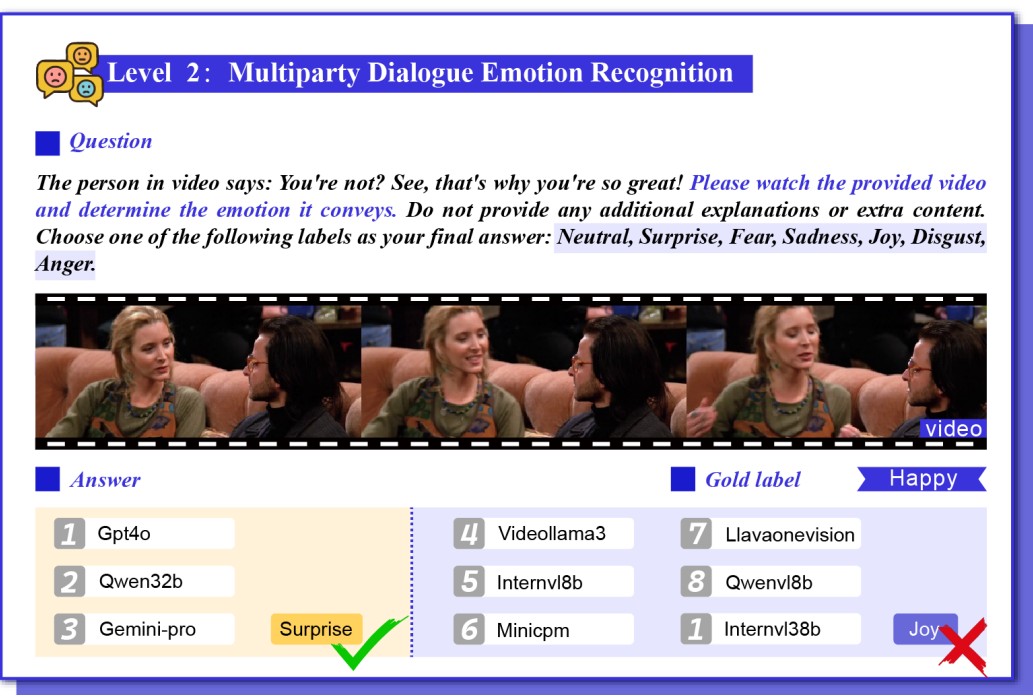

Figure 73: **Model answers on the same QA.**

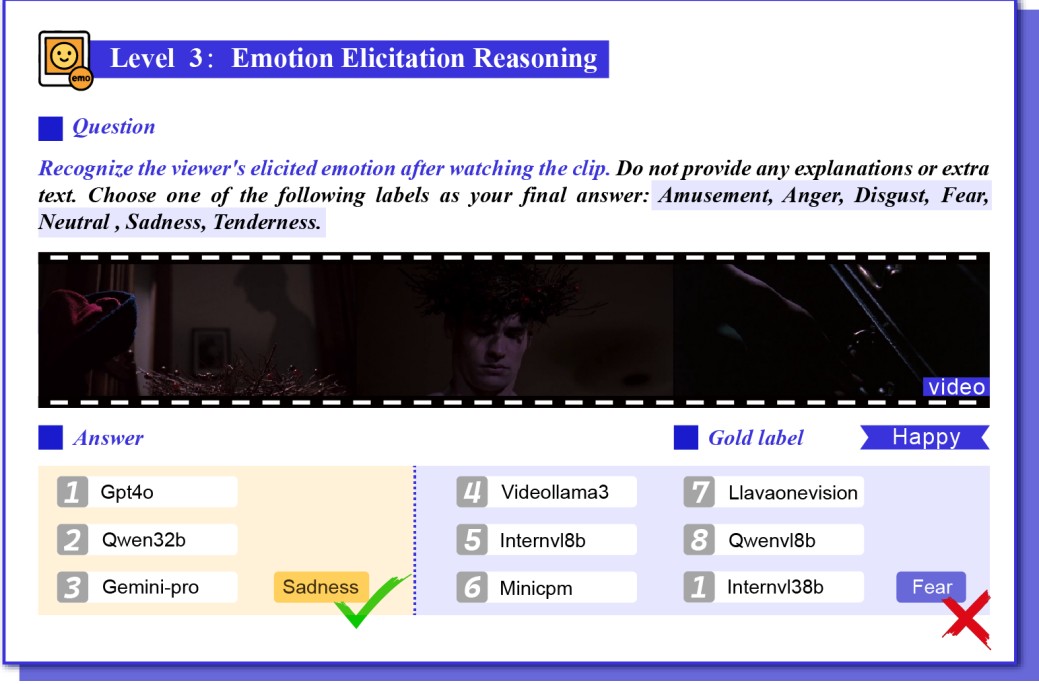

Figure 74: **Model answers on the same QA.**

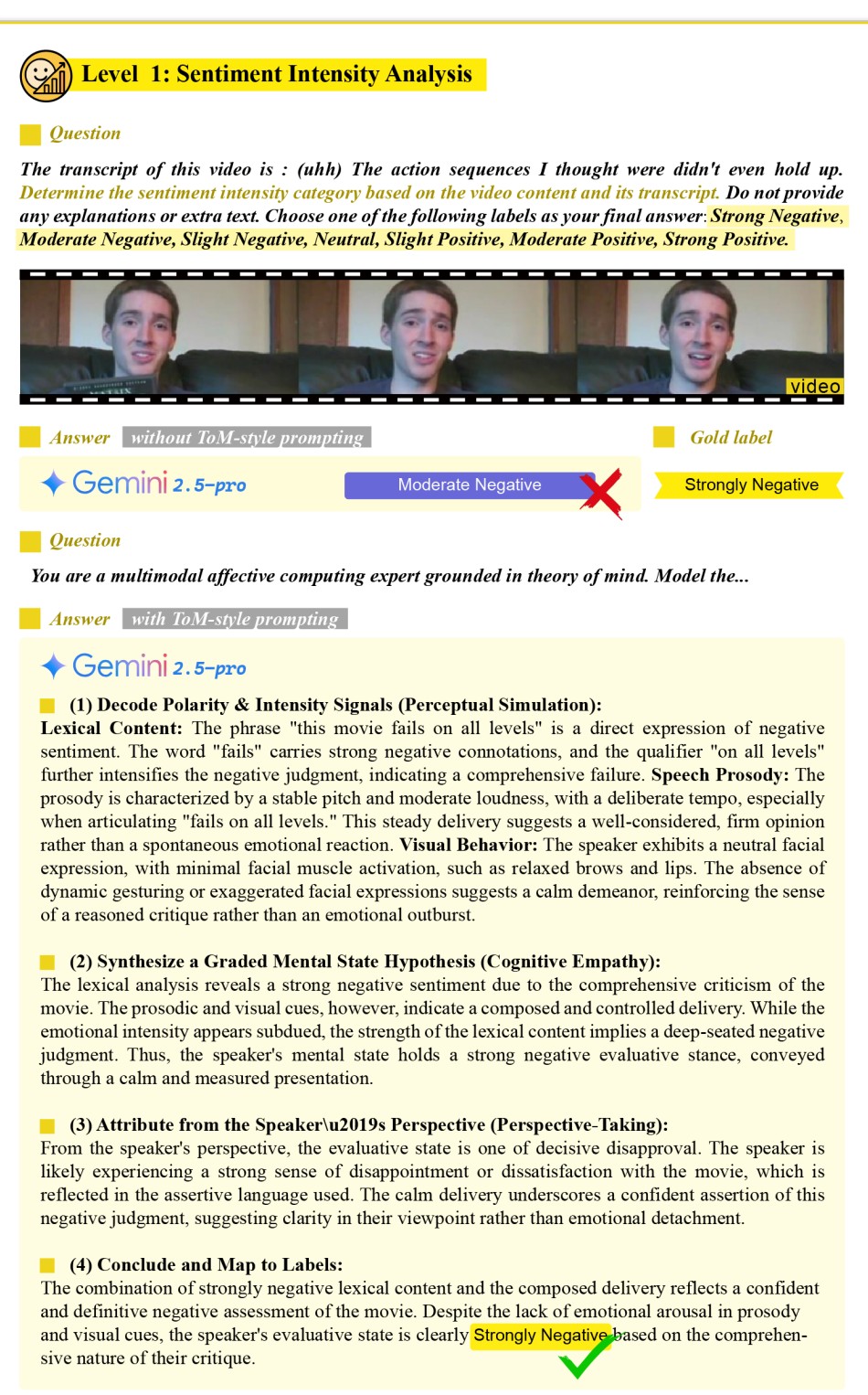

Figure 75: **Same model, direct answer vs. ToM prompting.** For the same QA, we compare one model's output when answering directly and when using our ToM prompting, illustrating how explicit reasoning changes the predicted label.

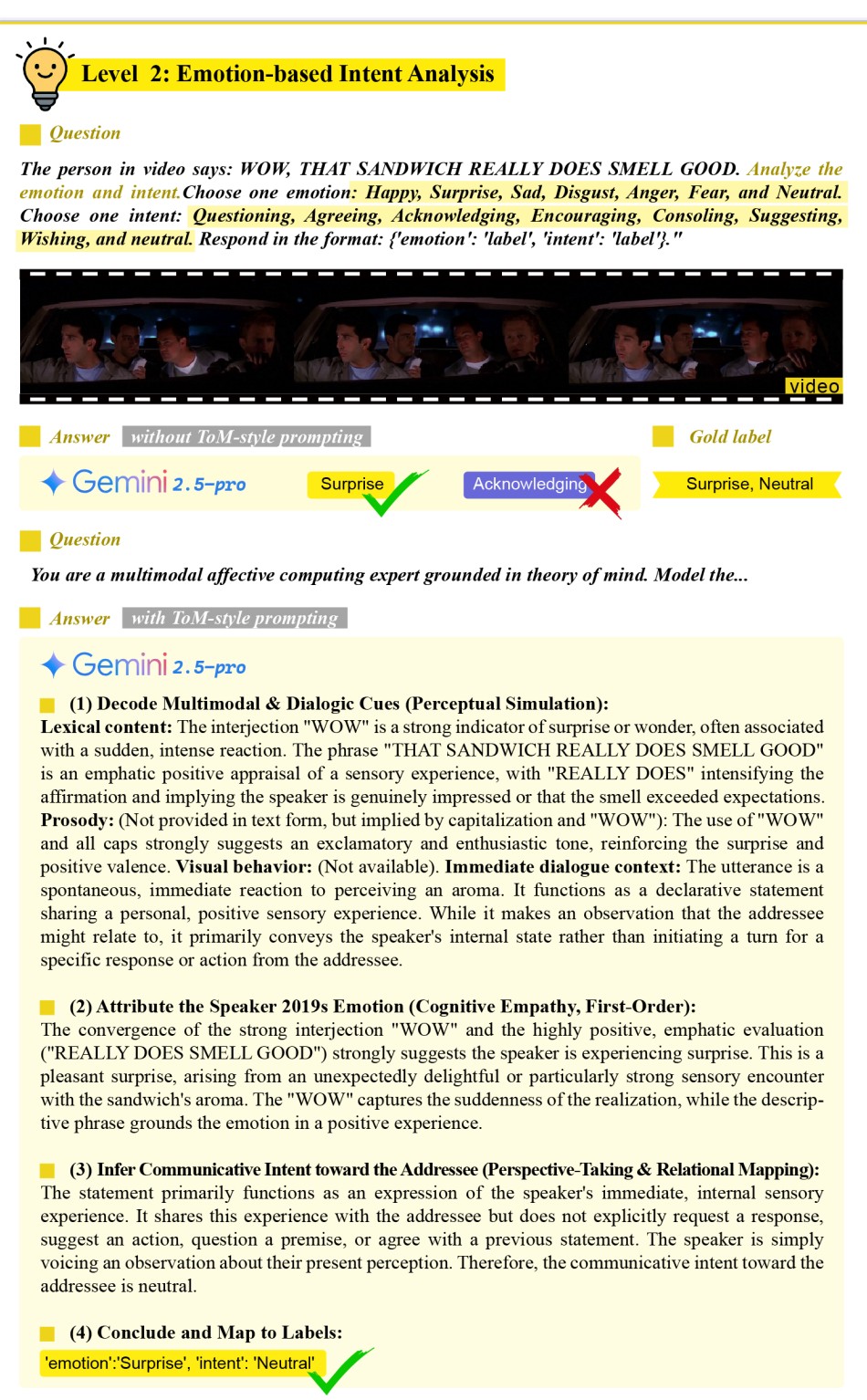

Figure 76: **Same model, direct answer vs. ToM prompting.**

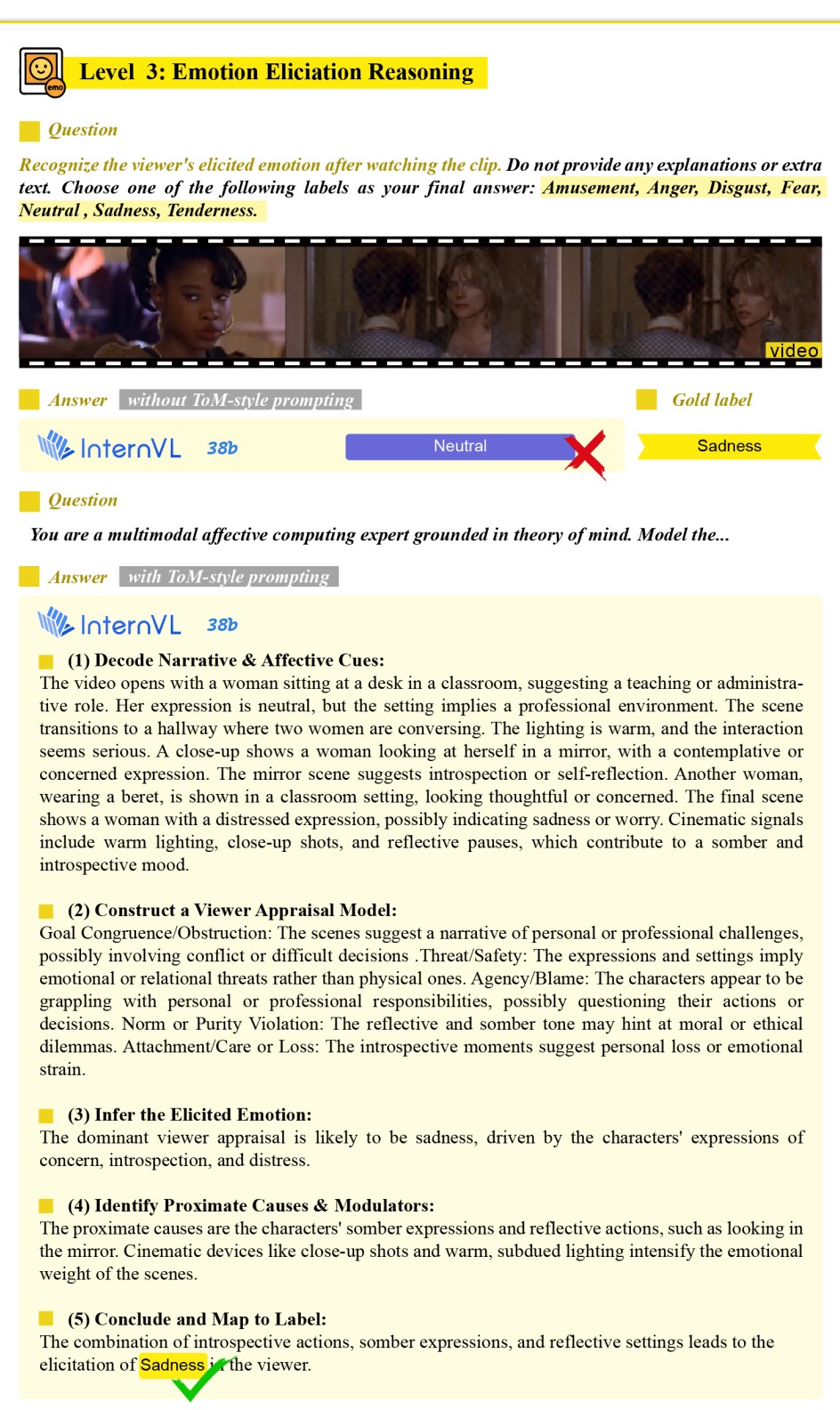

Figure 77: **Same model, direct answer vs. ToM prompting.**

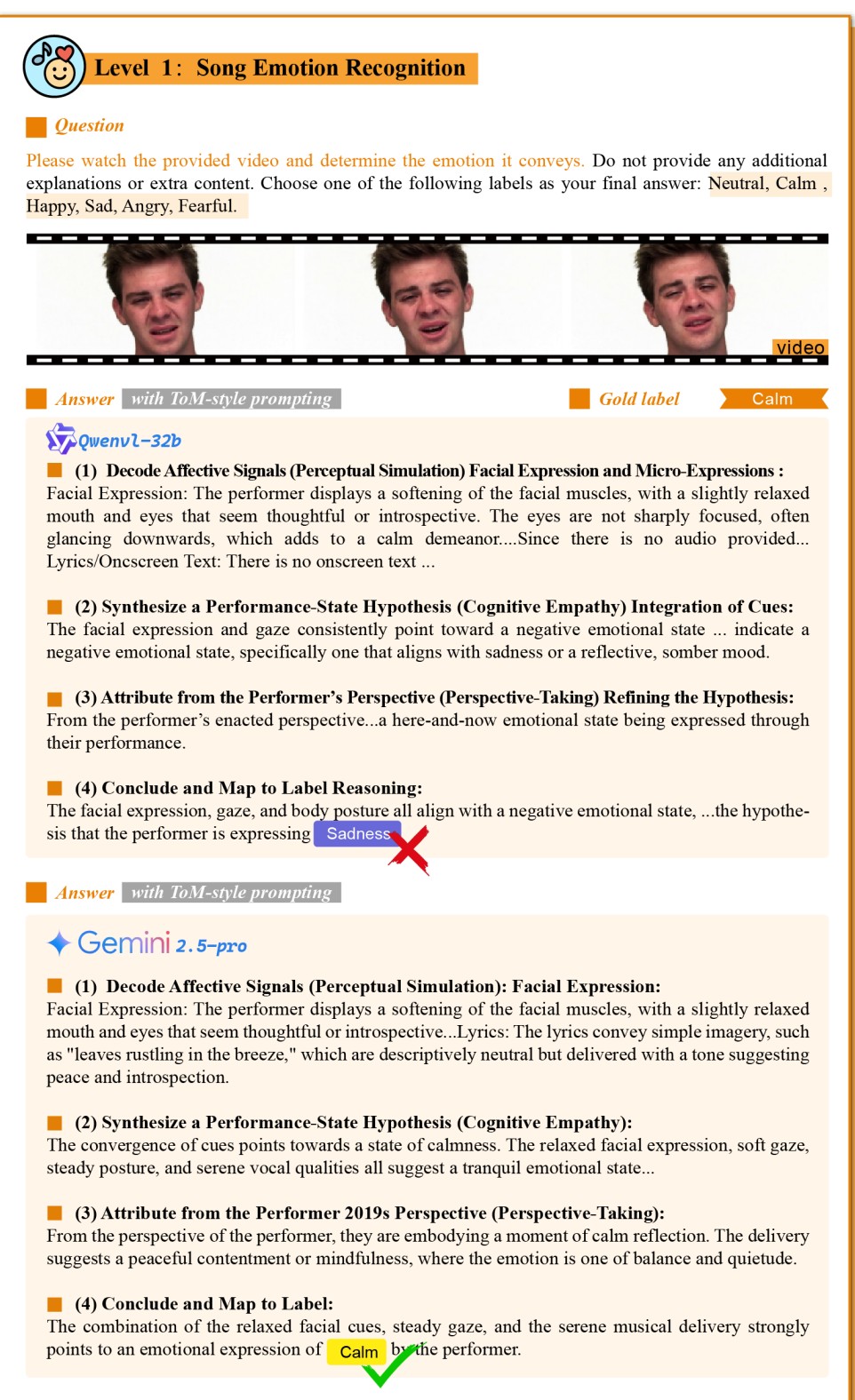

Figure 78: **ToM prompting answers from different models on the same QA.** We compare models' reasoning and decisions, illustrating cross-model differences in analytical paths and outcomes.

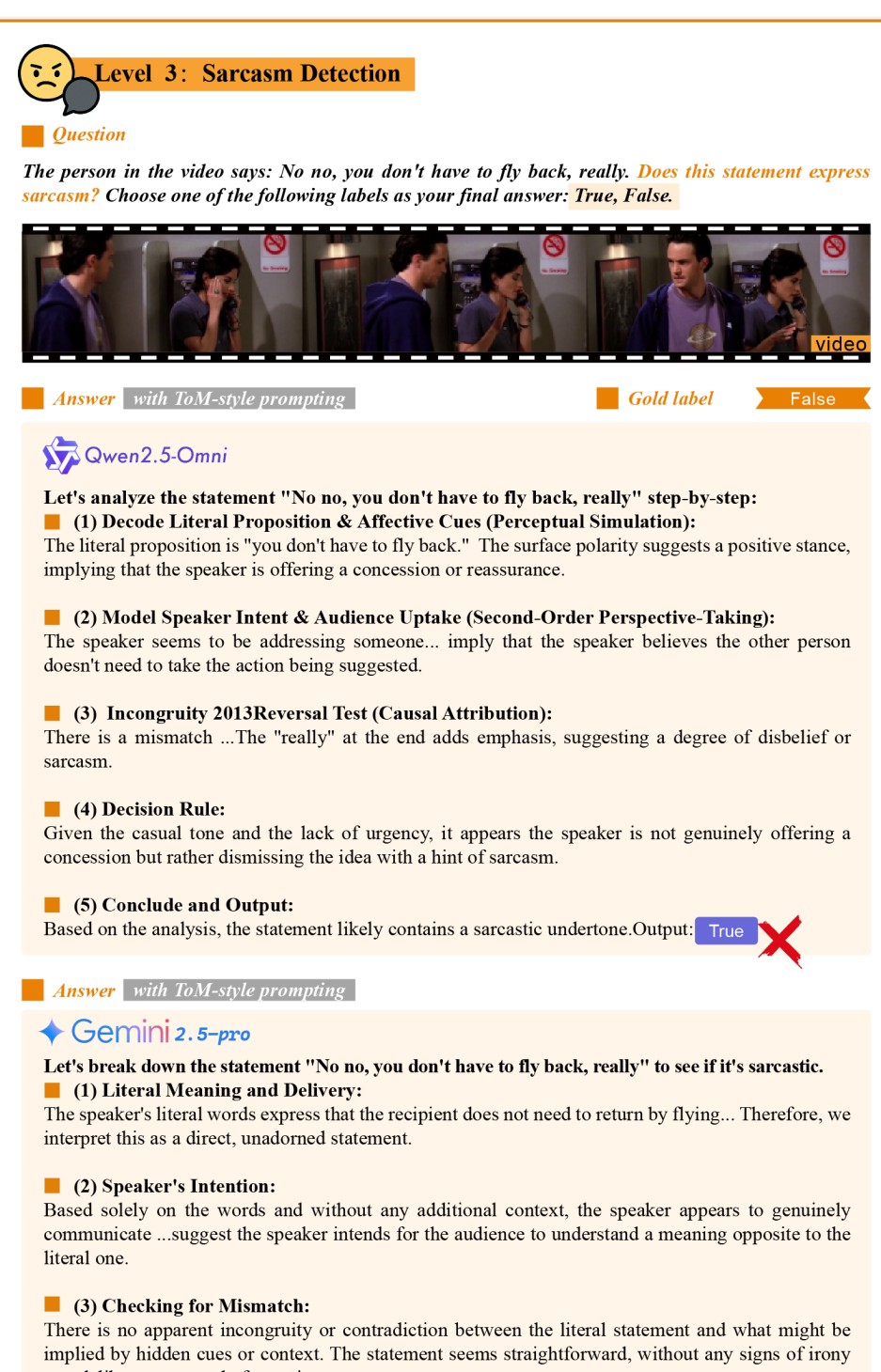

Figure 79: **ToM prompting answers from different models on the same QA.**

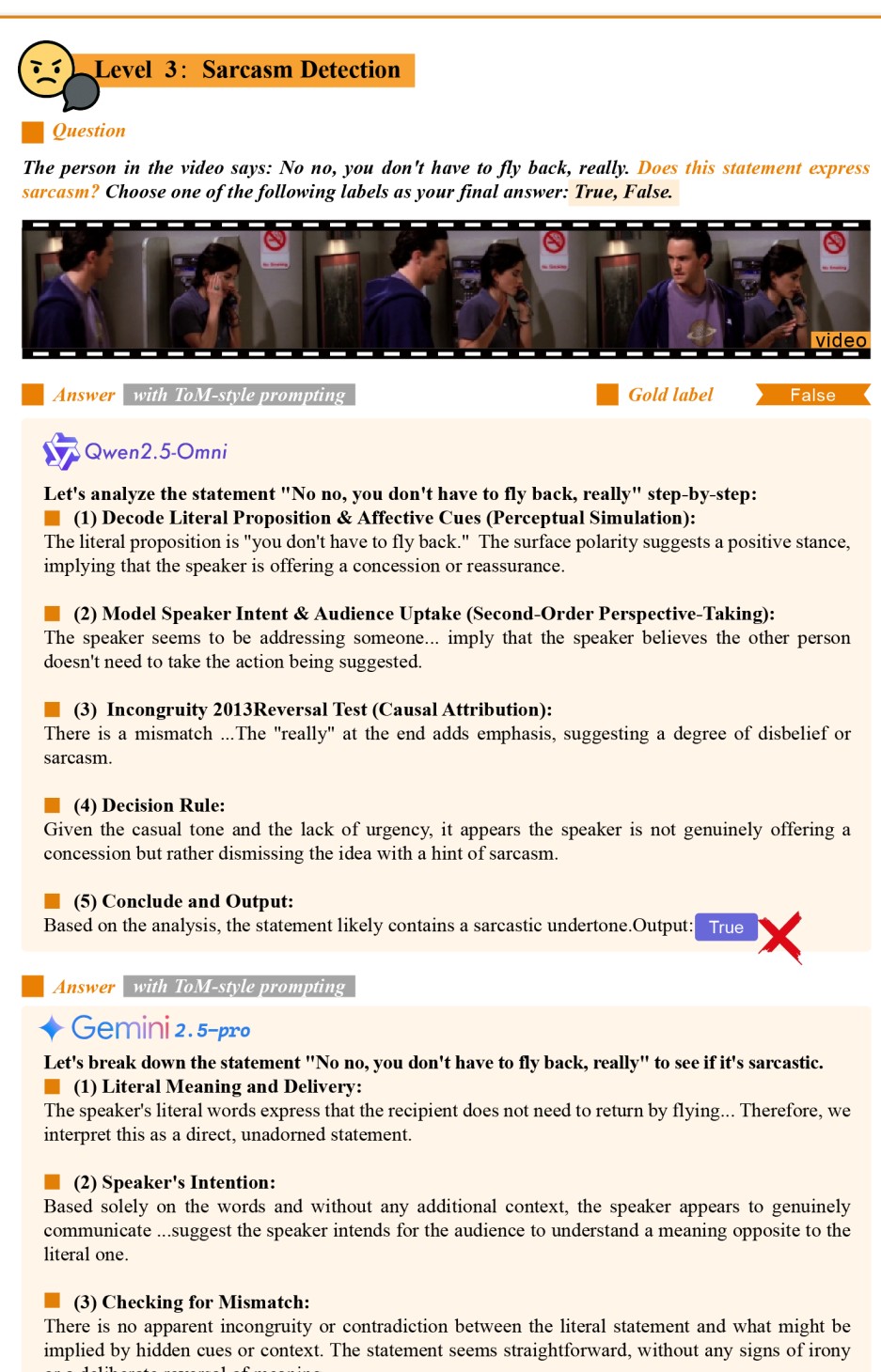

Figure 80: **ToM prompting answers from different models on the same QA.**

