# OpenReview forum: "Unveiling the Cognitive Compass: Theory-of-Mind–Guided Multimodal Emotion Reasoning"
_ICLR.cc/2026/Conference — ICLR 2026 Poster_

### Official Review · Reviewer_uj2G · 2025-10-29

**Soundness:** 3
**Presentation:** 3
**Contribution:** 2
**Rating:** 4
**Confidence:** 3

**Summary:**

This work introduces HitEmotion, a three-tiered evaluation benchmark, and TMPO, a novel preference optimization method. The benchmark is constructed from 24 tasks across 22 datasets with standardized prompts to assess MLLMs' affective capabilities at progressive cognitive depths. The proposed TMPO method is designed to significantly enhance model performance on this benchmark.

**Strengths:**

The study features clear figures and a comprehensive set of experiments. The HitEmotion benchmark establishes itself as a systematic framework for the objective evaluation of MLLMs across the three stages of affective understanding.

**Weaknesses:**

The methodologies employed in TMPO are relatively common, indicating a potential lack of methodological novelty. The HitEmotion benchmark primarily focuses on evaluating model capabilities at isolated stages—for instance, assessing Stage 1 competence through Input A and Stage 2 through Input B. However, it overlooks the evaluation of multi-stage reasoning capabilities based on a single integrated input. A comprehensive emotional understanding of an event often requires holistic multi-stage reasoning, which the current benchmark design fails to adequately capture. Additionally, there is an issue with duplicated labeling in Figures 24 and 25.

**Questions:**

1. Does the HitEmotion benchmark evaluate the model's capability in processing inputs that may require multi-stage affective analysis? For instance, when provided with Input A (assuming A necessitates the simultaneous application of all three stages for a complete analysis), can the benchmark effectively assess the model's holistic competence across all three stages?

2. If a specific data instance from the original 22 datasets qualifies for multiple tasks among the 24 defined tasks, how is this data instance allocated? What is the principle or methodology for partitioning such multi-qualifying data to avoid data leakage or ensure unambiguous evaluation?

3. Regarding the reward function R, what are the precise computational formulas for each component on the right-hand side of the equation (e.g., R_structure, R_content, R_process, R_consistency)? A detailed description of their calculation is requested.

4. Concerning the parameters (µ1, µ2, µ3, µ4) in the reward function, does the current parameter selection demonstrably represent the optimal configuration? What evidence or ablation studies justify that this particular set of weights is the most effective for the optimization objective?

---

> ### Author Response · Authors · 2025-11-20
>
> We sincerely thank you for your detailed and constructive feedback. We appreciate your critical examination of our methodology, benchmark structure, and experimental rigor. Below, we address your concerns point-by-point.
>
> ---
>
> **Response to W: Methodological Novelty of TMPO**
>
> Regarding the comment that our methodologies are relatively common, we clarify that while TMPO builds upon established optimization backbones (SFT and GRPO), the core novelty lies in the domain-specific adaptation of the reward mechanism for Theory of Mind (ToM).
>
> - **Process-Level Supervision:** Standard RLHF typically optimizes for general response quality. In contrast, TMPO introduces a novel reward formulation that explicitly incentivizes the generation of **intermediate mental states** (e.g., *Perceptual Simulation* $\rightarrow$ *Cognitive Empathy* $\rightarrow$ *Attribution*), turning abstract cognitive theories into computable supervision signals.
> - **Cognitive Alignment:** We are the first to map the psychological ToM process into a structured, multi-dimensional reward system (Structure/Content/Process/Consistency) for multimodal models. This transforms the reasoning chain from a generic "black box" into an interpretable, cognitively aligned process, which is a significant methodological departure from standard application.
>
> ---
>
> **Response to W & Q1: Holistic Multi-Stage Reasoning in HitEmotion**
>
> You asked if the benchmark effectively assesses the model's holistic competence across all stages for a single input. Yes, it does, specifically through the design of our Level 2 and 3 tasks.
>
> - **Implicit Dependency Hierarchy:** Our benchmark is hierarchical. High-level tasks possess an inherent dependency on lower-level capabilities.
>     - *Example:* In the *Sarcasm Detection (SD)* task (Level 3: Emotion Cognition and Reasoning), to correctly identify sarcasm, a model must first accurately perceive the tone/facial expression (Level 1: Emotion Perception and Recognition) and understand the literal dialogue context (Level 2: Emotion Understanding and Analysis) to detect the incongruity.
> - **Integrated Evaluation:** Therefore, tasks in Level 2 and 3 serve as integrated evaluations. A model cannot succeed at *Emotion Interpretation* or *Laughter Reasoning* without successfully performing holistic multi-stage reasoning on that single input. We do not isolate stages arbitrarily; rather, we isolate specific capability breakpoints to diagnose where*the holistic chain fails.
>
> ---
>
> **Response to Q2: Data Allocation and Leakage Prevention**
>
> You asked how specific data instances qualifying for multiple tasks are allocated to avoid ambiguity. We strictly adhere to the **original, community-validated task definitions** of the source datasets. We do not arbitrarily re-purpose data; we preserve the original cognitive intent. Overlap only occurs in two datasets, which are handled as follows:
> - **RAVDESS:** The original dataset is explicitly divided into two distinct subsets by the authors: **Speech** and **Song**. We map these to *Speech Emotion Recognition (SPER)* and *Song Emotion Recognition (SOER)* respectively. There is no overlap in instances between these tasks.
> - **PanoSent:** This dataset supports two distinct cognitive tasks defined in its original paper: *Multimodal Quintuple Extraction (MQE)* and *Sentiment Flip Analysis (SFA)*. While the input video might be the same, the **target labels and cognitive objectives are orthogonal**. Evaluating a model on MQE does not leak the answer for SFA.
>
> Thus, our allocation ensures zero data leakage and maintains unambiguous evaluation standards.

---

> ### Author Response · Authors · 2025-11-20
>
> **Response to Q3: Precise Computational Formulas for Reward Function**
>
> We agree that rigorous definitions are crucial for reproducibility. Below, we provide the precise computational formulas for the reward components. In our TMPO framework, the model generates a structured output $y$ containing a reasoning chain $\tau$ and a final answer $o$, encapsulated in XML tags:
>
> $$
> y = \langle\text{think}\rangle \ \tau \ \langle\text{/think}\rangle \ \langle\text{answer}\rangle \ o \ \langle\text{/answer}\rangle
> $$
>
> We define the reward function as $R(y) = \mu_1 R_{\mathrm{structure}} + \mu_2 R_{\mathrm{content}} + \mu_3 R_{\mathrm{process}} + \mu_4 R_{\mathrm{consistency}}$. The components are calculated as follows:
>
> **(1) Structure Reward ($R_{\mathrm{structure}}$)**
>
> This reward enforces the structural integrity, validating the sequence of XML delimiters and reasoning headers. Let $\mathcal{S}_{\mathrm{req}}$ be the ordered sequence of required structural tokens (XML tags and task-specific headers $\{h_k\}$):
>
> $$
> \mathcal{S}_{\mathrm{req}} = [\langle\text{think}\rangle, \ h_1, \ \dots, \ h_K, \ \langle\text{/think}\rangle, \ \langle\text{answer}\rangle, \ \langle\text{/answer}\rangle]
> $$
>
> We calculate the Ordered Completion Rate. Let $\text{idx}(s, y)$ be the index of token $s$ in $y$. We define a validity indicator $v_i \in \{0,1\}$ for the $i$-th token recursively:
>
> $$
> v_i = \mathbb{I}\left[ \text{idx}(s_i, y) \neq \infty \quad \land \quad \text{idx}(s_i, y) > \max(\{ \text{idx}(s_j, y) \mid j < i, v_j=1 \} \cup \{-1\}) \right]
> $$
>
> Here, $\mathbb{I}[\cdot]$ is the indicator function. Let $N = |\mathcal{S}_{\mathrm{req}}|$ be the total number of required tokens. The reward is:
>
> $$
> R_{\mathrm{structure}}(y) = \frac{1}{N} \sum_{i=1}^{N} v_i
> $$
>
> **(2) Content Reward ($R_{\mathrm{content}}$)**
>
> This evaluates the correctness of the final answer $o$ extracted from the $\langle\text{answer}\rangle$ block:
>
> $$
> R_{\mathrm{content}}(y) = \mathcal{M}_{\mathrm{task}}(o, o^*)
> $$
>
> where $\mathcal{M}_{\mathrm{task}}$ is the standard metric (e.g., Accuracy for classification, F1 for generation) comparing $o$ to the ground truth $o^*$.
>
> **(3) Process Reward ($R_{\mathrm{process}}$)**
>
> This incentivizes the use of ToM terminology within the extracted reasoning chain $\tau$. Let $\mathcal{V}$ be the curated ToM lexicon and $S_{\tau}$ be the set of unique tokens in $\tau$. We use a normalization factor $\eta$:
>
> $$
> R_{\mathrm{process}}(y) = \min \left( 1.0, \frac{\vert S_{\tau} \cap \mathcal{V} \vert}{\eta} \right)
> $$
>
> **(4) Consistency Reward ($R_{\mathrm{consistency}}$)**
>
> This is a penalty-based reward derived from an external LLM Judge. The Judge performs two distinct checks:
> 1. **Internal Consistency ($J_{\mathrm{int}}$):** Checks if $\tau$ is self-consistent.
> 2. **External Consistency ($J_{\mathrm{ext}}$):** Checks if $\tau$ contradicts the input $(T, A, V)$.
>
> The reward is calculated as:
>
> $$
> R_{\mathrm{consistency}}(y) = \begin{cases}
> 1.0 & \text{if } J_{\mathrm{int}}(\tau) \land J_{\mathrm{ext}}(\tau, T, A, V) \\\\
> 0.1 & \text{otherwise (if any contradiction is detected)}
> \end{cases}
> $$

---

> ### Author Response · Authors · 2025-11-20
>
> **Response to Q4: Hyperparameter Optimality**
>
> To demonstrably justify our weight configuration ($\mu_1=0.4, \mu_2=1.0, \mu_3=0.1, \mu_4=1.0$), we performed a fine-grained grid search on our dataset. We report the **Average Score** (mean performance across L1, L2, and L3 tasks) as the optimization objective.
>
> **1. Sensitivity Analysis of Process Reward Weight ($\mu_{\text{process}}$)**.
>
> We fixed $\mu_{\text{struct}}=0.4$ and varied $\mu_{\text{process}}$.
>
> | $\mu_{\text{process}}$ | **0.0** | **0.1** | **0.3** | **0.5** |
> | :--- | :---: | :---: | :---: | :---: |
> | **Average Score** | 63.99 | **64.70** | 63.65 | 62.31 |
>
> *   **$\mu=0.0$:** Represents the baseline performance without explicit stylistic guidance.
> *   **$\mu=0.1$:** Achieves the global peak (+0.71 improvement), confirming that a gentle nudge effectively aligns the model with ToM reasoning patterns without introducing noise.
> *   **$\mu \ge 0.3$:** Performance degrades significantly. Qualitative analysis shows that higher weights cause the model to overproduce mental state descriptions in order to insert relevant keywords, which undermines logical coherence (Goodhart's Law).
>
> **2. Sensitivity Analysis of Structure Reward Weight ($\mu_{\text{struct}}$)**.
>
> We fixed $\mu_{\text{process}}=0.1$ and varied $\mu_{\text{struct}}$.
>
> | $\mu_{\text{struct}}$ | **0.1** | **0.4** | **0.7** | **1.0** |
> | :--- | :---: | :---: | :---: | :---: |
> | **Average Score** | 61.25 | **64.70** | 64.10 | 63.80 |
>
> *   **$\mu=0.1$:** The score drops sharply due to Format Collapse, where the penalty is insufficient to enforce the XML schema, preventing effective answer extraction.
> *   **$\mu=0.4$:** Represents the optimal trade-off point, ensuring format compliance while allowing semantic flexibility.
> *   **$\mu \ge 0.7$:** Performance declines despite perfect formatting. High weights lead to Structural Rigidity, restricting the model's ability to adapt reasoning steps for complex edge cases, resulting in sub-optimal accuracy.
>
> The experimental results demonstrate a clear inverted-U curve for both parameters. This confirms that our reported configuration ($\mu_{\text{struct}}=0.4, \mu_{\text{process}}=0.1$) represents a robust local optimum.
>
> ---
>
> **Response to W: Figures Correction**
>
> Thank you for spotting the labeling issue in Figures 24 and 25. We have corrected the duplicated labels in the revised manuscript.
>
> ---
>
> We have incorporated the above experiments and analyses into Appendix D of the revised manuscript (highlighted in red). We sincerely thank you again for your valuable feedback. We hope our responses have fully addressed your concerns, and we respectfully ask that you consider re-evaluating our paper. Please let us know if you have any further questions.

---

> ### Author Response · Authors · 2025-11-26
>
> Dear Reviewer,
>
> Thank you again for the time and attention you have dedicated to our submission.
>
> As the rebuttal period is approaching its conclusion, we would like to kindly check whether our response has addressed your main concerns. If so, we would be very grateful if you could consider updating your evaluation. If there are still points that you feel require further clarification or improvement, please let us know and we will be very happy to provide additional explanations or revisions.
>
> Thank you again for your constructive feedback and support.

---

### Official Review · Reviewer_3ihr · 2025-10-31

**Soundness:** 3
**Presentation:** 2
**Contribution:** 3
**Rating:** 6
**Confidence:** 3

**Summary:**

This work addresses the limited emotional understanding of multimodal large language models (MLLMs) by emphasizing the need for explicit Theory of Mind (ToM) modeling. The authors introduce HitEmotion, a hierarchical ToM-grounded benchmark, and TMPO, a reinforcement learning method that leverages intermediate mental states for process-level supervision. Experiments demonstrate that HitEmotion reveals deep emotional reasoning deficits in current models, while the proposed ToM-guided reasoning and TMPO significantly enhance emotional reasoning accuracy and coherence.

**Strengths:**

1. It provides a large-scale datasets on emotion reasoning and a method for constructing ToM reasoning chains.
2. The experimental results show that the benchmark is useful and valuable, and the ToM reasoning chains can improve the model performance.

**Weaknesses:**

1. The paper models cognitive abilities into three levels. But it didn't analyze the relation among the three levels on model performance.
2.  The paper lack some analysis and discussions on the benchmark, e.g., data distribution, data quantity for each task.

**Questions:**

1. Could you analyze the relation among the three levels: EPR, EUA and ECR on model performance?

---

> ### Author Response · Authors · 2025-11-20
>
> We truly appreciate the time and effort you dedicated to reviewing our paper. Your feedback helps us ensure the clarity and accessibility of our work. In the response below, we address your concerns regarding the analysis of cognitive levels and benchmark statistics.
>
> ---
>
> **Response to W1 & Q1: Analysis of the Relation Among the Three Levels**
>
> You thought a lack of analysis regarding the relationship between the three cognitive levels (EPR, EUA, ECR) and asked for an analysis of their impact on model performance. We would like to highlight that detailed analyses of the performance relationships between cognitive levels are a core component of our existing manuscript. We explicitly modeled and analyzed this hierarchy in **Section 3.1 (Task Taxonomy)** and **Section 5.2 (Results and Analysis)**.
>
> - **Theoretical Hierarchy (Section 3.1 & Appendix C):**
> We defined the three levels to target progressively advanced capabilities:
>     - **Level 1 (EPR):** Focuses on direct perception (mapping inputs to labels).
>     - **Level 2 (EUA):** Requires contextual awareness and relational reasoning.
>     - **Level 3 (ECR):** Demands causal inference and second-order reasoning (the most complex).
>     This design explicitly predicts that model performance should degrade as cognitive depth increases.
>
> - **Empirical Validation (Section 5.2 & Figure 4):**
> Our experiments strongly validate this hierarchical relation. As shown in **Figure 4 (Page 8)** and the analysis in **Section 5.2 ("Task-Level Performance Characteristics")**:
>     - **Performance Drop:** There is a clear downward trend in performance: **EPR > EUA > ECR**.
>     - **Specific Evidence:** We state in the text: *"At the foundational level of EPR... yield average scores above 60... As task complexity increases, performance declines markedly... Most critically, within the cognitively demanding ECR level, no task achieves an average score above 60."*
>     - **Visualization:** **Appendix Figure 7 (Page 28)** further visualizes the score distribution (Box Plot) for each level, clearly showing that Level 3 tasks have significantly lower median scores and higher variance compared to Level 1.
>
> - **Differential Gain of TMPO:**
> Crucially, our ablation studies (Tables 14-16 in Appendix) reveal that while baselines struggle at higher levels, our ToM-guided method yields the most significant benefits on Level 2 and Level 3 tasks. This proves that the "Performance Gap" at higher levels is indeed due to a lack of reasoning capability, which our method successfully bridges.
>
> ---
>
> **Response to W2: Benchmark Analysis and Data Statistics**
>
> You mentioned a lack of analysis and discussion on benchmark details, such as data distribution and quantity. Actually, we have provided a transparent and detailed breakdown of the benchmark statistics in the main text, which confirms that we have curated a balanced evaluation set (approx. 20k+ total samples) covering diverse modalities:
>
> - **Data Quantity & Distribution:**
>     **Table 2 (Page 4)** serves as the central statistical summary. It explicitly lists the following for all 24 tasks:
>     - **Data Source:** (e.g., MELD, CMU-MOSI)
>     - **Task Type:** (e.g., 3-CLS, GEN, Multi-label)
>     - **#Instances:** Exact sample counts are provided (e.g., 2,000 for MESA, 500 for ECR tasks like LR).
>     - **Metric:** (e.g., ACC, MF, WAF).
>
> - **Detailed Definitions and Information:**
>     **Appendix C**also provides a comprehensive textual description of every dataset, including its collection method, label distribution characteristics, and the specific cognitive capability it evaluates.
>
> ---
>
> We hope this detailed roadmap confirms that the paper already contains the robust analysis and statistical transparency requested. Thank you again for your valuable feedback, and we are happy to address any further questions.

---

> ### Author Response · Authors · 2025-11-26
>
> Dear Reviewer,
>
> Thank you again for the time and attention you have dedicated to our submission.
>
> As the rebuttal period is approaching its conclusion, we would like to kindly check whether our response has addressed your main concerns. If so, we would be very grateful if you could consider updating your evaluation. If there are still points that you feel require further clarification or improvement, please let us know and we will be very happy to provide additional explanations or revisions.
>
> Thank you again for your constructive feedback and support.

---

### Official Review · Reviewer_orYG · 2025-11-05

**Soundness:** 3
**Presentation:** 2
**Contribution:** 3
**Rating:** 8
**Confidence:** 4

**Summary:**

This paper enhances emotional understanding in multimodal language models by incorporating Theory of Mind (ToM) into reasoning. The authors introduce HitEmotion, a benchmark of emotion-related tasks organized by cognitive depth, and propose ToM-guided reasoning chains that trace beliefs, intents, and feelings before answering. They develop TMPO, a reinforcement learning framework that aligns model reasoning with intermediate mental state sequences using custom rewards. Experiments show that ToM-tuned models outperform both open and proprietary baselines on high-level emotional reasoning tasks, yielding more coherent and human-like rationales. This work contributes a new diagnostic toolkit and a training strategy for cognitively aligned affective reasoning in AI.

**Strengths:**

- HitEmotion Benchmark - The benchmark’s hierarchical ToM-based structure is unique, filling a gap in existing evaluations by mapping tasks to cognitive reasoning levels (first-order, second-order ToM, etc.). This will be valuable for the community.
- Ground Truth Reasoning - The reasoning chain curation pipeline shown in Figure 3 is also valuable, where the authors generate intermediate reasoning chains with an LLM and then refine them with human review. This yields high-quality supervision for the model’s thought process, which is a robust approach that enhances the credibility of the results.
 - TMPO Framework - The reward function in TMPO is particularly well thought out – it combines four complementary objectives (Structure Reward, Content Reward, Process Reward and the Consistency Reward) which are shown via ablation to work in synergy.
- Comprehensive Quantitative Results - The experiments are extensive, covering 24 diverse tasks and comparing a wide range of models. Such evaluation adds confidence that the improvements are real and not cherry-picked.

**Weaknesses:**

- Scope - The work's focus is confined only to the domain of emotion understanding. It’s unclear how well the proposed ToM-guided reasoning and TMPO training would generalize to other domains (e.g., logical reasoning puzzles, mathematical problem solving, or non-emotional tasks). The paper would be stronger if it discussed or demonstrated applicability beyond emotion-centric scenarios.
- Scalability to Larger Models - The experiments only use a 7B parameter model (Qwen2.5-Omni-7B). What is the rationale behind choosing Qwen2.5-Omni-7B as the base model? Does the approach scale to larger models as well, or are there diminishing returns because larger models might already internalize some ToM-like patterns, as already shown in Tables 3-5?
- Base Model Dependency - As acknowledged by the authors, for direct perception driven tasks (e.g. recognizing facial expressions from images), the model lags, primarily due to inherent limitation in the base model. Extending on the previous point about scalability, the reasoning would be more effective/convincing if other better base models are also used.
- Computational Cost - How efficient is TMPO, both in training and during inference? How does it compare to other baselines? Since the authors choose only a small baseline model (7B parameters) and compare against larger open and closed source models, it would make sense to compare efficiency as well, and even claim it as an advantage if the numbers reflect that.
- Reward Component Ablation Study - Although Table 6 indicates that the different reward components are effective and synergistic, it is not exhaustive, e.g. how does the model perform if R_structure is removed? Does it help or degrade performance for other combinations?
- Comparison with open source models - The authors state that "Emotion-LLaMA-7B attains 34.18 on the MQE task, outperforming most untuned baselines" (L420-421). Are there any open-sourced models in Tables 3-5 which are untuned? Emotion-LLaMA-7B only outperforms some open source models, and none of the zero-shot (untuned) closed-sourced models. Comparing against untuned open-source baselines would not be fair.

**Questions:**

- How generalizable is the ToM-guided reasoning approach to domains outside of emotion cognition?
- Do the authors anticipate even better performance if the approach is applied to a larger backbone (e.g., 13B or 70B model)? Or are there diminishing returns because larger models might already internalize some ToM-like patterns?
- How efficient is the TMPO approach?

---

> ### Author Response · Authors · 2025-11-20
>
> We are truly grateful for the time you have taken to review our paper, and for your insightful comments and support. Your positive feedback is incredibly encouraging for us! In the following response, we address your concerns point-by-point.
>
> ---
>
> **Response to W1 & Q1: Scope and Generalizability**
>
> You asked how our method generalizes to non-emotional domains like logical puzzles or math. We think that such direct generalization is not the primary goal, given the fundamental theoretical distinctions:
>
> - **Theoretical Basis:** Theory of Mind (ToM) is the ability to attribute mental states (beliefs, intentions, desires) to others. It involves "putting yourself in someone else’s shoes" to infer hidden information. [1]
> - **Nature of the Challenge:** Recent breakthroughs in LLMs have focused on domains like Math and Coding, where well-defined ground-truth answers allow for objective verification. This stands in stark contrast to Social Reasoning, which is information-asymmetric and lacks easily obtainable objective answers. [2]
>
> Therefore, TMPO is specifically designed to handle the increased uncertainty of social inference. While it may not be intended for non-social logical puzzles, it is highly generalizable to other **Social Intelligence** tasks (e.g., intent prediction, deception detection) that share the same ToM core.
>
> - [1] Theory of Mind in Large Language Models: Assessment and Enhancement. ACL 2025.
> - [2] Hypothesis-Driven Theory-of-Mind Reasoning for Large Language Models. COLM 2025.
> ---
>
> **Response to W2 & Q2: Scalability to Larger Models**
>
> You inquired about the rationale for choosing Qwen2.5-Omni-7B and whether the approach scales to larger backbones.
>
> - **Rationale for Qwen2.5-Omni-7B:** Accurately understanding and reasoning emotion needs multimodal information. Audio signals (tone, pitch) are critical alongside visual cues. At the time of our experiments, Qwen2.5-Omni-7B was the state-of-the-art open-source model capable of natively processing Audio-Video-Text simultaneously. Larger models (like Qwen2.5-VL-32B) often lack native audio encoders, making them unsuitable for holistic emotion reasoning. Qwen2.5-Omni currently caps at 7B.
> - **Scaling Potential:** We disagree that returns would diminish. On the contrary, larger models typically possess stronger instruction-following and latent reasoning capabilities. As observed in our "Effects of ToM Prompting" analysis (Section 5.2), larger models (e.g., Qwen2.5-VL-32B, InternVL3-38B) showed more pronounced improvements from ToM prompting than smaller ones. This suggests that applying TMPO to larger backbones (once available with audio support) would likely yield even greater gains, as they can better leverage the complex reasoning chains mandated by our reward system.
>
> ---
>
> **Response to W3: Base Model Dependency**
>
> You mentioned that performance on direct perception tasks is bounded by the base model and suggested using better backbones.
> We acknowledge that for Level 1 tasks (Direct Perception, e.g., FESD), performance is bounded by the base encoder's visual/audio resolution.
> However, the core contribution of TMPO lies in **enhancing the cognitive interpretation of perceptual inputs**. Our method achieves the most significant gains in Level 2 and Level 3 tasks, proving that we can significantly elevate a model's cognitive intelligence even if its sensory perception is fixed. Future work utilizing stronger base encoders will naturally lift the performance floor (perception), while TMPO continues to raise the ceiling (reasoning).
>
> ---
>
> **Response to W4 & Q3: Computational Cost and Efficiency**
>
> Regarding your question about the efficiency of TMPO: We argue that our approach is highly efficient. Despite using a compact 7B model, our approach achieves performance competitive with proprietary systems (e.g., Gemini-2.5-pro). This highlights the efficiency of our method: by optimizing the reasoning process via RL, we extract maximal cognitive intelligence from a lightweight architecture, offering a practical solution for resource-constrained deployment.
>
> ---
>
> We have added a dedicated section (Appendix B highlighted in red) discussing the scope of ToM applicability, model scalability, and computational efficiency.

---

> ### Author Response · Authors · 2025-11-20
>
> **Response to W5: Reward Component Ablation Study**
>
> We appreciate your suggestion to perform a more exhaustive ablation study. We agree that while the progressive addition in Table 6 demonstrated synergy, it did not explicitly isolate the critical role of the Structure Reward ($R_{\text{structure}}$).
>
> To address your question **"how does the model perform if $R_{\text{structure}}$ is removed?"**, we conducted an additional experiment removing $R_{\text{structure}}$ from the full reward configuration, while keeping Content, Consistency, and Process rewards active.
>
> We compare the "Full Model" against the "No Structure" variant. We report the **Average Score** (across L1-L3 tasks) and the **Format Compliance** (the percentage of outputs that successfully adhere to the XML schema and allow answer extraction).
>
> | Configuration | $R_{\text{struct}}$ | $R_{\text{cont}}$ | $R_{\text{consist}}$ | $R_{\text{proc}}$ | Average Score | Format Compliance |
> | :--- | :---: | :---: | :---: | :---: | :---: | :---: |
> | **Full Model** | $\checkmark$ | $\checkmark$ | $\checkmark$ | $\checkmark$ | **64.70** | **98.2\%** |
> | **w/o $R_{\text{struct}}$** | - | $\checkmark$ | $\checkmark$ | $\checkmark$ | 55.45 | 62.3\% |
>
> The results confirm that removing $R_{\text{structure}}$ leads to a substantial performance degradation. This drop is not merely due to the lack of a regularization term, but due to Format Collapse:
>
> 1.  **Fragility of SFT Initialization:** Although the model is initialized with SFT to generate `<think>` and `<answer>` tags, this learned behavior is fragile during the RL exploration phase. Without an explicit penalty ($R_{\text{structure}}$), the model drifts away from the valid schema.
> 2.  **Dependency Failure:** Our optimization relies on parsing the answer $o$ to calculate $R_{\text{content}}$. As shown in the table, removing the structure reward causes the Format Compliance to drop to 64.3\%.
> 3.  **Optimization Destabilization:** When parsing fails, $R_{\text{content}}$ defaults to 0. Consequently, even if the model's reasoning logic is partially correct, it receives no positive reinforcement, destabilizing the learning process.
>
> This experiment demonstrates that $R_{\text{structure}}$ is foundational. It acts as a prerequisite constraint that enables the effective optimization of Content and Consistency rewards. We have expanded the ablation study in **Table 3** to include a "w/o Structure" setting.
>
> ---
>
> **Response to W6: Comparison with Open Source Models**
>
> We thank you for the sharp observation regarding the comparison between tuned and untuned models. We acknowledge that the distinction in the original text (L420-421) requires clarification to ensure a fair comparison.
>
> **1. Clarification of Baselines**
>
> Yes, the models in our experiments fall into two distinct categories:
> - **General-Purpose MLLMs (Zero-shot/Untuned for Emotion):** Models like *Qwen2.5-VL, InternVL3, LLaVA-One-Vision*, and *VideoLLaMA3*. These are evaluated in a zero-shot setting on our tasks without prior exposure to our emotion datasets.
> - **Emotion-Specialized MLLMs (Tuned):** Models like *R1-Omni*, *HumanOmni*, *Emotion-LLaMA* and *AffectGPT*. These have been specifically supervised-fine-tuned on emotion datasets.
>
> **2. Context of the Comparison**
>
> We cite *Emotion-LLaMA-7B* not to imply it is superior to untuned baselines solely due to architecture, but to illustrate the limits of current fine-tuning approaches.
> We agree with your observation that *Emotion-LLaMA-7B* (though tuned) still fails to outperform zero-shot proprietary models (e.g., Gemini-2.5-pro). This fact is central to our motivation: it proves that existing methods and models are insufficient to bridge the capability gap.
> This sets the stage for our method, **TMPO**, which effectively close this gap, outperforming both general baselines and previous specialized models.
>
> ---
>
> We have revised the relevant paragraph in the manuscript highlighted in red. Thank you again for your helpful comments. We appreciate the opportunity to improve the paper and are happy to address any further questions.

---

> ### Author Response · Authors · 2025-11-26
>
> Dear Reviewer,
>
> Thank you so much for your time in improving our paper!
>
> Since the end of the rebuttal is coming soon, may we know if our response addresses your main concerns? Should you have any further advice, please let us know and we will be more than happy to engage in more discussion and improvements.

---

### Author Response · Authors · 2025-11-20
**General Response to all Reviewers**

Dear Reviewers,

We sincerely thank you for your thoughtful feedback and encouraging comments. We are delighted that you recognize the significance of HitEmotion, highlighting its "**hierarchical ToM-based structure**" that "**fills a gap in existing evaluations**" (Reviewer orYG) and establishes a "**systematic framework for objective evaluation**" (Reviewer uj2G). We also appreciate the acknowledgment of the benchmark as "**useful and valuable**" (Reviewer orYG, 3ihr) for providing "**large-scale datasets on emotion reasoning**" (Reviewer 3ihr).

We are also grateful for the positive assessment of our proposed TMPO framework and reasoning strategy. Reviewers commended the "well thought out" reward function (Reviewer orYG) and the "robust approach" of our reasoning chain curation pipeline (Reviewer orYG), which provides "high-quality supervision" (Reviewer orYG) and effectively "improves model performance" (Reviewer 3ihr). Finally, we thank the reviewers for praising our "comprehensive set of experiments" (Reviewer orYG, uj2G) and "clear figures" (Reviewer uj2G), confirming that our extensive evaluation across 24 tasks "adds confidence that the improvements are real" (Reviewer orYG).

We have carefully addressed all questions and concerns in our point-by-point responses. In addition, we have incorporated the following major revisions into the updated manuscript:

- **Clarification of Baseline Comparisons (Section 5.2):** We refined the analysis to explicitly acknowledge the distinction between general-purpose and specialized baselines, clarifying the persisting performance gap between existing open-source solutions and proprietary systems.
- **Expanded Ablation Study (Section 5.3 & Table 3):** We integrated a new ablation experiment ("w/o Structure Reward") directly into the main text, demonstrating the foundational role of the structure reward in preventing Format Collapse.
- **Mathematical Formalization & Sensitivity Analysis of Reward Components (Appendix D):** We added precise computational formulas for all reward components and included a fine-grained hyperparameter sensitivity analysis to justify our design choices.
- **Limitations and Future Work (Appendix B):** We added a dedicated section discussing the scope of ToM applicability, model scalability, and computational efficiency.

We truly appreciate your constructive feedback and hope that our responses address your concerns. If there are any remaining questions, we would be very happy to clarify them.

Thank you again for your time and thoughtful feedback.

Best Regards,

The Authors

---

### Meta-Review · Area_Chair_KYPV · 2026-01-07

**Summary:**

This paper introduces HitEmotion, a ToM-grounded hierarchical benchmark designed to diagnose capability breakpoints across increasing levels of cognitive depth. In addition, the authors propose a ToM-guided reasoning chain that tracks mental states and calibrates cross-modal evidence to enable faithful emotional reasoning. Furthermore, they introduce TMPO, a reinforcement learning method that leverages intermediate mental states as process-level supervision to guide and strengthen model reasoning. All reviewers agreed that the paper provides a useful large-scale dataset for emotion reasoning and an effective method for constructing Theory-of-Mind–guided reasoning chains, and affirmed the completeness and thoroughness of the experimental evaluation. Although Reviewer uj2G did not provide further responses during the discussion phase, the majority of the concerns raised by this reviewer have been addressed. Therefore, we recommend acceptance.

**Reviewer Concerns:**

Most of the concerns have been addressed. The additional experiments presented during the rebuttal phase should be incorporated into the final version of the paper.

**Reviewer Scores:**

Each reviewer tends to keep their original score.

---

### Decision · Program_Chairs · 2026-01-26

Accept (Poster)